# STEP-TAGGING: TOWARD CONTROLLING THE GENERATION OF LANGUAGE REASONING MODELS THROUGH STEP MONITORING

## ABSTRACT

The field of Language Reasoning Models (LRMs) has been very active over the past few years with advances in training and inference techniques enabling LRMs to reason longer, and more accurately. However, a growing body of studies show that LRMs are still inefficient, over-generating verification and reflection steps. To address this challenge, we introduce the Step-Tagging framework, a lightweight sentence-classifier enabling real-time annotation of the type of reasoning steps that an LRM is generating. To monitor reasoning behaviors, we introduced ReasonType: a novel taxonomy of reasoning steps. Building on this framework, we demonstrated that online monitoring of the count of specific steps can produce effective interpretable early stopping criteria of LRM inferences. We evaluate the Step-tagging framework on three open-source reasoning models across standard benchmark datasets: MATH500, GSM8K, AIME and non-mathematical tasks (GPQA and MMLU-Pro). We achieve 20 to 50% token reduction while maintaining comparable accuracy to standard generation, with largest gains observed on more computation-heavy tasks. This work offers a novel way to increase control over the generation of LRMs, and a new tool to study behaviors of LRMs.

## 1 INTRODUCTION

For the past few years, the field of Language Reasoning Models (LRMs) has experienced significant growth in terms of capabilities. Initiated by the pioneering work on model prompting such as Chain-of-Thought (Wei et al., 2023) and Self-Consistency (Wang et al., 2023), Inference Time Scaling has emerged as a popular field with the goal of making models more accurate at reasoning. At the same time, fundamental work on Reinforcement Learning (RL) and Supervised Fine-Tuning (SFT) as part of Training Time Scaling has led to the release of strong reasoning models.

However, recent surveys have shown that LRMs need to generate a very large number of tokens—several thousands—in order to generate an accurate answer on challenging questions (Qu et al., 2025; Chen et al., 2025b; Sui et al., 2025a). This behavior makes reasoning models extremely inefficient - scaling in both compute resources and inference time. Although recent works have suggested solutions to this problem, most approaches overlook the possibility of monitoring the reasoning as it is being generated, and using this information to improve the inference such as its efficiency. Indeed, efficient inference-time scaling methods are either static - fixed budgets/black-box, lacking of interpretability - or dynamic, but they do not exploit the semantic content of reasoning steps to guide early stopping. To address this challenge, this paper aims to offer a new perspective on the efficiency of LRMs by focusing on online monitoring of models. Our contributions are as follows:

- **Formalization of the Reasoning step concept:** From the literature, we have observed numerous definitions of reasoning steps. We first present a comprehensive review and formalize a more generalized definition of what constitutes a reasoning step for LRMs. We then propose *ReasonType*, the first taxonomy of reasoning steps, enabling a structured identification of reasoning behaviors.
- **Step-Tagging module:** We introduce the *Step-Tagging* module (see Figure 1), an online lightweight sentence classifier capable of identifying the nature of each step that the LRMs are generating. This novel framework offers a tool to systematically monitor the generation of LRMs.
- **Interpretable Early-Stopping Framework:** We observed that LRMs often generate the correct answer early in the output sequence. Leveraging the *Step-Tagging* module, we found that the

Figure 1: Step-Tagging: a framework for monitoring the generation of LRMs - example on sample 39 from MATH500 test with DS-Qwen14B, using the ReasonType taxonomy - seed 42

type of reasoning steps plays a role in determining the early-stopping condition. Based on these observations, we built an interpretable early-stopping framework that dynamically stops token generation based on reasoning steps types and counts, calibrated on both models and problem complexity. Tested on three open-source LRMs across five reasoning datasets, our framework reduced token generation by 20-50% while maintaining a comparable accuracy.

The paper is organized as follows. We first review research on LRM efficiency. We then propose a definition of a reasoning step and a taxonomy of reasoning step types. Building on this, we present our Step-Tagging module that can segment and label the reasoning steps within the output of an LRM. We also propose an early stopping mechanism, based on frequency constraints defined on reasoning step types. Finally, we present a set of experiments that validate our framework.

## 2 RELATED WORK

To render models less verbose and more efficient, Train and Test Time Scaling approaches have been explored (Qu et al., 2025; Li et al., 2025; Chen et al., 2025a). Also, recent work has explored monitoring the generation of LRMs. The *Related-Work* section in the Appendix C complements this section, defining the inefficiency problem of reasoning models and its origins.

**Efficient Reasoning through Training.** Using SFT approaches, work such as Xia et al. (2025) explored fine-tuning models on compressed reasoning traces to limit the verbosity of LRM generation. Other papers have suggested various RL algorithms designed to make models more efficient. For instance, Luo et al. (2025a); Team et al. (2025); Yu et al. (2025) showed that including a length component in the reward function leads to more efficient training and inference.

**Efficient Reasoning during Inference.** Researchers have also explored Inference Time Scaling technique to increase the efficiency of models (Qu et al., 2025). *Model Switch* uses a router module to select small or large models for inference depending on the complexity of the problem (Ong et al., 2025). Similarly, *System Switch* looked at dynamically selecting inference settings based on the problem (Aytes et al., 2025). *Length Budgeting* aims to reduce the budget allocated to the generation of answers. Works such as Lee et al. (2025); Han et al. (2025); Xu et al. (2025) showed that careful prompt engineering can lead to more efficient generation compared to standard inference. In addition, Pu et al. (2025) demonstrated that calibration experiments can be performed to estimate the optimal number of tokens to solve particular problems. However, these techniques are hardly interpretable since they rely on either prompt engineering or black box techniques.

**Monitoring LRM generation.** We observe an emerging theme of research on monitoring LRM generation at a step level. Specifically, Lee & Hockenmaier (2025) proposes a taxonomy of reasoning traces evaluators. However, the authors acknowledged that existing monitoring approaches are not adapted to complex reasoning traces. Similarly, Golovneva et al. (2023) prompts a model to generate step-by-step reasoning, and defines an error-type taxonomy to evaluate reasoning steps. Nevertheless, this method is post-hoc and focuses on measuring the quality of reasoning. On the training side, Zeng et al. (2025) showed that monitoring the generation of LRMs can enhance their performance by balancing both Exploration and Exploitation. But their technique does not monitor the reasoning traces at a step-level. Furthermore, Luo et al. (2025b) decomposes long reasoning traces by prompting an LLM to parse reasoning steps and assigns to each step one high-level class - from a taxonomy of four step-types. However, this approach is applied for training purposes and does not monitor reasoning during inference. As a result, existing works often overlook the question of how to dynamically monitor LRMs reasoning during single inferences (Table 1 in Appendix C).

**Monitoring for dynamic efficient inference.** As stated earlier, recent work has explored improving inference efficiency through early-stopping criteria, but existing approaches remain static. These techniques rely on fixed budgets or template constraints defined before inference and therefore do not adapt to the evolving reasoning traces of the model. As a result, they cannot exploit the information contained in the model's own generated reasoning during inference. Closer to our work, Yang et al. (2025) proposes dynamic early stopping criteria based on the model's confidence (DEER). Similarly, Wang et al. (2025) suggests an early-stopping criteria on the entropy of the tokens generated after forcing the generation of end-of-thinking tokens (EAT).

To the best of our knowledge, no prior inference-time efficiency framework uses online monitoring of the *semantic content of the model's generation* to drive stopping decisions (see Table 2 in Appendix C). Our work fills this gap by dynamically tracking the nature of the step generated to implement an interpretable early-stopping controller. To better understand how the generation reflects on the model's reasoning, we must begin by precisely defining what constitutes a *reasoning step*.

## 3 How to define a reasoning step?

The concept of a reasoning step is central in evaluating and improving the generation of LRMs. However, defining a reasoning step remains a non-trivial problem. As highlighted by Yao et al. (2023); Lee & Hockenmaier (2025); Cao et al. (2025), the step segmentation depends on the models, the problem, and different research goals lead to various definitions. In this section, we survey existing approaches and select the one that leads to the most robust definition.

**Token-per-token generation.** From Schuurmans et al. (2024), we can formalize the auto-regressive generation of text of LLMs, and thereby LRMs. We assume $x_{1:s}$ is an $(1, s)$ dimensional vector containing the tokens of the input sequence, where each token $x_i \in V = \{v_1, ..., v_V\}$, $|V|$ being the size of the vocabulary. We can approximate the next-token generation as following:

$$P_{\pi_\theta}(y|x_{1:s}) \approx \prod_{i=1}^{n} P_{\pi_\theta}(y_i|x_{1:s+i-1}) \tag{1}$$

where $P_{\pi_\theta}(y|x_{1:s})$ is the probability of generating the output sequence $y = y_{1:n} = x_{s+1:n+s}$ in an auto-regressive manner, and $\pi_\theta$ is the model parametrized by $\theta$.

### 3.1 What is a reasoning step?

Rather than viewing the model's output as a monolithic text sequence, recent work has shifted toward decomposing generation of LRMs into discrete steps. This decomposition enables finer-grained analysis of model behavior and facilitates targeted interventions. From the literature, we identified four principal methods to segment the output from a model into distinct thoughts (Appendix E):

- **Token or sentence level:** Näively, thoughts can be decomposed into token (Yao et al., 2023) or sentence level (Fu et al., 2023). However, for complex reasoning problems, these definitions are not ideal since reasoning steps are composed of multiple sentences in mathematical reasoning.
- **Paragraph level:** LLMs and LRMs such as Deepseek-R1, QwQ, or GPT are natively generating back-to-line symbols between two thoughts (e.g. .\n\n). Since this observation is model agnostic, it has been adopted by several works (Cao et al., 2025; Park et al., 2024; Lightman et al.,

2023). However, Cao et al. (2025) emphasized that this approach on its own is not enough to correctly distinguish each steps. Models tend to output these symbols frequently, and so using them as delimiters for reasoning can result in over-estimating the real number of steps. Figure 8 in the Appendix E supports this observation.

- **Dynamic steps using special token:** Another common approach is to prompt the model to force the generation of special tokens to split the thoughts (e.g. `<next_step>`). While some works have used this strategy (Zelikman et al., 2024; Sui et al., 2025b; Paul et al., 2024), it suffers from low reliability and efficiency. Indeed, this approach artificially generates more tokens, and prompt engineering could cause mistakes since models are not pre-trained to perform this sub-task.
- **Prompting segmentation:** Another approach is to prompt a model to segment its output into steps. Luo et al. (2025b) prompts a model to parse the raw reasoning output into structured steps. Golovneva et al. (2023) prompts the model to generate step-by-step reasoning. Similarly as the previous method, this approach rely on a model to perform the step segmentation.

To clearly identify and monitor reasoning steps, most approaches are insufficient to split the reasoning traces of models. This motivated us to find an alternative approach to segmenting the traces.

## 3.2 MODEL AGNOSTIC REASONING STEP DEFINITION

Since most models separate paragraphs and thoughts using back-to-line symbols, using this token is a useful starting point to segment reasoning steps. However, to mitigate the over-segmentation problem caused when back-to-line symbols are generated too frequently, Cao et al. (2025) set a minimal number of tokens $k$ per reasoning step and merge reasoning steps shorter than $k$ together. We adopt a similar definition of a reasoning step: a step is delimited by ".\n\n", and we set a minimal number of tokens $k$ on the size of each step to avoid considering many small steps.

**Step Generation.** As introduced in the previous section, we decompose the output sequence of LRMs into discrete reasoning steps. Building upon Cao et al. (2025) and the definition in Equation 1, we first formalize the notion of *stepwise generation*. Let $y = y_{1:n} \in V^n$ be the output token sequence generated by the model over the vocabulary $V$. We define a reasoning delimiter token $\alpha \in V$, such as $\alpha$ = ".\n\n". Let $R = \{r_0 = 1, \ldots, r_i, \ldots, r_{T'} = n\}$ denote the indices in $y$ corresponding to the occurrence of $\alpha$ in $y$. $r_0$ and $r_{T'}$ correspond to the first and last indexes of $y_{1:n}$. Based on these indices, we define a sequence - of length $T'$ - of reasoning steps formed by $y_{1:n}$ with the delimiter $\alpha$:

$$S^* = \{s_1^*, \ldots, s_i^*, \ldots, s_{T'}^*\}, \text{ such as } s_i^* = y_{r_{i-1}:r_i} \qquad (2)$$

where each step $s_i^*$ corresponds to a sub-part of the full output $y$. However, we observe that models tends to generate back-to-line symbols frequently. To reduce redundancy and noise from short or fragmented steps - highlighted by Cao et al. (2025) - we introduce a minimum token threshold $k \in \mathbb{N}$ such as:

$$S = \{s_1, \ldots, s_T\}, \quad \text{with } |s_j| \geq k \text{ for all } j \in [1, T]. \qquad (3)$$

For any original step $s_i^* \in S^*$ such as $|s_i^*| < k$, we continue the generation until the merged span reaches another delimiter and its length exceeds the threshold $k$. In this formulation, each new reasoning step is initiated by the generation of $\alpha$, offering a more consistent definition between models. Algorithm 1 in Appendix F.1 formalizes our definition.

## 4 STEP-TAGGING MODULE

In the previous section, we formalized the method we selected to segment reasoning steps of LRMs. Building on this, we introduce *Step-Tagging*, a lightweight module capable of identifying, discriminating, and tagging reasoning steps in real-time during inference.

**Objective.** Our definition of a reasoning step enables users to segment reasoning steps within model outputs. However, this definition alone does not allow the user to annotate the segmented steps with reasoning types. This annotation would enable users to track logical transitions within model outputs. To do this, we must first define a tag dictionary $\mathcal{T}_{\text{tags}}$ (i.e., a label space of reasoning step tags) that covers the types of reasoning steps generated by models. Essentially, given a sequence of reasoning steps $S = \{s_1, s_2, \ldots, s_T\}$, we wish to label each step $s_i$ with a tag $\tau_i \in \mathcal{T}_{\text{tags}}$. Formally, we are looking to construct a step-tagging function $\phi$ such as:

$$\forall i \in [1, T], \phi(s_i) = \tau_i \qquad (4)$$

Figure 2: ReasonType - A taxonomy of reasoning step types as per `gpt-4o-mini`

where $s_i \in S$ is a reasoning step from the full output sequence $y$, where $|s_i| \geq k$, $\phi$ is the step-tagging function, and $\tau_i \in \mathcal{T}_{tags}$ is the reasoning tag associated to the step $s_i$.

**Taxonomy of the type of steps.** To enable fine-grained monitoring of reasoning behavior, we need to know the different types of reasoning steps that are typically generated by LRMs (i.e., we need to define $\mathcal{T}_{tags}$). To do so, we created a taxonomy based on the outputs of both DeepSeek-R1-Distill-Llama-8B (DeepSeek-AI et al., 2025) and QwQ-32B (Team, 2025) models.

Inspired by prior work on model behavior analysis (Galichin et al., 2025; Kuznetsov et al., 2025), we first created a prompt to identify distinct types of reasoning steps in the traces (see Appendix M.1). We then sampled 40 reasoning traces from the MATH500 train dataset (covering two samples per difficulty level for each model) and using our prompt submitted the traces to GPT-4o-mini (OpenAI et al., 2024). The prompt resulted in a series of different step-types. We merged overlapping categories, to construct a taxonomy that reflects the temporal and reasoning progression of model's traces. We refer to this taxonomy as *ReasonType* (Figure 2) encompassing 13 categories, including early-stage behaviors such as *Problem Re-statement*, later reasoning stages like *Verification* and *Exploration*. To validate our taxonomy, we conducted a series of ablation studies in Appendices H and I. Our results support that our taxonomy enables meaningful and efficient monitoring.

**Early-stopping criteria.** In the following section, we see that LRMs tend to generate the answer early in the output sequence, with step-types following an ordered pattern. Based on this observation, the central challenge that we address is to determine when to stop the generation of LRMs based on step tags, creating an interpretable stopping criterion. Assuming that our Step-Tagging framework can effectively monitor the steps (Equation 4), we can define a constraint on the frequency of a given step type. Each constraint operates online, over a running sequence of reasoning steps $S_{\text{running}} = \{s_1, \ldots, s_j\}$, where each step $s_i$ is associated with a tag $\tau_i \in \mathcal{T}$. We define the constraint $c_{\tau^*}$ as:

$$c_{\tau^*}(S_{\text{running}}, \delta) = \mathbf{1}[f_{\text{freq}}(S_{\text{running}}, \tau^*) \leq \delta] \text{ with } f_{\text{freq}}(S_{\text{running}}, \tau^*) = \sum_{i=1}^{j} \mathbf{1}[\tau_i = \tau^*] \quad (5)$$

where $c_{\tau^*}(S_{\text{running}}, \delta)$ is the constraint on the tag type $\tau^*$ over the step-sequence $S_{\text{running}}$ being generated, given the threshold $\delta$. $f_{\text{freq}}(S_{\text{running}}, \tau^*)$ is the occurrence of the type-step $\tau^*$ over the running sequence $S_{\text{running}}$. While the constraint $c_{\tau^*}$ is satisfied, the generation continues. If the constraint is violated, the generation stops (see Appendix F.2 for more implementation details).

To facilitate the evaluation of early-exit answers, we prompted the models right after the last step being generated, and allowed an additional budget of 100 tokens. We used the following prompt: *"\n\n I am confident in my answer. Here is the final answer.\n\n **Final Answer**"*. We borrowed this approach from Muennighoff et al. (2025), who showed that this intervention helped the model to provide explicitly its current best answer - thereby facilitating evaluation.

## 5 EXPERIMENTAL SETTING

Our paper contains two objectives. First, our goal is to prove that lightweight classifiers can effectively monitor the generation of LRMs. Furthermore, we show that the Step-Tagging framework can be used to implement an interpretable early-stopping criterion to make the generation of LRMs more efficient. We will first motivate our choices of datasets and inference settings followed by the step-tagging pipeline and the choice of metrics to measure the performance of this pipeline.

**Datasets.** To assess our approach, we selected two state-of-the-art reasoning datasets:

- **MATH500** (Hendrycks et al., 2021): This dataset includes $12,500$ mathematical questions spanning 5 different levels of complexity, allowing diversity in analysis of efficiency of reasoning be-

haviors. We selected the curated version from Lightman et al. (2023), containing $500$ test samples and selected $1,000$ training samples to form an equivalent distribution of complexity level.

- **GSM8K** (Cobbe et al., 2021): This dataset contains $8,792$ mathematical questions. We selected $3,000$ train instances, and the original $1,318$ test samples. Overall, this dataset is more homogeneous where questions involve logical mathematical reasoning, and include a larger number of questions - which is good for benchmarking models.

**Model selection.** To apply our framework a user must have access to the fine-grained reasoning traces of LRMs. However, many high-performing closed-source models (such as, o3 and Claude 3.7) do not expose raw reasoning traces. Instead, these models output summaries of thinking tokens generated, which can bias the estimation of their efficiency compared to open-source models. In contrast, open-source models like DeepSeek-R1 and QwQ consistently provide reasoning traces. For this reason, we focus our analysis exclusively on *DeepSeek-R1-Distill-Llama-8B*, *DeepSeek-R1-Distill-Qwen-14B* and *QwQ-32B*, which offer the granularity needed to monitor the reasoning process. This choice is motivated by their variety in term of size and performance, full open-source availability, and diversity in providers.

**Inference setting.** To monitor the steps and intervene in the generation process, we suggest a new definition of the generation process of LRMs. We assume that each model generates one token at a time, and we split the steps dynamically. However, for the purposes of our experiments instead of re-designing the generation process, we performed standard inference and applied our *Step-Tagging* and *Early-Stopping* algorithms *offline*. To ensure the robustness and reproducibility of our approach, we generated five outputs per test sample using fixed random seeds (namely $40$, $41$, $42$, $43$, and $44$), with deterministic decoding. A detailed analysis of the expected *online* runtime of our framework is provided in Appendix J.1. As well, Appendix J.2 shows that the inference savings enabled by our framework quickly offsets the training and calibration cost during deployment.

**Metrics.** To assess the model's performance on challenging reasoning tasks, the Avg@$k$, Pass@$k$, and Cons@$k$ are common metrics (Chen et al., 2021; 2025a; Yu et al., 2025). The Pass@$k$ measures the proportion of the samples where at least one of $k$ attempts leads to the correct answer, while the Cons@$k$ consider a sample correct if all $k$ attempts are correct. Since we are interested about both performance and robustness of our approach, we selected the Avg@$5$, the Pass@$5$ and the Cons@$5$ as the quantitative metrics. Assessing the performance of LRMs on mathematical questions is challenging. This is due to the open nature of the question. For our experiments, we selected the *Math-Verify*[1] library which is a common metric to assess mathematical problems. It uses text extraction and formal verification. This metric also reported strong correctness compared to other evaluation methods such as Harness (Zhibin Gou, 2024) or Qwen-Math Verifier (Huang et al., 2025).

**Baselines.** To assess the effectiveness of our early-stopping approach, we define two baselines:

- **Ideal Early stopping - $\mathcal{IES}$:** We observe a growing understanding that, up to a token-budget, thinking longer may be leading to worse results. Muennighoff et al. (2025) observes that certain models achieved correct answers at the beginning, but sometimes backtracked to a wrong answer. Inspired by this work, we define the *Ideal Early Stopping*, which prunes the remaining steps after the first occurrence of the correct answer based on our metric - if any. In this case, this baseline is theoretical since the ground truth label is needed for each inference (see Appendix N).
- **Prompt-guided efficiency - $\mathcal{P}_{\textbf{guided}}$:** We also observe that LRMs are sensitive to the input prompt (Lee et al., 2025). In this case, we compare our framework with user-prompt and system-prompt variants, with Zero-Shot and Few-Shot prompts that aim to reduce the reasoning computation while retaining accuracy. We explicitly instructed the models to not generate verbose output, or over-verification steps. We selected 4 variants, namely: zero-shot user and system prompt, and few-shot system prompt with 1 and 3 examples: $\mathcal{P}_{\text{user}}^{(0)}$, $\mathcal{P}_{\text{system}}^{(0)}$, $\mathcal{P}_{\text{system}}^{(1)}$, $\mathcal{P}_{\text{system}}^{(3)}$, respectively. The prompts used to establish these baselines are listed in Appendix N.

## 5.1 IMPLEMENTATION OF THE STEP-TAGGER MODULE

**Training data generation.** Given that our reasoning step taxonomy was created using GPT-4o-mini (OpenAI et al., 2024) the most direct way to label a reasoning trace would be to use GPT-4o-mini. However, this GPT-4o-mini annotation is costly, each step requiring more than a second to

---

[1]https://github.com/huggingface/Math-Verify

be annotated (see Table 19 in Appendix M.2). Consequently, instead, we used `GPT-4o-mini` to label a dataset of reasoning traces with the labels from the taxonomy that we use to train lighter weight reasoning step classifiers. We constructed training datasets by running each LRMs on $1,000$ samples from MATH500 train and $3,000$ samples from GSM8K train datasets (with a seed of $42$). For each step $s_i$ in generated outputs, we prompted GPT-4o-mini to assign a tag $\tau_i$ (Appendix M.1). Appendix I.2 confirms the reliability of GPT-4o-mini to annotate the reasoning steps.

**Sentence classifiers.** We selected the `bert-base-uncased` sentence classifier (Devlin et al., 2019) to construct our Step-Tagging framework, including a single hidden layer. Given the large and fine-grained nature of our taxonomy (13 distinct step types), training a multi-class classifier is challenging due to significant class imbalance. To address this, we trained separate binary classifiers for each step-type. This approach notably improved detection accuracy across low-frequency categories, and fits our definition of early-stopping constraint: one step-type per early-stopping criteria. We used a *balanced cross-entropy* to enhance the performance of the models on low-represented classes. We implemented an early-stopping criteria, and a maximum of 5 epochs. The batch size is 16 and we used an AdamW optimizer with a learning rate of $2.10^{-5}$. To evaluate the performance of our classifiers, we computed the Macro-F1 and Micro-F1 on the test datasets. While the *Macro-F1* helps to identify the classifier's ability to detect rare classes, the *Micro-F1* offers a more global view on the step detector's performance across all steps.

## 5.2 STEP-SPLIT SETTINGS

**Minimal step size $k$.** To apply our definition of reasoning steps, users first need to set the value of $k$. From the literature, this task is not straightforward since Cao et al. (2025) uses different values of $k$ based on the problems and models (e.g. $k \approx 100s$). The value of $k$ directly affects the granularity of the monitoring. A *small value* would imply very small steps, sometimes splitting the same thoughts between many steps, leading to extremely frequent monitoring. Conversely, a *large value* of $k$ would imply significantly large steps, including multiple thoughts and hence leading to biased monitoring.

**Selecting the optimal $k$ value.** To balance fine-grained monitoring and reasoning efficiency, we rely on two proxies to determine the value of $k$. First, the $\mathcal{IES}$ accuracy provide us a way to assess if individual steps contains more than one thought. Second, the Step-Tagger performance informs us about the semantic meaning of steps under a given a value of $k$. Based on three ablation studies that we conducted in Appendix G, we set $k$ to 60, 30 and 100 for DS-Llama8B, DS-Qwen14B and QwQ-32B, respectively.

## 5.3 EARLY-STOPPING CONSTRAINTS

**Early-Stopping calibration using a Pareto curve.** To select the correct constraints (tag-type $\tau$ and threshold $\delta$) we rely on the training datasets, and on the synthetic generated tags. Figure 3 presents the number of tokens vs. accuracy of every tag-type with values of threshold ranging from 0 to 20, for the DS-Llama8B model on our train MATH500 per complexity level. We first observe that LRMs tend to generate an increasing number of tokens when the complexity of the prompt increases. Figure 3 also shows that early-stopping constraints are dependent on the complexity. For this reason, we selected one constraint per complexity level for the MATH500 dataset, while for the GSM8K dataset, we chose a unique constraint since we assume problems to carry equivalent complexity. Furthermore, we observe that constraints form a *Pareto curve* (Lee et al., 2025), and each step-types results in different trade-off between accuracy and token-count (see Appendix H.3). On the strength of this observation, we set three Early-Stopping criteria that target specific trade-offs between accuracy and efficiency: *ST-ES 95%*, *ST-ES 90%* and *ST-ES 85%*. For each setting, we select the tag-type and threshold that lies closest to the Pareto frontier. Selected constraints and calibrations for the other models and datasets are shown in the Appendix O.

**LRM Router: dynamic inference.** The MATH500 dataset involves various complexity levels. For efficient inference, dynamic routing is needed. However, in real-world settings, the complexity level of a question is sometimes unknown. To address this challenge, we grouped levels $\{1, 2\}$ and $\{3, 4, 5\}$ into two complexity levels, and identified common constraints for both clusters. To route the inference settings, we trained a BERT classifier on the input problem, using the full MATH500 train dataset (see Appendix P). For the MATH500 dataset, we included a fourth Early-Stopping criteria, namely *ST-ES Router*. To study the robustness of our Router configuration, we analyze how router error impacts the efficiency of our framework in Appendix J.3.

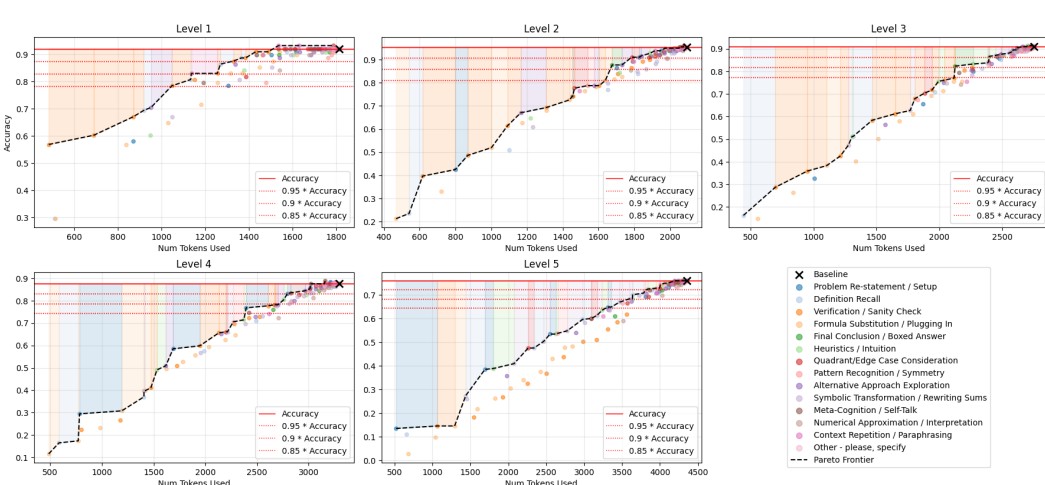

Figure 3: Early-Stopping selection using Pareto - DS-Llama8B on MATH500 train - seed 42

# 6 MONITORING LRMS USING STEP-TAGGER

To validate our taxonomy, we analyse the distribution and sequence of the step-tag labels generated by `GPT-4o-mini`, then we evaluate the performance of our sentence classifiers.

**Reasoning patterns.** First, we observe that our Step-Tagging framework allows us to clearly follow the reasoning progression of the model. Figures 49 and 50 in the Appendix M.2 present an analysis (and validation) of reasoning patterns exhibited by models based on the step-types identified in the reasoning traces generated by the models.

**Step frequency.** Figure 4 presents the frequency of each step-type in the `GPT-4o-mini` labels. The plot shows a high frequency of *Verification*, confirming our observations from the literature. We also note that the frequency seems to depend on the problem complexity and models. For DS-Llama8B and DS-Qwen14B, *Formula Substitution* steps are very occurrent for GSM8K ($\approx$60%), while *Exploration* and *Self-Talk* steps are more frequent for QwQ-32B on both datasets.

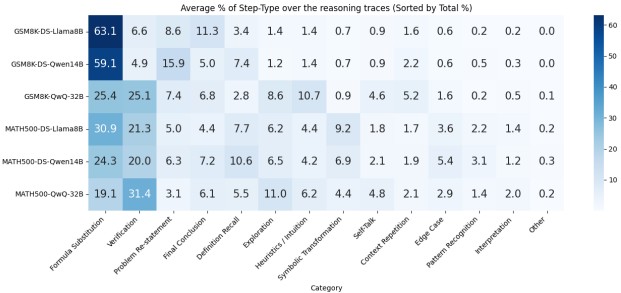

Figure 4: Step-type distribution from `GPT-4o-mini`

**Performance of step monitoring.** Figure 5 presents the performance of the binary step-classifiers on the selected step-types constraints for the *DS-Llama8B model*. We observe that the Micro-F1 is generally high across most steps for all models across all datasets - ranging from $0.89$ to $0.97$, which demonstrates that the classifiers are good at detecting step-tags. Moreover, we also reported the macro-F1 score since the distribution of step-types is highly imbalanced (see Figure 4).

We observe lower scores, notably for *Context Repetition* with $0.65$ (*Context Repetition* is a rare step type, representing $1.7\%$ of the labels, and so we attribute this relatively low score to label imbalance). However, the scores remain relatively high, particularly for *Verification* and *Exploration*.

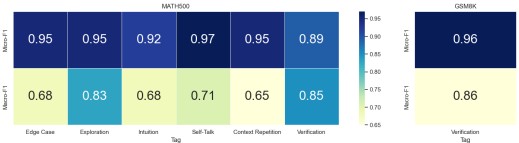

Figure 5: Step-Tagger performance - DS-Llama8B

We interpret the strong performance of the classifiers as validating our reasoning step taxonomy in the sense that it indicates that the step types are distinct (i.e., they reflect types with separable properties). Figures 57 and 58 in the Appendix Q present similar results for *DS-Qwen14B* and *QwQ-32B*, respectively, and Appendix I.3 validates our approach on two additional LRMs.

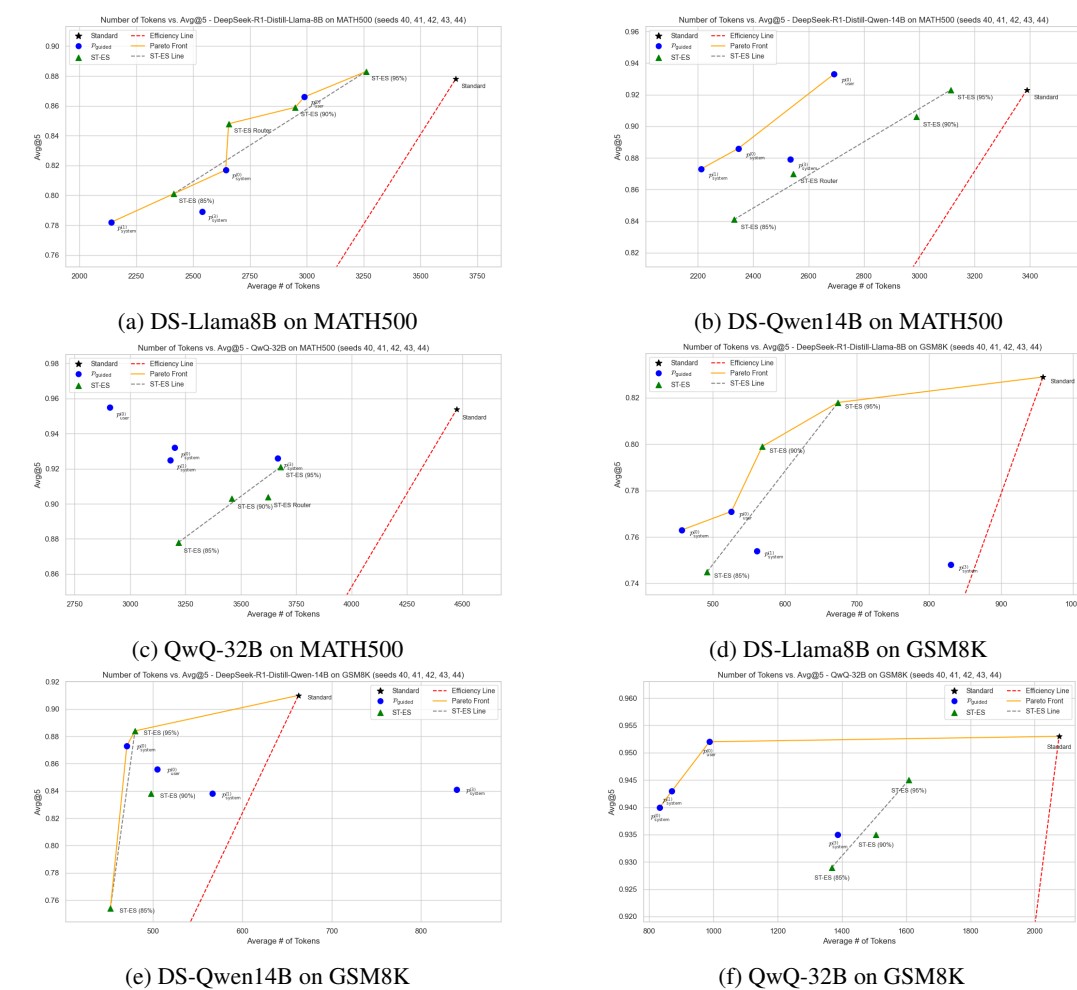

(a) DS-Llama8B on MATH500

(b) DS-Qwen14B on MATH500

(c) QwQ-32B on MATH500

(d) DS-Llama8B on GSM8K

(e) DS-Qwen14B on GSM8K

(f) QwQ-32B on GSM8K

Figure 6: Number of Tokens vs. Avg@5 - $\mathcal{P}_{\text{guided}}$ Baselines vs. ST-ES criteria - The efficiency lines in red highlight the configurations that improve the efficiency relative to the standard inference, while the Pareto frontiers in yellow show the most efficient approaches. The Step-Tagging Early-Stopping framework achieved up to 30 to 40% of token-count saving, with minimal accuracy loss.

# 7 STEP-TAGGING EARLY-STOPPING (ST-ES) CRITERIA

Next, we show in this section that Step-Tagging modules can effectively be used as an early-stopping criteria. Figure 6 presents the average token count against the Avg@5 for the three LRMs on the MATH500 and GSM8K datasets. Each plot compares the performance trade-offs between the baselines and the ST-ES criteria. Table 22 in the Appendix S reports the quantitative metrics of the baselines and our approach on the three models, for the 5 seeds that we selected.

$\mathcal{P}_{\text{guided}}$ **baselines.** We first notice that simple instruction on the models results in strong token-reduction, achieving 20% to 60% saved tokens across configurations. Specifically, it seems that the baselines are giving much better results on QwQ-32B, and the system-prompt variants generally lead to more token-reduction for the Deepseek models.

**Strong performance of the ST-ES.** Next, we observe that our ST-ES criteria effectively leads to more efficient generation, with all ES-ST settings lying on the left side of the Efficiency line compared to the Standard inference for all models. Furthemore, the ST-ES criteria appear to outperform most $\mathcal{P}_{\text{guided}}$ baselines for both Deepseek models.

Indeed, we observe that our ST-ES criteria is performing well on the DS-Llama8B model on both datasets since almost all ST-ES configurations lies on the Pareto front. On MATH500 (Figure 6(a)), ST-ES Router and ST-ES 85% achieved approximately the same token reduction as $\mathcal{P}_{\text{system}}^{(0)}$ and $\mathcal{P}_{\text{system}}^{(3)}$ (27% and 34%, respectively), while achieving higher accuracy. On GSM8K (Figure 6(d)),

ST-ES 90% achieves the same token reduction as $\mathcal{P}_{\text{system}}^{(1)}$ (around 41%) while maintaining higher Avg@5 (0.799 vs. 0.754, respectively). Furthermore, results on the DS-Qwen14B model also show good performance of criteria leading to significant token-reduction (10 to 32%), with some configurations lying on the Pareto front. However, the $\mathcal{P}_{\text{guided}}$ settings appears to lead to more efficient inference, notably for MATH500. In addition, the criteria suffers from more accuracy loss, as the ST-ES Line are more vertical than for DS-Llama8B.

**Generalization to other tasks.** To assess robustness beyond MATH500 and GSM8K dataset, we evaluate our framework to AIME - a harder mathematical dataset (Appendix K.1) - as well as GPQA-Diamond and MMLU-Pro - non-mathematical dataset (Appendix K.2). Our results indicate that ST-ES scales well on more complex tasks. Moreover, it shows that reasoning-step monitoring is not tied to mathematical structure and that the ST-ES remains effective across diverse reasoning tasks.

**ST-ES faces challenges on the QwQ model.** In contrast, the ST-ES criteria shows nuanced results on the QwQ-32B model. Baselines are stronger, and for the same token gains as the Deepseek models, the accuracy loss seems higher. We suspect that this observation can be due to two factors. First, Figure 7 presents the average per percentage of the full output sequence for the three models.

DS-Llama8B and DS-Qwen14B appear to generate correct answers earlier in their output sequences, but sometimes continue reasoning, leading to the destruction of the correct current answer. In particular, this is the case on easier problems (Level 1-3 MATH500) and GSM8K, where a drop in the accuracy can be observed at around 40-50% stopping. By contrast, QwQ-32B exhibits more stable accuracy gains as the token count increases, which suggests that the model is more conservative in its way of expressing its current solution. This behavior is further analyzed in Appendix L, further confirming our claim.

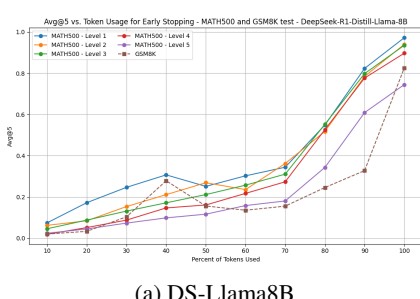

(a) DS-Llama8B

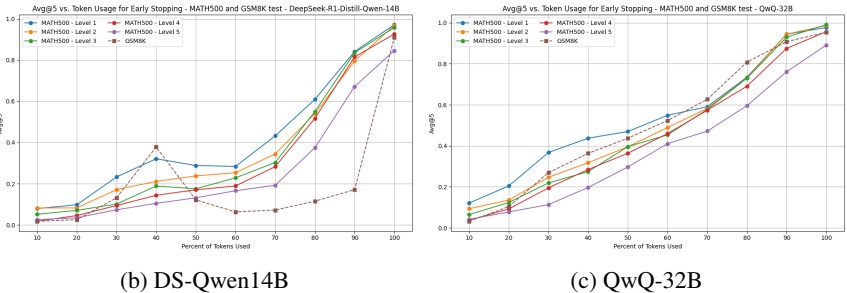

(b) DS-Qwen14B         (c) QwQ-32B

Figure 7: Early-Stopping Avg@5 per percentage of the full output sequence

Second, we suspect that larger models are better at controlling the length of their generation using specific prompts. We observe that the baselines from QwQ-32B are much more efficient than the ones from both DS-Llama8B and DS-Qwen14B, which are smaller models. We note that Lee et al. (2025) report similar findings with larger models achieving higher *Upper Bound of Token Reduction* when prompted to compress their reasoning.

# 8 CONCLUSION

This work offers a novel view on both monitoring and efficiency of LRMs. We propose *ReasonType*, a novel taxonomy of reasoning steps, and demonstrated that users can effectively track the reasoning flow of the generation using our *Step-Tagging* framework. Our taxonomy proved robustness across models, and satifying performance of our classifier suggests that LRMs exhibits structured and separable reasoning behavior, paving the way for more work on the monitoring of reasoning steps.

Furthermore, we show that differentiating the step-type in the generation of LRMs is important, and can be used as a reliable and interpretable early-stopping criterion. Through careful monitoring of specific step-types, our framework can enhance the control of the generation of RLMs enabling a significant token-count saving (up to 50%) while preserving performance, with the largest gains on complex tasks.

## REPRODUCIBILITY STATEMENT

We took several measures to ensure the reproducibility of our experiments, namely:

- **Code availability:** The source code that we developed to conduct our experiments is available in the submission ZIP folder.
- **Experimental Settings:** We listed in Section 5 the experimental settings. This includes the datasets used, the models (open-source available on HuggingFace), the parameters of the algorithms, the prompts of the models, the evaluation functions, and the environment setups (seeds and deterministic decoding). We also included scripts to reproduce the experiments we lead. We used one or two A100-80GB GPUs to run our experiments.

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

# Appendix

## Table of Contents

## A    LLM USAGE

We acknowledge the use of Large Language Models for the purpose of our experimentation in our paper. Specifically, as stated in Section 4, we relied on `GPT-4o-mini` to set our *ReasonType* taxonomy. This approach is borrowed from work on behavior analysis of LLMs, such as Galichin et al. (2025); Kuznetsov et al. (2025).

## B    LIMITATIONS AND FUTURE WORK

Our definition of reasoning step is taken from previous work, and relies on empirical evaluation. We believe that our step taxonomy can enhance the step definition. Future work should look at leveraging the performance of step-classification to better define reasoning steps.

To train accurate Step-Tagger modules, we suspect that significantly increasing the number of traces could lead to better results. Also, down-sampling could render our training more effective, and increase the Macro-F1. In addition, a better definition of a step could lead to more effective monitoring. For instance, it would be interesting to explore dynamic values of minimal number of tokens $k$, making our approach even more agnostic.

Further, our ST-ES criteria is näive, since it is simply based on frequency monitoring. Some works have started to explore confidence-based methods on the uncertainty of the logits. We believe that integrating this knowledge to our criteria could further enhance our early-stopping criteria.

## C    RELATED-WORK

### C.1    LANGUAGE REASONING MODELS (LRMS)

The field of reasoning models has been very active over the recent years. The literature tends to divide existing models into two distinct categories, namely *System-1* and *System-2* (Li et al., 2025; Qu et al., 2025). System-1 models refer to intuitive and fast LLMs. These correspond to standard instructed LLMs. In contrast, System-2 models are slower and deeper thinking, designed to perform explicit multi-step thought, and are referred to as LRMs.

**Building System-2 models.** Driven by substantial research efforts, LLMs now excel at standard capabilities such as Natural Language Processing (NLP) (Touvron et al., 2023; OpenAI et al., 2024), code generation (Mishra et al., 2024; Rozière et al., 2024) and Function-calling (Abdelaziz et al., 2024; Patil et al., 2023). However, traditional LLMs perform poorly on reasoning task as shown by work such as Williams & Huckle (2024); Seßler et al. (2024); Hosseini et al. (2024). Research on enhancing LLM reasoning capabilities can be decomposed into two categories: Training, and Inference Time Scaling (Raschka, 2025).

**Training Time Scaling.** To enhance the performance of LLMs on reasoning tasks, a substantial body of work has explored different training techniques. A promising path has been to fine-tune LLMs on reasoning traces using RL and SFT techniques. For example, Deepseek (Shao et al., 2024) introduced the Group Relative Policy Optimization (GRPO). Unlike the classic Proximal Policy Optimization (PPO) (Schulman et al., 2017), GRPO estimates the reward using group scores, which naturally enhances model's capability to generate reasoning traces. Nevertheless, this process renders the models to be much more verbose.

**Inference Time Scaling.** By contrast, Inference Time Scaling aims to enhance model performance on reasoning tasks after training. To do so, researchers looked for reasoning instances at sampling (Wang et al., 2023; Snell et al., 2024), or prompt engineering techniques (Wei et al., 2023; Muennighoff et al., 2025). Since the generation of LLMs is often highly variable, multiple inferences could lead to a wide diversity of answers. These techniques help the reliability and correctness of models. However, they comes at the cost of both compute resources and runtime.

### C.2    LRMS ARE INEFFICIENT

From the literature, we can observe a strong push for more efficient LRMs. This section will aim to define the efficiency problem of LRMs, and the underlying gaps in the literature.

**Patterns of inefficiency.** It is only recently that a few papers highlighted that LRMs tend to generate thousands of tokens to solve reasoning problems. Chen et al. (2025b) was the first to formalize this issue, known as *model overthinking*, where LRMs tend to generate a disproportionate number of tokens for fairly simple problems. In this emerging field, work such as Qu et al. (2025) began to look for evidence of inefficiency and showcased examples where reasoning models exhibit patterns of inefficiency mainly due to redundancy such as verbose problem reformulation, or over-verification. Munkhbat et al. (2025) also support this claim and gives a general overview of findings in the field. Su et al. (2025) claims that majority of tokens generated by LRMs ensure textual coherence rather than core reasoning. Experimentation presented by Luo et al. (2025a) supports the argument that longer answers from models does not necessarily lead to more accurate answers, and in some cases can even lead to worse answers. However, to the best of our knowledge, the literature lacks tools to systematically identify patterns of inefficiency such as redundant generation.

**Origins of inefficiency.** First, Sui et al. (2025a) observed from the Deepseek paper DeepSeek-AI et al. (2025) that GRPO training leads to a positive correlation between the accuracy of the model and the average number of tokens generated. Furthermore, Galichin et al. (2025) highlighted that LRMs such as Deepseek-R1 tend to generate tokens that are responsible for the generation of certain types of reasoning steps. Referred to as *reasoning tokens* (e.g. `Wait`, `Hum`, `Let me verify`, ...), the authors demonstrated that these tokens are responsible for guiding the generation of intermediate stages of reasoning, such as pausing the thought, re-evaluating the current answer, or exploring novel solutions. However, their empirical evaluation on activation steering showed that the over-reliance on such tokens tends to increase the verbosity of model output. Muennighoff et al. (2025) also supports this claim. Indeed, they demonstrated that prompting reasoning tokens during the generation forces the model to reason more.

## C.3 COMPARISON TO PRIOR WORK ON MONITORING LRMS

| Reference | Stage | CoT steps | Taxonomy | Description |
|---|---|---|---|---|
| (Zeng et al., 2025) | Training | | | Self-improvement by balancing Exploration and Exploitation; does not segment CoT into steps |
| (Luo et al., 2025b) | Training | ✓ | ✓ | Segments long CoT using LLM prompting; Taxonomy composed of 4 high-level classes to structure CoT steps; used for distillation |
| (Lee & Hockenmaier, 2025) | Inference (post-hoc) | ✓ | ✓ | Assumes step segmentation; Taxonomy of evaluation criteria; CoT evaluation |
| (Golovneva et al., 2023) | Inference (post-hoc) | ✓ | ✓ | Assumes step segmentation; Taxonomy of error types; CoT evaluation |
| (Ours) | Inference (online) | ✓ | ✓ | Heuristic segmentation; Taxonomy composed of 13 fine-grained step types; Efficient inference Time-scaling and interpretability of CoT steps |

Table 1: Comparison of prior work on LRM monitoring

## C.4 COMPARISON TO PRIOR EARLY-STOPPING METHODS

| Reference | Static Methods | | |
|---|---|---|---|
| | **Prompt Compression** | **Token-count** | **Black-box estimation** |
| (Xu et al., 2025) | ✓ | | |
| (Lee et al., 2025) | ✓ | ✓ | |
| (Han et al., 2025) | | ✓ | ✓ |
| (Pu et al., 2025) | | | ✓ |
| | **Dynamic Methods** | | |
| | **Confidence** | **Entropy** | **Semantic Step-Tracking** |
| (Yang et al., 2025) | ✓ | | |
| (Xu et al., 2025) | | ✓ | |
| (Ours) | | | ✓ |

Table 2: Comparison of prior early-exit methods for LRMs. The table is split into two blocks: *Static methods* does not look at the model's intermediate outputs, while *Dynamic methods* use signals obtained by monitoring the model's intermediate outputs during inference.

## C.5 OPEN-SOURCE REASONING PATH

Table 3 shows the difference between close-source and open-source models. Close-source models tends to hide the raw reasoning traces generated by models.

| LRMs | Open Weights | Reasoning Traces |
|---|---|---|
| **DeepSeek-R1** | Yes | Yes |
| **QwQ** | Yes | Yes |
| **o3 / o4** | No | Partial |
| **Claude 3.7** | No | Partial |
| **Gemini 2.5 Pro** | No | Partial |

Table 3: Comparison of LRMs and reasoning traces - *Partial* stands for models that do give access to full reasoning traces

# D MATHEMATICAL REASONING DATASETS

Table 4 presents the selected reasoning datasets, including their references, and the number of samples per training and testing folds. Due to computational resources constraints, we limit the size of the training datasets to smaller subsets of their full versions - specifically, 1,000 samples for MATH500 Lightman et al. (2023) and 3,000 samples for GSM8K Cobbe et al. (2021). These values were selected to have approximately twice the size of the test datasets of training samples. We used a seed of 42 to infer training datasets.

| Dataset | # Train | # Train used | # Test |
|---|---|---|---|
| MATH500 | 12,000 | 1,000 | 500 |
| GSM8K | 7,474 | 3,000 | 1,318 |

Table 4: Description of selected mathematical dataset

# E    DEFINITION OF REASONING STEPS

Table 5 gives an overview of the different definitions of reasoning steps in the literature. Notably, we can define 4 categories, namely: token, sentence, paragraph, and dynamic step delimitation levels.

| Step Type | Ref. | Method | M.A. |
|---|---|---|---|
| Token | Light et al. (2025) | – | x |
| | Yao et al. (2023) | – | x |
| Sentence | Fu et al. (2023) | [. , ;] | x |
| | Yao et al. (2023) | – | x |
| Paragraph | Cao et al. (2025) | . \n\n | x |
| | Lightman et al. (2023) | \n | |
| | Fu et al. (2023) | \n | x |
| | Park et al. (2024) | \n\n\n, ###, *** | x |
| | Yao et al. (2023) | i | |
| Dynamic | Light et al. (2025) | Sampling | x |
| | Zelikman et al. (2024) | < \|startofthought\| > | |
| | Sui et al. (2025b) | <next_step> | |
| | Fu et al. (2023) | "step i" | |
| | Paul et al. (2024) | <equation> | |
| | Welleck et al. (2022) | special tokens | |

Table 5: Overview of types of step-delimiters - M.A. for Model Agnostic

Figure 8 presents an example of reasoning step from the Deepseek-R1-Distill-Llama8B model. We observe that the delimiter . \n\n is very often being generated, specifically when the model executes a series of operations.

---

**Example of Reasoning step**

**[...]** **\n\n** Wait, let me double-check my calculations to make sure I didn't make any mistakes.**\n\n** First, f(-2): **\n\n** 3*(-2) is -6, minus 2 is -8. Denominator: -2 -2 is -4. -8 divided by -4 is 2. Correct.**\n\n** f(-1): **\n\n** 3*(-1) is -3, minus 2 is -5. Denominator: -1 -2 is -3. -5 divided by -3 is 53. Correct.**\n\n** f(0): **\n\n** 3*0 is 0, minus 2 is -2. Denominator: 0 -2 is -2. -2 divided by -2 is 1. Correct.**\n\n** Adding them up: 2 + 53 + 1.**\n\n** Converting to thirds: 2 is 63, 1 is 33, so 63 + 53 + 33 = 143. Yep, that seems right.**\n\n** So, I think I did everything correctly. I don't see any mistakes in my calculations or reasoning.

---

Figure 8: Extract of the reasoning traces of DS-Llama8B - from inference of question 2 of MATH500 test dataset. The extract of the reasoning trace showcases a verification step of the model. Back to line symbols are highlighted in bold.

# F  ALGORITHMS

## F.1  STEP-WISE GENERATION

To generate the reasoning traces of models step-by-step, we need to modify the `model.generate` function from Hugging Face. However, this process comes at the cost of latency in model generation since we need to interrupt the generation process at each step. The algorithm is presented in Algorithm 1.

---

**Algorithm 1** Step-wise Generation

---

**Require:** Prompt $x$; reasoning delimiter $\alpha \in V$; minimal step size $k \in \mathbb{N}$; max steps $T_{\max}$; language model $\mathcal{M}$; tokenizer $\mathcal{T}$; EOS token

1: $y \leftarrow \mathcal{T}(x)$                       ▷ Tokenized input
2: $S \leftarrow []; \beta \leftarrow \emptyset$                  ▷ Initialize output and buffer
3: $s \leftarrow 0$
4: **while** $s < T_{\max}$ **do**
5:  $t \leftarrow \mathcal{M}(y)$                   ▷ Generate next token
6:  $y \leftarrow y + t$
7:  $\beta \leftarrow \beta + \mathcal{T}^{-1}(t)$             ▷ Add decoded token to buffer
8:  **if** EOS in $y$ **then**           ▷ Stop inference if EOS generated
9:   Append $\beta$ to $S$
10:   **break**
11:  **end if**
12:  **if** $\beta$ ends with $\alpha$ **then**
13:   **if** $\beta > k$ **then**             ▷ Complete and valid step
14:    Append $\beta$ to $S$
15:    $\beta \leftarrow''$                 ▷ Empty the buffer
16:    $s \leftarrow s + 1$              ▷ Increase $S$ by one step
17:   **else**
18:    Continue        ▷ Continue until next $\alpha$ or EOS is generated
19:   **end if**
20:  **else**
21:   Continue
22:  **end if**
23: **end while**
24: **return** $S$

---

## F.2   EARLY STOPPING ALGORITHM

Algorithm 2 lists the Step-Tagging Early-Stopping criteria. The user needs to define a constraint $\{\tau*, \delta\}$, and input a Binary Step-Tagger $\phi_{\tau^*}$, which returns 1 if the step tag is $\tau^*$ and 0 otherwise. If the constraint breaks, the algorithm stops the generation, and prompts the model with $\mathcal{P}_{\text{exit}}$ to give the current best answer.

---

**Algorithm 2** Step-Tagger Early-Stopping

---

**Require:** Prompt $x$; reasoning delimiter $\alpha \in V$; minimal step size $k \in \mathbb{N}$; max steps $T_{\max}$; Reasoning Language Model $\mathcal{M}$; tokenizer $\mathcal{T}$; EOS token $\gamma$; Constraint $\{\tau*, \delta\}$; Binary Step-Tagger $\phi_{\tau^*}$; Early-Exit Prompt $\mathcal{P}_{\text{exit}}$

1: $y \leftarrow \mathcal{T}(x)$                                                                  ▷ Tokenize the input
2: $S_{running} \leftarrow [\,]$;                                                              ▷ Initialize output
3: $t \leftarrow 0$
4: $f_{\tau*} \leftarrow 0$                                                        ▷ Initialize frequency track of $\tau^*$
5: **while** $c_{\tau*}(S_{running}, \delta)$ **do**                              ▷ Generate until constraint breaks
6:     Generate step $s_i$ using $\mathcal{M}, \alpha$, where $|s_i| > k$
7:     $y \leftarrow s_i$
8:     **if** $\phi_{\tau*}(s_i)$ **then** $f_{\tau*} \leftarrow f_{\tau*} + 1$                      ▷ Increase the counter
9:     **else**
10:        Continue the generation
11:    **end if**
12:    $t \leftarrow t + 1$
13: **end while**
14: $y \leftarrow \mathcal{M}(y + \mathcal{P}_{\text{exit}})$                          ▷ Infer $\mathcal{M}$ with the early exit prompt
15: **return** y

---

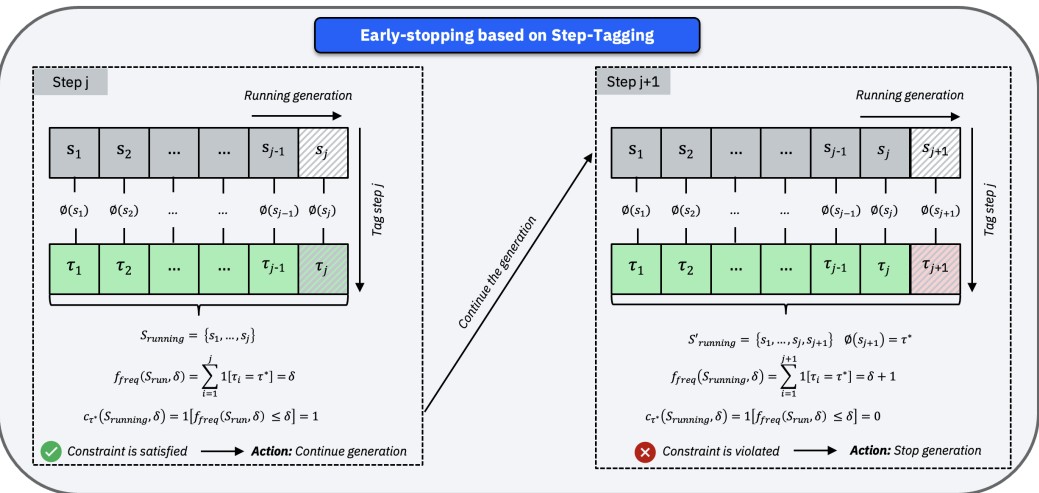

Figure 9: Illustration of early-stopping based on Step-Tagging

# G SELECTING THE MINIMAL NUMBER OF TOKEN $k$

## G.1 IDEAL EARLY-STOPPING AS A PROXY FOR THE STEP-SIZE

**Grounding our approach in the literature.** One crucial component of our work is the formalization of the reasoning step. From the literature, we selected the definition that seems to be the most agnostic to the model, and use-cases Cao et al. (2025) (see Section 3.2). Prior research in social sciences, which we believe can be compared to mathematical reasoning, further support that a reasoning step should be self-contained: *"each reasoning step $e_i$ represents a single piece of evidence contributing toward the social inference to select an answer $A_a$ from $A$"* (Mathur et al., 2025)[p.3].

**Objective and Motivations.** To support this claim, we are looking to produce a segmentation such that each unit clearly reflect a contribution toward the final answer. In addition, we note that this claim supports our problem setting: monitoring becomes more informative when the segmentation of the reasoning is well conducted. In our step definition, the step segmentation is controlled by a minimal number of token per step $k$. To apply this claim, we should first find a way to see if steps contain one or multiple thoughts given a value of $k$. This selection of $k$ is important since we base the rest of our analysis and work on this step definition.

**Methodology.** To select optimal values of $k$ for the three models studied, we rely on the *Ideal Early-Stopping* ($\mathcal{IES}$) baseline as a signal for the quality of the step segmentation (see Section 5). First, we consider the accuracy of $\mathcal{IES}$, which reflects the point at which a correct answer first appears in the reasoning trace. When $k$ is set too large, reasoning steps are likely to contain multiple distinct thoughts. In such cases, correct intermediate answers may be overwritten by later steps, which can potentially reduce $\mathcal{IES}$ accuracy, and thereby contradicting our claim (more than one thought per step). Second, we analyze the mean number of tokens per sample given by the $\mathcal{IES}$ baseline. If $k$ is too large, we expect the average step length to increase, potentially erasing all efficiency gained that the $\mathcal{IES}$ baseline is designed to provide. We applied our methodology on reasoning traces obtained on the MATH500 train dataset, for its diversity in problem complexity.

**Evaluation.** Figure 10 showcases the accuracy of the $\mathcal{IES}$ baseline (red), and its average number of tokens per sample (blue), for the three LRMs on the MATH500 train dataset, using values of $k$ ranging from 1 to 1,000. As expected, we observe that the accuracy generally drops when $k$ increases. In the meantime, the minimal number of tokens of the ideal early-stopping criteria increases when $k$ increases.

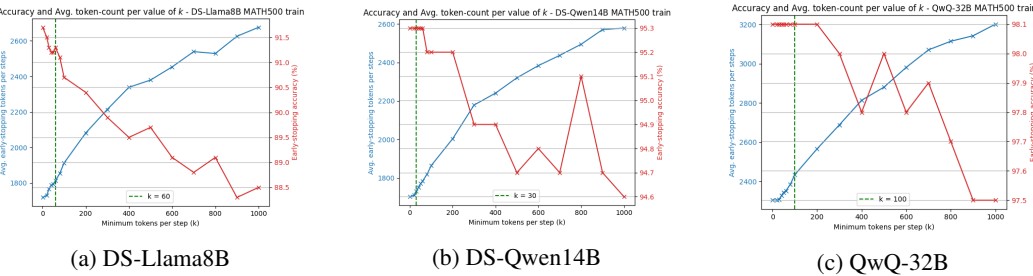

|  |  |  |
|---|---|---|
| (a) DS-Llama8B | (b) DS-Qwen14B | (c) QwQ-32B |

Figure 10: Selecting optimal $k$ - Efficiency of $\mathcal{IES}$

We interpret the accuracy drops when $k$ increases as a signal that some steps are including multiple thoughts. Indeed, our metric is by definition taking the latest solution contained in each step (if any, e.g. located at the last sentence of the step). If the steps are really small, the steps are containing a minimal number of thought (as per the model's step segmentation when generating `.\n\n`). This explains why $k = 1$ obtained the highest accuracy for every models. However, $k$ needs to be set higher than 1 to avoid over-segmentation (see Appendix E). Now, if a step contained a correct answer with a lower value of $k$, increasing $k$ results in adding additional reasoning text - potentially incomplete or erroneous. Then our metric would assign the outcome to the latest solution within the same step, potentially overriding the correct sub-conclusion. Figures 11 and 12 illustrate this.

**Takeaway.** On the strength of these observations, we identified sweet-spots for each models, aiming to balance monitoring efficiency and supports our claims on our step-definitions. We set $k = 60$, $k = 30$, and $k = 100$ for DS-Llama8B, DS-Qwen14B, and QwQ-32B, respectively.

**Example.** Figures 11 and 12 show an example of $\mathcal{IES}$ traces for the sample 658 from MATH500 train using the DS-Qwen14B with a reasoning step-split parameter $k$ of 30 and 300, respectively. We selected this example because the model initially drafted a correct answer, but overwrote it with a wrong reasoning. While the $\mathcal{IES}$ algorithm returned the correct answer parsed at step 8 for the parameter $k = 30$, the same algorithm applied for a parameter $k = 300$ return an incorrect answer, and failed to early-stop. Indeed, with $k = 300$, the same step correspond now to the step 5, and include an additional self-corrected sentence leading to a different answer. This observation confirms that the model sometimes over-write/overthink its correct initial reasoning, and suggests that lower values of $k$ tends to lead to better segmentation of the reasoning traces. Overall, this observation comforts our observation from the previous section and our selection of $k$.

**Step 1 -** Okay, so I've got this geometry problem here. It says that line AB is parallel to line CD, and the measure of angle AXF is 118 degrees. I need to find the measure of angle FYD. Hmm, let me try to visualize this. There's an Asymptote diagram provided, but since I can't actually see the image, I'll have to go off the description \n\n
**Step 2 -** First, I know that when two lines are parallel, certain properties hold, like corresponding angles being equal, alternate interior angles being equal, and so on. Maybe I can use those properties here \n\n
**Step 3 -** Looking at the Asymptote code, it defines points A, B, C, D, E, F, X, and Y. A, B, C, D are the main points, with AB and CD being the parallel lines. E is another point below AB, and F is above CD. Lines are drawn from E to F, which intersects AB at point X and CD at point Y. So, essentially, line EF is a transversal cutting through the two parallel lines AB and CD \n\n
**Step 4 -** Given that, angle AXF is 118 degrees. Let me try to identify where that angle is located. Point A is on AB, point X is the intersection of EF and AB, and point F is on EF. So angle AXF is at point X, formed by lines AX and FX. Since AB is parallel to CD, and EF is a transversal, there might be some corresponding angles or alternate interior angles that I can use \n\n
**Step 5 -** I need to find angle FYD. Point F is on EF, point Y is the intersection of EF and CD, and point D is on CD. So angle FYD is at point Y, formed by lines FY and DY. That angle is also on the transversal EF, but on the other side of the parallel lines \n\n
**Step 6 -** Since AB is parallel to CD, and EF is a transversal, the corresponding angles at X and Y should be equal. So, if angle AXF is 118 degrees, then the corresponding angle at Y should also be 118 degrees. But wait, angle AXF is on the same side as angle FYD? Or is it on the opposite side? \n\n Wait, angle AXF is above AB, and angle FYD is above CD. Since AB and CD are parallel, the angles formed by the transversal EF should be corresponding. So, angle AXF and angle FYD are corresponding angles. Therefore, they should be equal. So, angle FYD should also be 118 degrees \n\n
**Step 7 -** But hold on, let me double-check. When you have two parallel lines and a transversal, corresponding angles are equal. So, angle AXF is at the intersection above AB, and the corresponding angle above CD would be angle FYD. So yes, they should be equal \n\n
**Step 8 -** Alternatively, I can think about the consecutive interior angles. If I consider angle AXF and the angle adjacent to it on the same side, they should add up to 180 degrees because they are consecutive interior angles. So, if angle AXF is 118 degrees, the adjacent angle on the same side would be 180 - 118 = *62 degrees* \n\n
[...]
**Step 37 -** **Final Answer** $\boxed{118}$ $< \backslash think >$ \n\n [...]

Figure 11: Sample 658 from MATH500 train —— Solution: $62°$ —— $k = 30$ with DS-Qwen14B

**Step 1 -** Okay, so I've got this geometry problem here. It says that line AB is parallel to line CD, and the measure of angle AXF is 118 degrees. I need to find the measure of angle FYD. Hmm, let me try to visualize this. There's an Asymptote diagram provided, but since I can't actually see the image, I'll have to go off the description. \n\n First, I know that when two lines are parallel, certain properties hold, like corresponding angles being equal, alternate interior angles being equal, and so on. Maybe I can use those properties here \n\n
**Step 2 -** Looking at the Asymptote code, it defines points A, B, C, D, E, F, X, and Y. A, B, C, D are the main points, with AB and CD being the parallel lines. E is another point below AB, and F is above CD. Lines are drawn from E to F, which intersects AB at point X and CD at point Y. So, essentially, line EF is a transversal cutting through the two parallel lines AB and CD. \n\n Given that, angle AXF is 118 degrees. Let me try to identify where that angle is located. Point A is on AB, point X is the intersection of EF and AB, and point F is on EF. So angle AXF is at point X, formed by lines AX and FX. Since AB is parallel to CD, and EF is a transversal, there might be some corresponding angles or alternate interior angles that I can use \n\n
**Step 3 -** I need to find angle FYD. Point F is on EF, point Y is the intersection of EF and CD, and point D is on CD. So angle FYD is at point Y, formed by lines FY and DY. That angle is also on the transversal EF, but on the other side of the parallel lines \n\n
**Step 4 -** Since AB is parallel to CD, and EF is a transversal, the corresponding angles at X and Y should be equal. So, if angle AXF is 118 degrees, then the corresponding angle at Y should also be 118 degrees. But wait, angle AXF is on the same side as angle FYD? Or is it on the opposite side? \n\n Wait, angle AXF is above AB, and angle FYD is above CD. Since AB and CD are parallel, the angles formed by the transversal EF should be corresponding. So, angle AXF and angle FYD are corresponding angles. Therefore, they should be equal. So, angle FYD should also be 118 degrees \n\n
**Step 5 -** "But hold on, let me double-check. When you have two parallel lines and a transversal, corresponding angles are equal. So, angle AXF is at the intersection above AB, and the corresponding angle above CD would be angle FYD. So yes, they should be equal.\n\n Alternatively, I can think about the consecutive interior angles. If I consider angle AXF and the angle adjacent to it on the same side, they should add up to 180 degrees because they are consecutive interior angles. So, if angle AXF is 118 degrees, the adjacent angle on the same side would be 180 - 118 = 62 degrees.\n\n But angle FYD is on the other side, so maybe it's another corresponding angle. Wait, no, actually, angle FYD is the corresponding angle to angle AXF, so it should be equal, *118 degrees* \n\n
[...]
**Step 12 -** Wait, but just to make sure, let me think about the transversal EF. At point X, we have angle AXF = 118 degrees. Since AB is parallel to CD, the corresponding angle at Y should be the same. So, angle FYD is 118 degrees. \n\n Yes, that makes sense. \n\n Alternatively, if I consider triangle FXD, but I don't think that's necessary here because we can solve it using just the properties of parallel lines and transversals. \n\n So, I think I'm confident that angle FYD is 118 degrees. \n\n **Final Answer** $\boxed{118}$ $< \backslash think >$ \n\n
[...]

Figure 12: Sample 658 from MATH500 train —— Solution: $62°$ —— $k = 300$ with DS-Qwen14B

## G.2 SEMANTIC PROPERTIES IMPLIED BY THE PARAMETER $k$

**Objective.** To complete the previous section, we adopted a complementary approach to reinforce our selection. We examine the semantic properties of reasoning steps. We show that the choice of the segmentation parameter $k$ also affects the meaningfulness and coherence of each step. This ablation study has two objectives:

1. We demonstrate that certain values of $k$ enhance the semantic meaning of individual steps, making them more self-contained and interpretable,
2. An appropriate value of $k$ improves the accuracy of our step-tagger, since the semantic coherence impact the detection performance of step types using text classifiers.

**Methodology.** To address these objectives, we selected different values of $k$, namely: $1, 30, 100, 150, 200, 300$. We then constructed their corresponding *datasets* by segmenting the same reasoning traces according to the different parameters $k$ that we selected (Section 3.2). We selected the MATH500 train dataset using the seed $42$, obtained from the DS-Qwen14B model. To access the ground-truth, we re-labeled each datasets using the same method defined in Section 5.1.

**Impact of $k$ on the tag distribution.** Figure 13 shows that the parameter $k$ affects both number of steps obtained and the distribution of ground-truth labels step-types. To control this distributional effects, we considered two dataset variants by pre-processing the datasets as follows: **(a) Balanced:** sampling the data to achieve a 50/50 distribution of positive and negative classes, **(b) Downsampling:** down-sampling the datasets to obtain the same number of samples (as per the smallest dataset - i.e. $k = 300$).

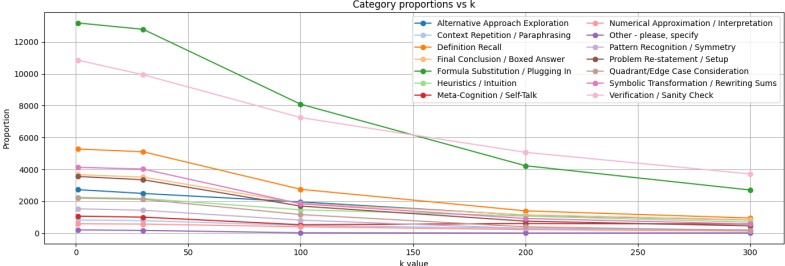

Figure 13: Step-tags distribution as per GPT-4o-mini for each values of $k$ - *ReasonType* taxonomy

For each dataset, we trained binary BERT classifiers with identical hyperparameters (same as in Section 5.1). Similarly to our Step-Taggers, the classification task is step-type detection, where the model predicts whether a given step corresponds to a certain step type $\tau^*$. We conducted experimentation on 2 step-types: **(a) Validation**, **(b) Exploration**. While *Validation* is the one of the most occurrent step-type, *Exploration* has some interesting semantic properties, and is less frequent.

Figure 14 and 15 shows the label distribution of the three variants on the *Verification* and *Exploration* step-type, respectively. While *Downsampling* enable clear comparison of performance for different values of $k$ - same dataset size, *Balancing* labels of datasets investigates the performance of the models with ideal distribution for step-type detection. In contrast, switching off the pre-processing steps preserve the natural distribution of labels, providing insights under more realistic conditions.

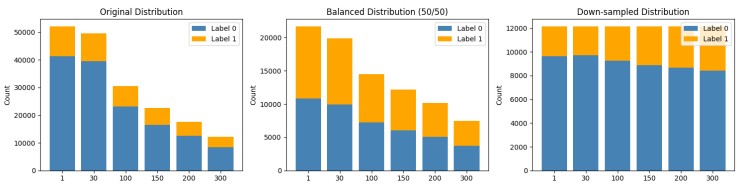

Figure 14: Distribution of dataset variants - Verification step-type

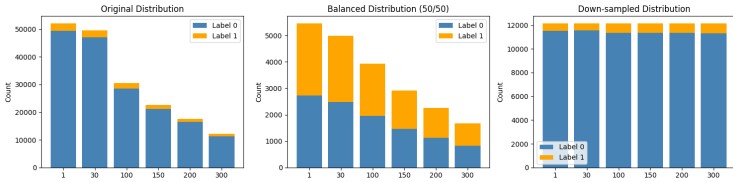

Figure 15: Distribution of dataset variants - Exploration step-type

**Evaluation.** Figure 16(a) and 16(b) present the Micro-F1 and Macro-F1 of the binary BERT classifiers trained on the different dataset types for different values of $k$ and for the *Verification* and *Exploration* step-types, respectively. We first note that the training performed on the Exploration steps lead to higher accuracy than the one achieved on the Verification step (Macro-F1 $0.8 - 0.97$ vs. $0.8 - 0.87$). It could be explain by the nature of the steps. The Verification steps might be more diverse, while the Exploration steps might carry more semantic meaning, making them easier to detect (specifically for the balanced dataset). For imbalanced datasets (Original and Down-sampled), it is worth noting that reporting the Macro-F1 was also important in order to access to the performance of the minority class (positive).

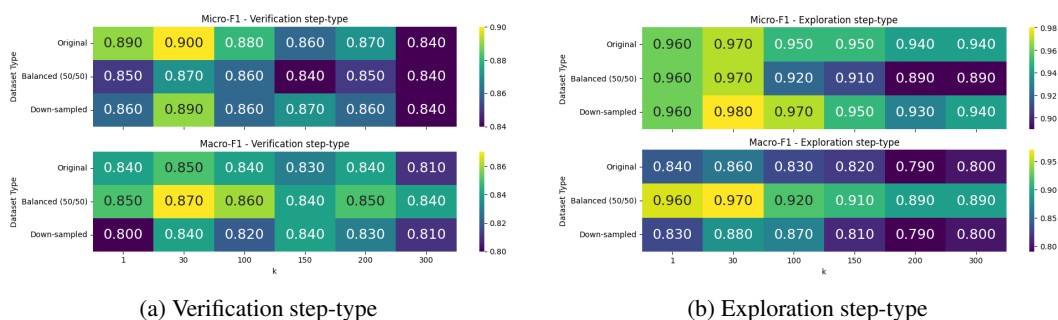

(a) Verification step-type

(b) Exploration step-type

Figure 16: Step-Tagger performance per dataset distribution and step segmentation parameter $k$

Importantly, we observe a general trend of performance for the different configuration. The parameter $k$ seems to impact the performance of the classifiers. Specifically, the performance seems to decrease when $k$ is increase from values ranging from 30 to 300. For the Original Validation, the Micro-F1 drops from $0.9$ for $k = 30$ to $0.84$ for $k = 300$, similarly for the Macro-F1 (from $0.85$ for $k = 30$ to $0.81$ for $k = 300$). Surprisingly, the value $k = 1$ lead to lower performance compared to $k = 30$. We suspect that very small value of $k$ (such as $k = 1$) imply noisy steps. Indeed, Figure 8 in Appendix E confirms this observation. When $k$ is very small, a lot of noisy steps are created, and this could perturb the training.

In addition, for both labels, the Balanced dataset allows for higher performance in term of Macro-F1. It means that classifiers are better at detecting positive classes, at the cost of the performance on the negative class (lower Micro-F1 with respect to the two other datasets). However, same effects on the values of $k$ can be noticed. Moreover, down-sampling the dataset seems to harm slightly the performance, especially for the Verification step-type. It indicates that limiting the dataset size reduces the diversity of training samples for lower values of $k$.

**Takeaway.** This ablation study supports our selection of $k$. Based on the semantic properties of the steps, we found that higher values of $k$ lead to a loss of semantic meaning of the steps. Similarly, small values of $k$ implies noise in the step segmentation - which could perturb the monitoring analysis of the reasoning.

### G.3 Influence of the parameter $k$ on the Step-Tagging Early-Stopping criteria

**Objectives.** In the two previous ablation studies, we have seen that the step segmentation parameter $k$ influenced the amount of information contained in each reasoning step. The following ablation study focus on assessing the impact of the parameter $k$ on the performance of our step-tagging early-stopping criteria. It is composed of two sections: We first evaluate the impact of the parameter $k$ on the constraint threshold $\delta$ of our criteria (Section G.3.1), and then assess the impact of $k$ on the performance of the criteria (Section G.3.2).

#### G.3.1 ST-ES thresholds $\delta$

We have seen that the step segmentation parameter $k$ influences both size and amount of information contained in each steps. To further validate our values of $k$, we are looking at its influence on the threshold $\delta$ of our Step-Tagging Early-Stopping constraint $(\tau, \delta)$, for each step-types $\tau$ of our taxonomy.

**Impact on the constraint values.** To assess the influence of $k$ on the constraint values $\delta$, we adopt a different point of view. For each value of $k$, we re-used the datasets obtained from the reasoning traces of the DS-Qwen14B model on the MATH500 of the training datasets labeled by `GPT-4o-mini` (see Appendix G.2). For each values of $k \in \{1, 30, 100, 150, 200, 300\}$, Figures 18 and 18 compare the Accuracy and the Average number of tokens per sample for each constraints $\delta \in [1, 10]$ for each step-types of the taxonomy, respectively. The positions and speed of convergence will allow us to assess the impact of the $k$ on the constraints values $\delta$.

**Evaluation.** First, we observe that lower values of $k$ (e.g. $k \in \{1, 30\}$) reveal smoother trade-off for both accuracy and token count. Indeed, the curves converge less quickly to the original traces (plateau when $\delta \to \infty$, tending to no-constraints i.e. the standard inference).

In comparison, when $k$ grows, the curves are sharper and tends to converge faster, which offers less trade-off between accuracy and token-count. Furthermore, the curves of the different step-types becomes less distinguishable and overlaps much more than for lower values of $k$. In the context of our Early-Stopping framework, these observations confirm the importance of keeping lower values of $k$ to enable efficient and interpretable inference through the constraints $\{\tau, \delta\}$.

**Takeaway.** This ablation study shows that higher values of $k$ tends to decrease the controllability of the inference using our framework.

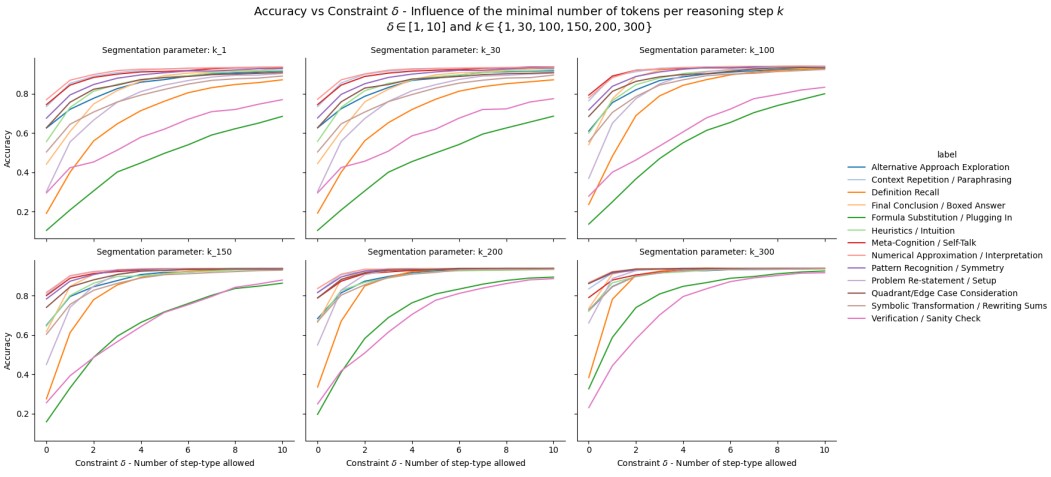

Figure 17: Constraint values $\delta$ vs. Accuracy for each value of $k \in \{1, 30, 100, 150, 200, 300\}$ - Qwen14B on MATH500 train (1,000 samples) - Seed 42, $\delta \in [1, 10]$

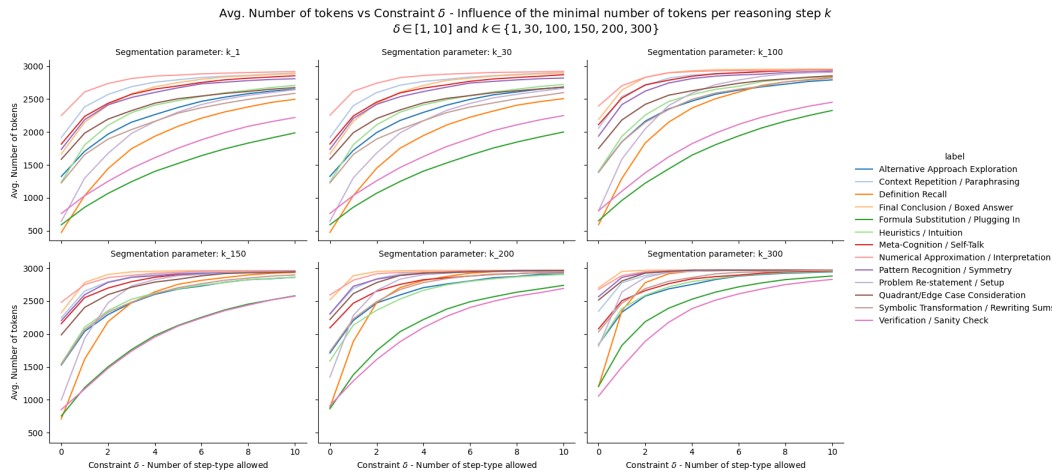

Figure 18: Constraint values $\delta$ vs. Average Number of Tokens per sample for each value of $k \in \{1, 30, 100, 150, 200, 300\}$ - Qwen14B on MATH500 train (1,000 samples) - Seed 42, $\delta \in [1, 10]$

### G.3.2 INFLUENCE OF THE PARAMETER $k$ ON THE ST-ES PERFORMANCE

We complete this analysis with an ablation study on the influence of the step segmentation parameter $k$ on the calibration of the Step-Tagging Early-Stopping. We are assessing if the parameter $k$ affects the performance of our early-stopping framework (i.e. selecting optimal constraints).

**Methodology.** To address our objective, we performed the calibration experiment presented in Section 5.3, using the datasets obtained on our previous experiment, i.e. for the values of $k \in [1, 30, 100, 150, 200, 300]$ (see Section G.2). For each value of $k$, the experiment resulted in one *Pareto Curve*, corresponding to the most efficient constraint parameters among the set of thresholds $\delta$ and step-types $\tau$. For each value of $k$, we re-used the datasets obtained from the reasoning traces of the DS-Qwen14B model on the MATH500 of the training datasets labeled by `GPT-4o-mini`.

**Evaluation.** Figure 19 presents the Pareto Curves of the Step-Tagging Early Stopping applied to DS-Qwen14B on MATH500 train, for different segmentation parameters $k$. To enhance the analysis, Table 6 showcases the AUC of the Pareto curves (restricted to overlapping token ranges between curves). We can observe that the lower values of $k$ (i.e. 1 and 30) lead to higher efficiency (higher accuracy and lower token-count) since their associated Pareto curves are above others for almost every levels of complexity. Table 6 confirms this observation, with $k \in \{1, 30\}$ presenting the higher AUC values.

It is worth noting that the disparities between curves are greater when the complexity is lower. We have seen that the complexity increases the verbosity of models. Therefore, more complexity also increases the number of steps, regardless of the value $k$. For this reason, more complexity increases the opportunity of early-stop, minimizing the impact of $k$.

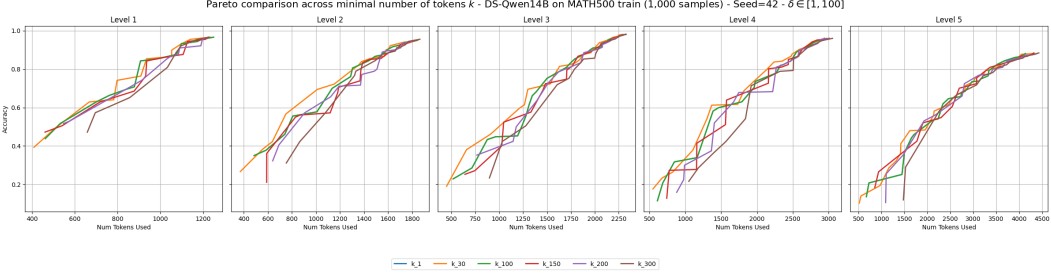

Figure 19: Calibration of Step-Tagging Early-Stopping criteria - Comparison of Pareto Curve for each value of $k \in \{1, 30, 100, 150, 200, 300\}$ - Qwen14B on MATH500 train (1,000 samples), $\delta \in [1, 100]$

| $k$ value | Levels | | | | | Average across Levels |
|---|---|---|---|---|---|---|
| | 1 | 2 | 3 | 4 | 5 | |
| 1 | **456.15** | 832.90 | **1053.85** | 1376.53 | **1829.24** | **1109.74** |
| 30 | 455.42 | **833.33** | 1053.79 | **1376.66** | 1811.43 | 1106.13 |
| 100 | 450.06 | 801.29 | 1017.66 | 1327.18 | 1790.32 | 1077.30 |
| 150 | 440.02 | 774.62 | 1004.62 | 1304.16 | 1775.65 | 1059.81 |
| 200 | 436.50 | 774.68 | 993.03 | 1277.69 | 1789.72 | 1054.32 |
| 300 | 421.18 | 742.81 | 949.12 | 1192.54 | 1726.99 | 1006.53 |

Table 6: Area Under the Curve (AUC) of the Pareto Curves - Restricted to overlapping token range between curves

**Takeaway.** The parameter $k$ influences the performance of the Step-Tagging Early-Stopping. Lower values of $k$ seems to increase the performance since these values lead to finer-grained segmentation (and therefore more flexible early-stopping). However, the divergence in performance seems to reduce when the complexity of question (i.e. verbosity of the model) increases.

### G.4 SUMMARY OF TAKEAWAYS

Our ablation studies contributed to find and validate the parameter $k$, minimal number of token per step, of our definition of reasoning step. From the Sections of this Appendix, we can formulate three main takeaways:

1. The Ideal-Early Stopping ($\mathcal{IES}$) criteria seems to be a good signal for selecting the segmentation parameter $k$. We applied this method and compared early-stopped traces from different values of $k$. For the three LRMs that we selected, we observe that the $\mathcal{IES}$ accuracy drops when values of $k$ increase. Lower values of $k$ seems to result in steps being more self-contained, including a minimal number of thoughts/conclusion, allowing better early-stopping performances.

2. Based on the performance of sentence classifier trained on datasets with different values of $k$, we found that higher values of $k$ lead to a loss of semantic meaning of the steps. Similarly, small values of $k$ implies noise in the step segmentation - which could perturb the monitoring analysis of the reasoning.

3. An ablation study on the calibration process - to find the most efficient constraints per dataset and models - shows that higher values of $k$ tends to decrease the performance and controllability of the inference using our framework.

For our reasoning step definition, these takeaways confirms that users needs to carefully select optimal value of $k$, since this parameter has direct implication on both information contained and semantic properties of the steps.

# H  VALIDATION OF THE REASONTYPE TAXONOMY

## H.1  REASON-TYPE TAXONOMY FOR IDENTIFYING REASONING BEHAVIORS

**Objective.** This ablation study is looking at further validating our ReasonType taxonomy. In other words, we are investigating whether our proposed taxonomy captures meaningful distinctions in reasoning steps. We are looking to demonstrate that:

1. The ReasonType taxonomy enable semantic distinction of the type of reasoning.
2. Our annotation method with the `GPT-4o-mini` model, coupled with the ReasonType taxonomy, is a robust method to access to the ground-truth labels of the reasoning steps.

**Methodology.** To address our objective, we compare the performance of BERT classifiers across Original labels (OG - from `GPT-4o-mini` annotation using the ReasonType taxonomy), and shuffled labels for three step-types, namely: *Verification*, *Exploration* and *Self-Talk*. For the shuffled labels version, we took the exact same proportion of positive labels as in Original datasets, and used random shuffle with a seed of $42$. Each experiment is run on the same training and testing dataset, i.e. the steps obtained with a segmentation parameter $k = 30$, from the MATH500 training dataset on the DS-Qwen14B model. We trained BERT classifiers following the exact same training configuration (see Section 5.1). To compare performances, we report both training loss, and classification metrics (precision and recall on both classes, along with macro and micro average.)

**Evaluation.** Figure 20 shows the training loss of the Original and Shuffled versions, for the three labels. We observe that models trained on the Original labels presents significant lower losses, and are smoothly decreasing. It demonstrate that the Original datasets contains meaningful patterns between reasoning steps and their labels. In comparison, the models trained on shuffled labels present almost constant loss, relatively higher than the one from the Original labels.

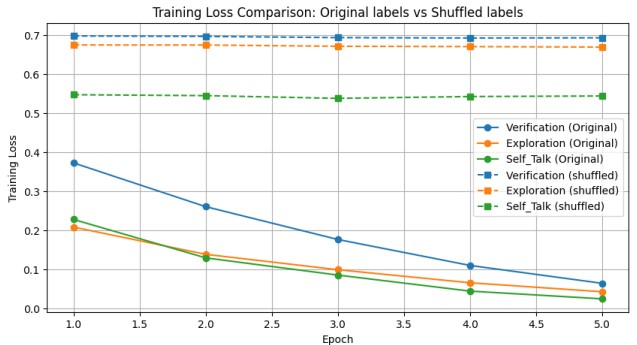

Figure 20: Training losses - ReasonType vs. Shuffled labels

Furthermore, Figures 21(a) and 21(b) show the Precision and Recall classification metrics on the testing dataset, respectively. For Original runs, both classes (0 and 1) achieve good performance despite dataset imbalancity, with Macro average Precision and Recall lying between $0.76$ and $0.90$ across labels. In comparison, shuffled runs presents poor results, with models failing in predicting positive classes - Precision and Recall of class $1$ between $0.00$ and $0.06$. Along with the training loss, theses metrics highlight that the models trained on shuffles labels cannot learn meaningful relations between steps and labels. In comparison, Original labels (from the ReasonType taxonomy) resulted in satisfying model performance, and smooth training.

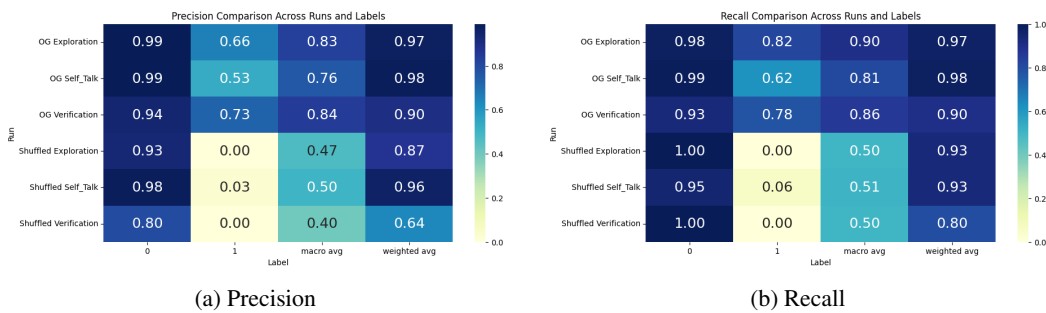

(a) Precision                              (b) Recall

Figure 21: Precision and Recall - ReasonType vs. Shuffled labels

**Takeaway.** Overall, these results comforts our finding that the ReasonType taxonomy labels enable annotation methods to results in reasoning steps carrying semantic meaning.

## H.2 TRACKING STEP-TYPES TO DESIGN INTERPRETABLE EARLY-STOPPING CRITERIA

**Objective.** This ablation study is looking at motivating our Step-Tagging Early-Stopping (ST-ES) framework, as well as our taxonomy. We have seen that the ST-ES approach is an interpretable early-stopping criteria since the user can select which type of reasoning step to limit to make inference more efficient. However, we are wondering if simpler approaches such as a *simple step-count* is more effective. Despite having less requirements - since no step-tagging is required - only stopping the reasoning based on the number of steps generated could potentially yield to better results.

**Methodology.** To assess this alternative approach is more effective, we lead the same calibration study as in Section 5.3 on the MATH500 training dataset, using the 3 selected LRMs. For clarity, we applied a common threshold across every levels of complexity of the dataset, and reported their average accuracy and token-count. To compare our taxonomy with the *simple step-count* approach, we present two types of runs:

- **ReasonType taxonomy:** We performed the same experimentation for each single labels of the ReasonType taxonomy. For each tags $\tau_i \in \mathcal{T}$, we combined a threshold value $\delta \in [1, 20]$.
- **Simple Step-count:** The same experiment is done without the labels. To do so, we only applied simple step-count for thresholds $\delta \in [1, 100]$.

**Evaluation.** Figures 22, 23, and 24 present the experiment on the DS-Llama8B, DS-Qwen14B, and QwQ-32B models, respectively. For both types of runs, each combination of threshold and step-type results in a point (Average Number of Tokens, Accuracy). For the ReasonType taxonomy, each step-types forms a blue curve, with color gradient and different markers to differentiate the tags of the taxonomy. The same experiment with a unique label (equivalent to a simple step-count early-stopping) resulted in a unique curve, print in red.

We observe over the three models that the red curve is almost constantly under the blue curves, in particular for the DS-Qwen14B model. Specifically, for token-count from $1,500$, the red curve is bellow the curves of the *Problem-Restatement*, *Exploration*, and *Intuition*, for the DS-Llama8B, DS-Qwen14B, and QwQ-32B models, respectively. Furthermore, for lower token-counts, the red curve is still under the blue curves of the *Problem-Restatement* for both DS-Llama8B, DS-Qwen14B models. Overall, it means that for a given number of tokens, implementing a stopping criteria based on the type of the steps seems to yield to higher accuracy than simply stopping the generation based on the number of steps.

**Takeaways.** Beyond interpretability, this experiment shows that tracking the types of steps yields to better performance than an early-stopping criterion based on the step-count.

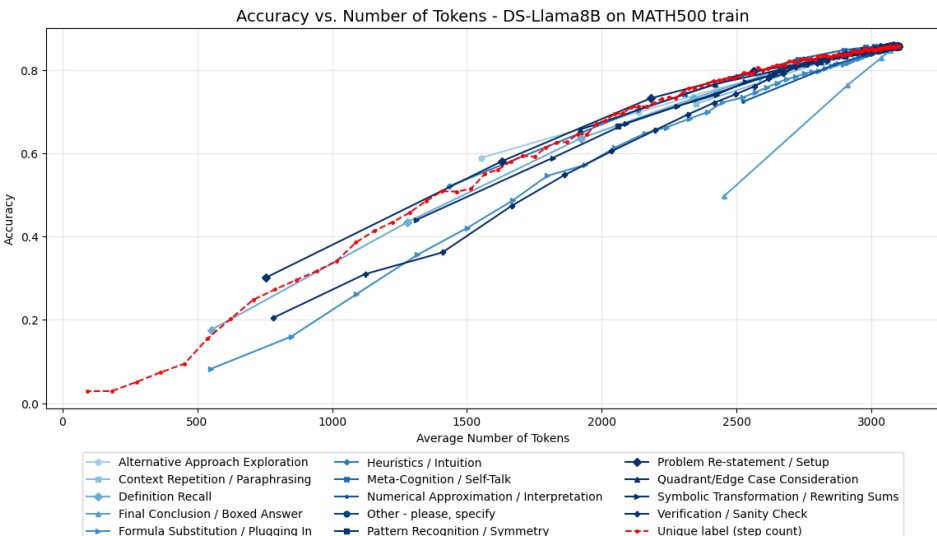

Figure 22: Accuracy vs. Average number of tokens - Step-Tagging Early-Stopping curves per step-types (in blue) and simple step-count (in red) - DS-Llama8B on MATH500 train

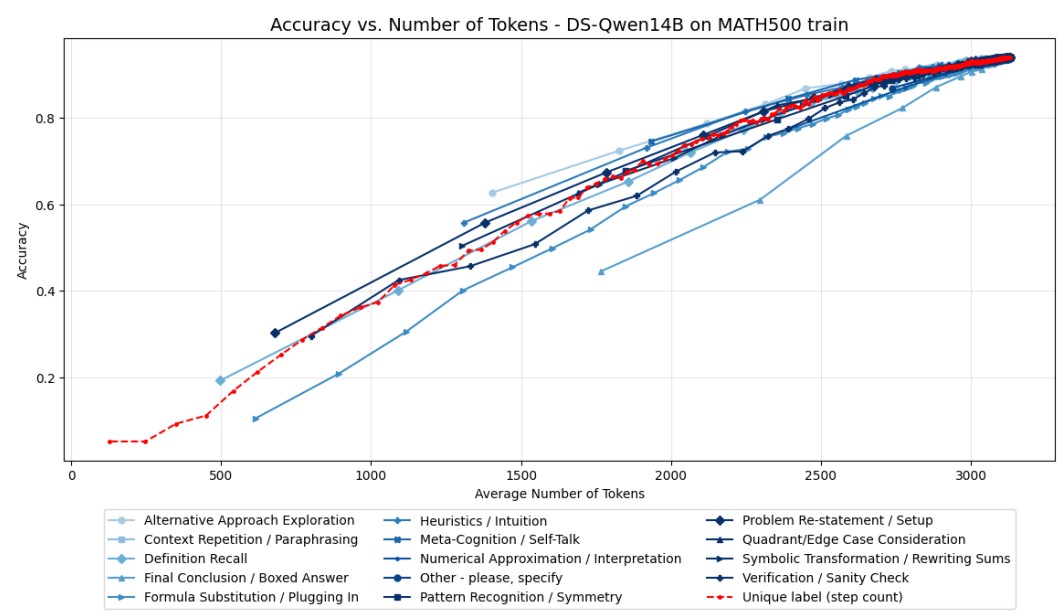

Figure 23: Accuracy vs. Average number of tokens - Step-Tagging Early-Stopping curves per step-types (in blue) and simple step-count (in red) - DS-Qwen14B on MATH500 train

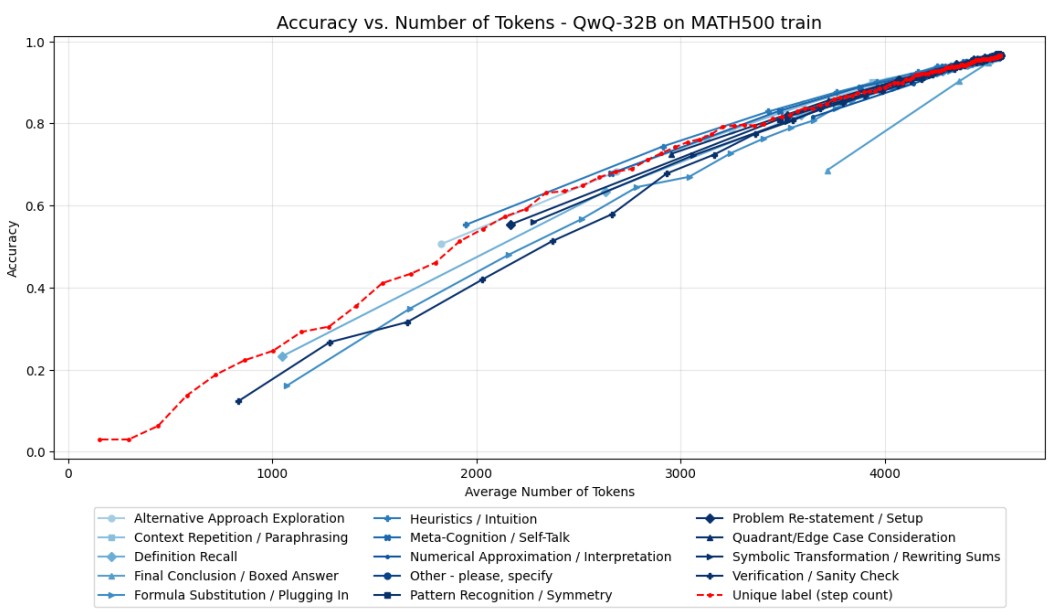

Figure 24: Accuracy vs. Average number of tokens - Step-Tagging Early-Stopping curves per step-types (in blue) and simple step-count (in red) - QwQ32B on MATH500 train

### H.3 CERTAIN STEP-TYPE ARE BETTER EARLY-STOPPING CRITERIA

**Objective.** In section 5.3, we observed that selecting the step-tagging early-stopping constraint (both threshold and step-types) was not trivial since it depends on models, and complexity of a given problem. This ablation study is looking at showing that certain step-types are better to apply our constraint. Specifically, we are looking to demonstrate that our taxonomy allows us to state that different step-types are leading to different efficiency trade-off.

**Methodology.** Similarly to the sub-section H.2, we are showing the same experiment, but we focus on the curves resulting from the ReasonType taxonomy (in blue). We plot the same Figures, but with one distinct color and marker for each step-type to better differenciate the curves.

**Evaluation.** Figure 25 present the different curves obtained applying our early-stopping framework on the labels of the ReasonType taxonomy, for the DS-Qwen14B on the train MATH500 dataset. We observe that each step-type results in curves with different lengths (token-count range), and widths (accuracy range).

First, the *Exploration* curve seems to be the most efficient for moderate to high token-count range (from approximately $1,400$ to $3,000$). Indeed, the curve stands above all other curves, meaning that for constraints with the *Exploration* tags results in highest accuracy with equivalent token-budget.

In contrast, other step-types such as *Problem Re-statement*, *Verification* or *Definition Recall* cover larger token-count range (from $500 - 750$ to $3,000$). They introduce more flexibility for limited token-budget, but appears less efficient when the token-count grows.

Figures 26 and 27 present similar findings for the DS-Llama8B and QwQ-32B models, respectively.

**Takeaway.** Overall, we can conclude that our taxonomy coupled to our methodology allows us to demonstrate that all reasoning step types are not equally contributing to the reasoning progression.

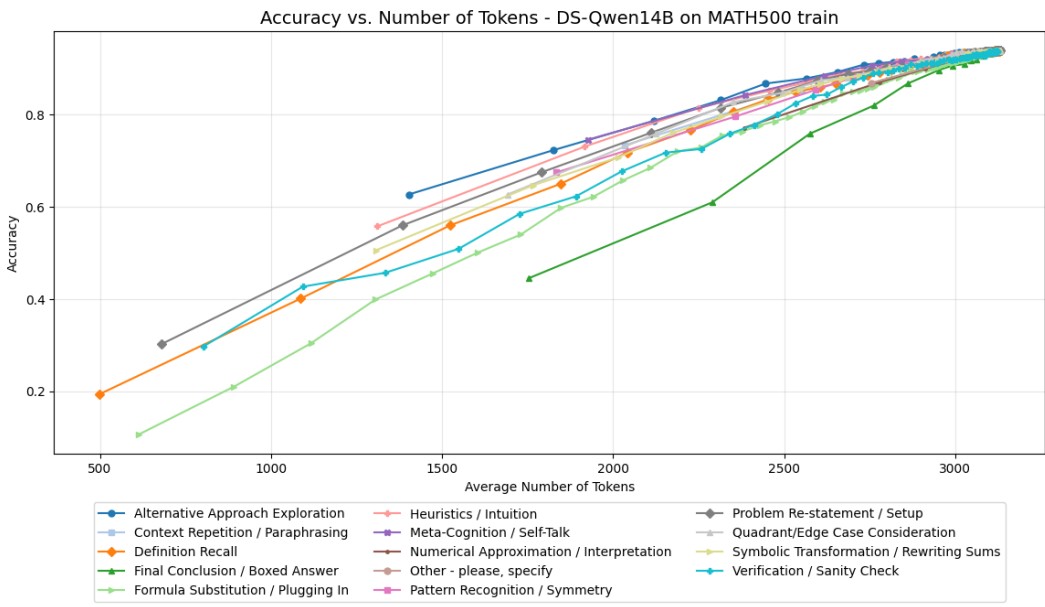

Figure 25: Accuracy vs. Average number of tokens - Step-Tagging Early-Stopping curves per step-types - one color per step-type - DS-Qwen14B on MATH500 train

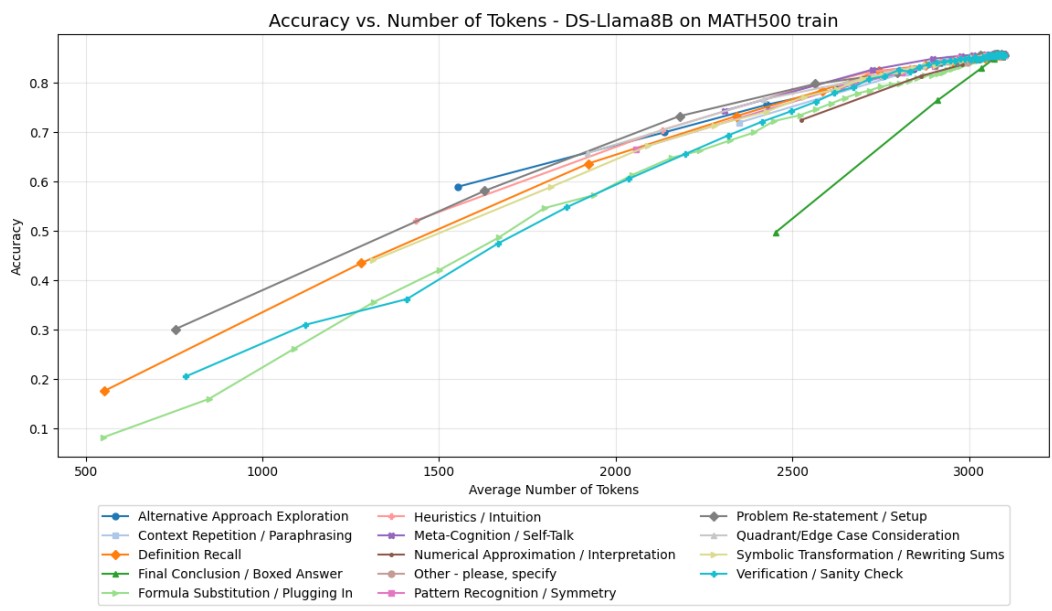

Figure 26: Accuracy vs. Average number of tokens - Step-Tagging Early-Stopping curves per step-types - one color per step-type - DS-Llama8B on MATH500 train

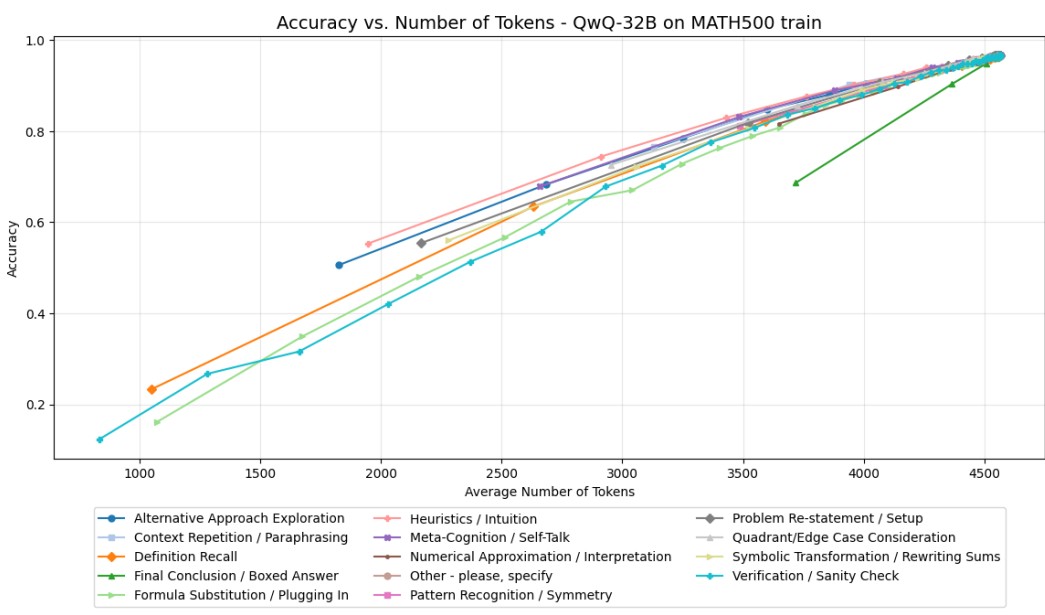

Figure 27: Accuracy vs. Average number of tokens - Step-Tagging Early-Stopping curves per step-types - one color per step-type - QwQ-32B on MATH500 train

## H.4 COMPARISON TO ALTERNATIVE TAXONOMY

**Objective.** In this ablation study, we assess how robust our taxonomy is for the Step-Tagging Early-Stopping criteria. To do so, we run the calibration experiment that we conducted on different alternative versions of our original taxonomy.

**Alternative Taxonomies.** Our original taxonomy is wide and fine-grained, containing 13 categories of labels (excluding the placeholder label "Other"). Therefore, we reduce the number of labels in the taxonomy, and grouped similar labels at different levels of abstraction. Table 7 shows resulting the taxonomies, considering from $13, 6, 4, 2$, and 1 labels. A unique label represent the simplest form of constraint, where we obtain only a constraint on the number of steps.

| Original Taxonomy | 6-labels | 4-labels | 2-labels | 1-label |
|---|---|---|---|---|
| Problem Re-Statement Context Repetition Definition Re-call | Setup | Early Reasoning | Early Reasoning | Reasoning |
| Formula Substitution Symbolic Transformation | Manipulation | Mid Reasoning | | |
| Edge Case Pattern Recognition | Analysis | | | |
| Verification Heuristic / Intuition | Checking | Late Reasoning | Late Reasoning | |
| Exploration Interpretation Self-Talk | Meta Reasoning | | | |
| Final Conclusion | End Reasoning | End Reasoning | | |

Table 7: Alternative taxonomies - we regrouped labels at different levels of abstraction to observe the impact of the taxonomy on the Step-Tagging Early-Stopping criteria

**Methodology.** To address our objective, we performed the calibration experiment presented in Section 5.3, using the different taxonomies (i.e. vocabulary of tags $\mathcal{T}$). For each taxonomies, the experiment resulted in *Pareto Curve*, corresponding to the most efficient constraint parameters given a threshold $\delta$ and a type-step $\tau$ (lying in the taxonomy tested). For each models, we re-used the MATH500 and GSM8K training datasets labeled by `GPT-4o-mini` using our methodology explicated in Section 5.1. We then merged labels as in Table 7.

The process of merging labels artificially increases the number of labels per datasets. Therefore, to encompass a wider range of early-stopping values, we performed the calibration for threshold values $\delta \in [1, 100]$. To evaluate the impact of downsizing the original taxonomy, we look at the position of the Pareto Curves compared to each other. The taxonomy giving the most efficient constraints is the one located at the top left (maximizing the accuracy while minimizing the number of tokens).

**Evaluation.** We can note that downsized tag vocabularies outperformed our Original taxonomy for low complexity queries (e.g. 1 and 2), specifically for 1 and 2-labels taxonomies. It can be justified by the nature of the low-label taxonomies. For the purple curve, it acts like a simple step count constraint. For low-level complexity, the reasoning traces are less verbose (less tokens), meaning that it is easier to find a simple form of constraint.

However, the Original and Label-6 taxonomy seem to perform well on higher level of complexity. For higher level of complexity, it seems that the semantic meaning of the steps plays a role in the determination of the final constraint. Nevertheless, it is worth noting that the differences observed are relatively small. Pareto Curves of Original and Label-6 are sensibly close, meaning that our approach seems to be robust to the granularity of the taxonomy.

**Takeaway.** Our framework seems to be robust with regards to the taxonomy selected. Nevertheless, fine-grained taxonomies seems to lead to better performance for higher degree of complexity of questions, while simpler taxonomies might be more adapted to simpler problems (i.e. less verbose).

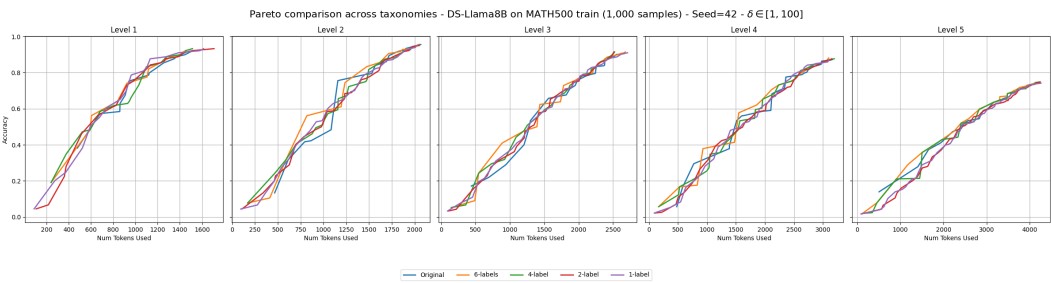

Figure 28: Calibration of Step-Tagging Early-Stopping criteria - Comparison of Pareto Curve for each taxonomies - Llama8B on MATH500 train (1,000 samples)

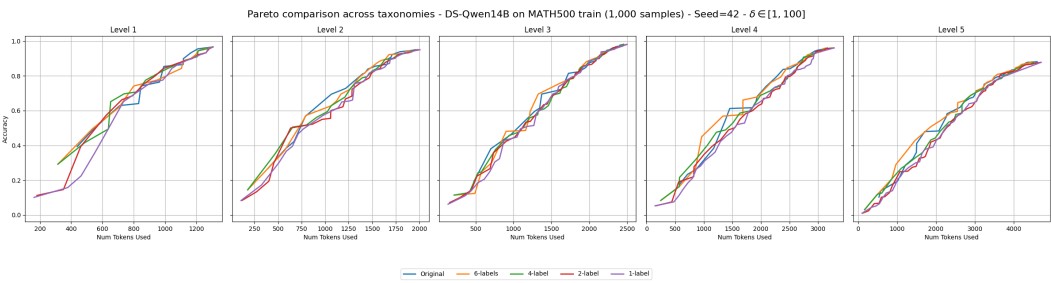

Figure 29: Calibration of Step-Tagging Early-Stopping criteria - Comparison of Pareto Curve for each taxonomies - Qwen14B on MATH500 train (1,000 samples)

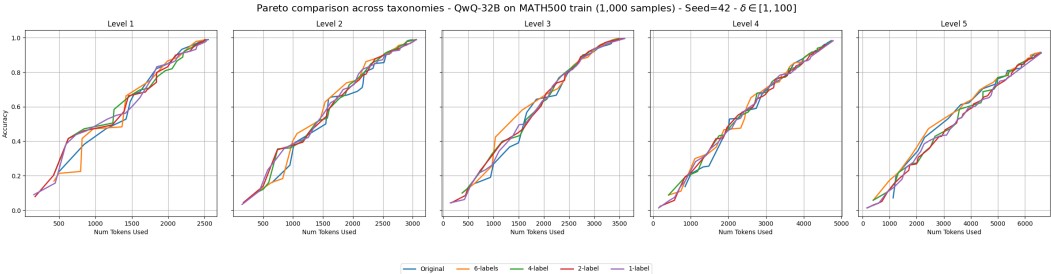

Figure 30: Calibration of Step-Tagging Early-Stopping criteria - Comparison of Pareto Curve for each taxonomies - QwQ-32B on MATH500 train (1,000 samples)

### H.5 SUMMARY OF TAKEAWAYS

Overall, our ablation studies validate our taxonomy. Key takeaways are the followings:

1. The ReasonType taxonomy labels enable annotation methods to results in reasoning steps carrying semantic meaning.

2. Beyond interpretability, tracking the types of steps using the ReasonType taxonomy yields to better performance than an early-stopping criterion based on simple step-count.

3. The ReasonType taxonomy coupled to our methodology allows us to demonstrate that all reasoning step types are not equally contributing to the reasoning progression.

4. Our framework seems to be robust with regards to the taxonomy selected. Nevertheless, fine-grained taxonomies - such as ReasonType - seems to lead to better performance for higher degree of complexity of questions, while simpler taxonomies might be more adapted to simpler problems (i.e. less verbose inferences).

# I GENERATION AND RELIABILITY OF THE REASONTYPE TAXONOMY

## I.1 GENERATION OF THE TAXONOMY

In Section 5, we provided a high-level description of our methodology to generate a taxonomy of reasoning step. In this section, we will describe our methodology in more details.

**Methodology.** Our goal in constructing the *ReasonType* taxonomy is to observe the behavior of LRMs and look at aggregating similar types together to follow the generation of LRMs closely. Because no taxonomy of reasoning steps exists, we used an open-ended labeling procedure: we prompted a model to generate free-form step-type labels, and manually merged common labels together. We took inspiration from previous work, who relied on the summarization and behavior detection capabilities of strong models such as the `GPT-4o-mini` model, as this method has proven to be effective (Galichin et al., 2025; Kuznetsov et al., 2025).

| Dataset | Model | # Steps | # Unique Tags |
|---------|-------|---------|---------------|
| MATH500 | DS-Llama8B | 376 | 162 |
|         | QwQ-32B | 430 | 181 |

Table 8: Step for taxonomy generation

We first generated 100 reasoning traces from the MATH500 train dataset, using DS-Llama8B and QwQ-32B, to obtain a pool of reasoning steps (20 samples from each complexity level). We obtained a pool of 806 reasoning steps (see Table 8). Each step is then passed to `GPT-4o-mini` using our Taxonomy prompt (see Figure 46 in Appendix M). We obtained an open-ended label for each step-type.

Figure 31 presents the most frequent labels generated. We observe that, even though our prompting was open-ended (without particular instruction for the model to generate certain type of labels), some labels were obtained frequently, such as *Problem Identification/Re-Statement*, *Substitution*, or *Verification*. We manually inspected the labels obtained, and merged labels that seemed to have a common signification. Table 9 illustrates several examples.

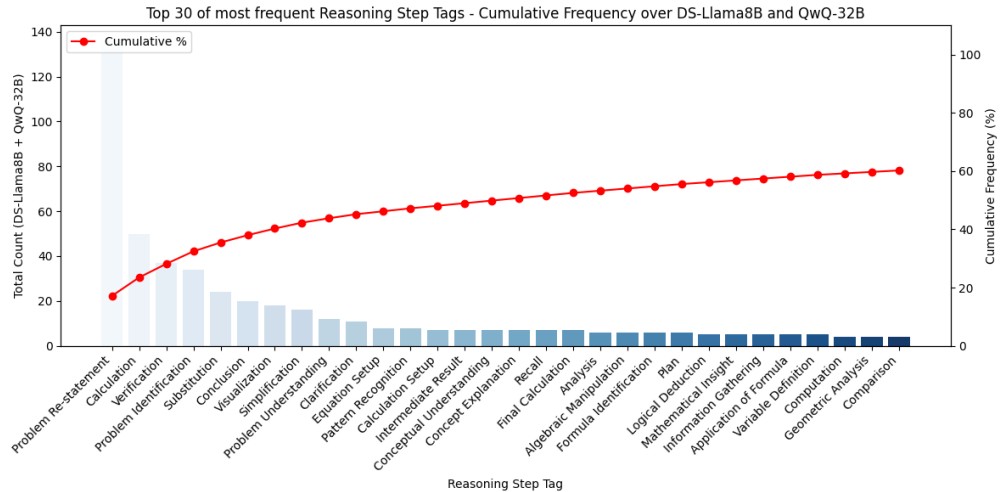

Figure 31: Most frequent labels obtained from open-end label generation (see prompt in Figure 46 in Appendix M)

**Limitations.** To over-come the domain dependency, we could rely on the OpenThoughts-114k dataset. Indeed, it seems to provide a diverse set of possible behavior for the models. However, it would come at a high processing cost. While the ReasonType taxonomy might not be optimal, we showed that it reflects the different steps of the model, and offers enough consistency and granularity to train accurate classifiers for our study.

| Tags from ReasonType | DS-Llama8B | QwQ-32B |
|---|---|---|
| Problem Re-statement / Setup | Problem Re-statement, Problem Identification, Clarification | Problem Re-statement, Coordinate Setup, Step-by-Step Breakdown |
| Definition Recall | Recall of Relevant Concepts, Definition Recall | Recall, Rule Recall, Definition Explanation |
| Formula Substitution | Value Substitution, Application of Formula, Calculation | Equation Setup, Equation Manipulation, Application of Formula |
| Exploration | Approach Exploration, Exploration of Alternatives | Exploration |
| Self-Talk | Procedure Explanation, Plan of Action, Comparison of Options | Confirmation, Reflection, Logical Breakdown |
| Verification | Confirmation, Verification | Backward Calculation, Example Verification, Assumption Checking |
| Final Answer | Final Evaluation, Final Calculation | Final Evaluation, Final Calculation |

Table 9: Example of categories from the ReasonType Taxonmy obtained by merging labels from the annotation - each set of labels were generated using our prompt and sampled reasoning steps, and we manually merged common labels to create the ReasonType taxonomy.

## I.2 RELIABILITY OF GPT-4o-MINI AS AN ANNOTATOR

**Claim.** Our framework rely on the annotation capability of the GPT-4o-mini model. Since the model is large and achieved great performance on a range of tasks, we assumed that the model is able to provide us with labels of good quality. In this section, we will observe and analyse the reliability of the annotation of the steps by the GPT-4o-mini model.

**Methodology.** To verify our claim, we sampled $1,000$ reasoning steps of DS-Qwen14B model from its inference on the MATH500 dataset. We then annotated each steps 5 times using GPT-4o-mini, and observe the agreement of each annotation. In addition, we also compared the annotation agreement between the GPT-4o-mini model, and 3 additional models: GPT-4o (a larger, closed-source model), llama-3-3-70b-instruct, and Mixtral-8x22B-Instruct-v0.1 (both open-source and smaller relatively to the two model selected). This setup allows us to assess both the internal consistency of GPT-4o-mini and its alignment with other model annotators.

**Experimental Design.** We refer to self-model agreement as the *Inner-model agreement* (agreement between different runs of the same model on the same reasoning steps). The inner-model agreement is measured using the Fleiss' kappa metric. Indeed, the Fleiss' kappa measure the agreement between more than 2 annotators, making it suitable to compare many annotation trials (Moons & Vandervieren, 2025). Furthermore, we call agreement between two different model the *Inter-model agreement*. To compute this, we first selected the most consistent label generated across the different runs for each model, and we measured the Cohen's Kappa (Badshah & Sajjad, 2025). For this experiment, we used the OpenAI default decoding parameters for the GPT models (temperature = 1.0 and top-p = 1.0). Same parameter was set for the other models selected.

| Models | GPT-4o-mini | GPT-4o | Llama-3-3-70b | Mixtral-8x22B |
|---|---|---|---|---|
| GPT-4o-mini | **0.780** | 0.601 | 0.457 | 0.392 |
| GPT-4o | | **0.799** | 0.445 | 0.384 |
| Llama-3-3-70b | | | **0.722** | 0.398 |
| Mixtral-8x22B | | | | **0.587** |

Table 10: Agregation metrics of the annotation process - $1,000$ samples reasoning steps from the DS-Qwen14B model on the MATH500 dataset - **Fleiss' kappa** in bold (diagonal - inner-model aggregation) - **Cohen's kappa** otherwise (inter-model aggregation)

**Internal consistency of `GPT-4o-mini`.** The Fleiss' Kappa score of 0.780 (see Table 10 in the diagonal) indicates high agreement among multiple independent runs of `GPT-4o-mini` on the same reasoning steps. This demonstrates that the model produces stable and consistent annotations when prompted repeatedly. We observe that the Fleiss' Kappa score seems to decrease as the model size becomes smaller.

**`GPT-4o-mini` against other models.** When comparing `GPT-4o-mini` with other models, we observe significant agreement with `GPT-4o`, with a Cohen's Kappa of 0.601, and moderate agreement with smaller models (0.457 with Llama70b and 0.392 with Mixtral). The higher agreement with `GPT-4o` suggests that larger models tend to produce more stable and higher-quality annotation, while smaller models exhibits more variability.

**Takeaway.** Overall, these results supports that `GPT-4o-mini` is a reliable annotator for reasoning steps. Indeed, for our annotation task, the model appears to be consistent accross multiple runs, and shows meaningful agreement with other strong models. The results obtained on smaller models confirms our decision of selecting `GPT-4o-mini` as a annotator to label the reasoning steps of LRMs.

### I.3 GENERALIZABILITY OF THE TAXONOMY

We proved in Appendix I.2 that `GPT-4o-mini` is a reliable annotator model for our taxonomy and the task of tagging the steps. However, the taxonomy is generated from specific samples of specific models. We already proved to some extend that the Taxonomy is generalizable. Indeed, we obtained satisfying performance of the step-tagging classifiers on models and datasets that were not used to derive the taxonomy, namely DS-Qwen14B, and AIME, GPQA and MMLU-Pro (see Section K).

**Claim.** To further validate our taxonomy, we will test its applicability to 2 additional reasoning models. Our claim is that if the annotation process from `GPT-4o-mini` results in accurate training of step-tagging classifiers, it means that the ReasonType taxonomy is applicable to other models, therefore generalizable.

**Methodology.** To verify our claim, we inferred 2 additional models on both MATH500 and GSM8K, specifically `microsoft/Phi-4-reasoning` and `Qwen/Qwen3-30B-A3B-Thinking-2507` since they are both reasoning models, and comes from additional providers or training methods. We then trained 4 BERT classifiers for each model and dataset. We used 500 and 3,000 samples from the MATH500 and GSM8K train datasets, respectively. We then conducted same analysis as in Appendix G.3.1 to determine the value of $k$ for our definition of steps. From Figure 32, we selected $k = 20$ and $k = 30$ for the Phi-4 and Qwen3-30B-A3B models, respectively.

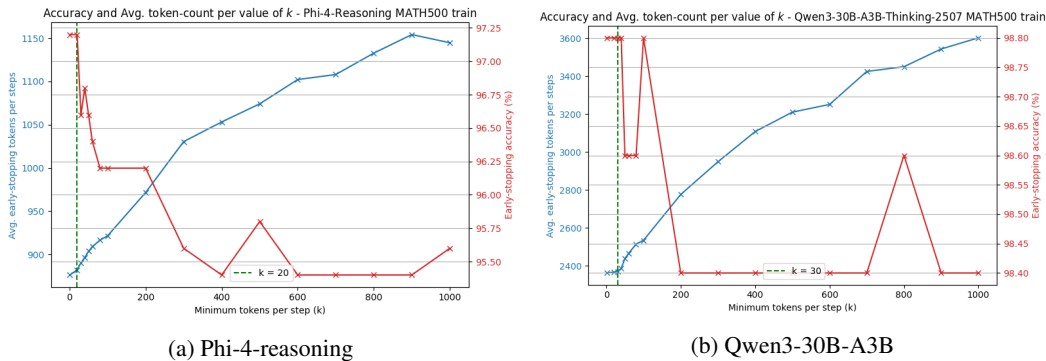

(a) Phi-4-reasoning        (b) Qwen3-30B-A3B

Figure 32: Selecting optimal $k$ - Efficiency of $\mathcal{IES}$

**Performances of the Step-Taggers.** We split the resulting annotated datasets following random 80:20 train/test split. Figure 33(a) and 33(b) show the micro-F1 and macro-F1 for the Phi-4 and Qwen3-30B-A3B models on MATH500 and GSM8K, respectively. We observe satifying performances of Step-Taggers on both models. Specifically, on the `Phi-4` model, we obtained between 0.7 to 0.98 and 0.63 to 0.87 macro-F1 on MATH500 and GSM8K datasets, respectively. The lower

macro-F1 on Exploration can be explained by its low representation in the datasets (around 1% of labels in both datasets).

Similarly, `Qwen3-30B-A3B` obtained satifying perfromance on both MATH500 and GSM8K, with $0.72$ to $0.84$ and $0.76$ to $0.90$ macro-F1, respectively. Performances of both models are comparable to results obtained on the 3 reasoning models selected for our experimentations (see Appendix Q).

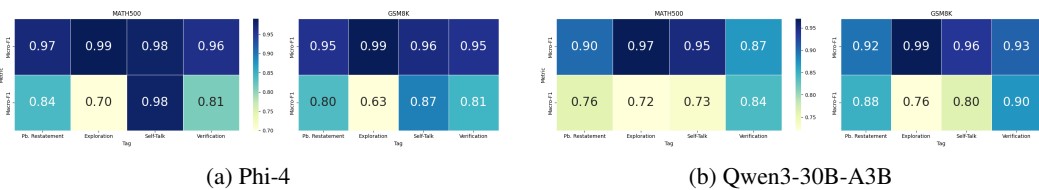

(a) Phi-4                           (b) Qwen3-30B-A3B

Figure 33: Performance of Step-Taggers - seed 42

**Takeaways.** We interpret the satisfying performances of the Step-Taggers trained on other models as further validating the applicability of our taxonomy to other models. Indeed, these models were not used to create our *ReasonType* taxonomy, but still resulted in good performance of Step-Taggers trained on their reasoning traces using our methodology.

# J ANALYSIS OF THE COST OF THE ST-ES FRAMEWORK

## J.1 LATENCY ANALYSIS OF THE LIGHT-WEIGHT STEP-TAGGER

In our experimentation, we reported the number of tokens generated by the models as a proxy for the resources and latency of the inferences of the models. Indeed, we performed the experimentation offline and the number of tokens could have been computed without the need of re-running the inferences.

**Motivations.** By definition, our early-stopping criteria requires an external module to perform on-line annotation of the generated reasoning steps. To further validate our approach, this sub-section aims to analyze the latency introduced by our lightweight step classifiers.

**Methodology.** Our experiments have been performed offline. However, we have access to both standard runtime, number of tokens, and the number of tokens of the early-stopped samples. In addition, we recorded the runtime of the lightweight classifiers when annotating each steps. To allow fair comparison of the inference latency of the early-stopped samples, we need to estimate the early-stopped runtime.

The runtime of the early-stopped samples is composed of three components:

$$r_{\text{ST-ES}} = r_{\text{stopped}} + r_{\text{step\_classifier}} + r_{\text{completion}} \tag{6}$$

where $r_{\text{ST-ES}}$ is the total runtime of the early-stopped sample, $r_{\text{stopped}}$ is the runtime of the model generating tokens from the start to the early-stopping condition, $r_{\text{step\_classifier}}$ is the runtime of the lightweight classifier annotating each reasoning step, and $r_{\text{completion}}$ is the additional runtime of the model when prompted to generate its current best answer (see Early-Stopping criteria in Section 4).

**Linear Assumption.** While $r_{\text{step\_classifier}}$ and $r_{\text{completion}}$ are accessible though our computations, we need to estimate $r_{\text{stopped}}$ since we ran our experimentation offline. In the literature, researchers seems to acknowledge a linear relationship between the runtime and the number of token generated: *"During the AR [Auto-Regressive] decoding, output tokens are generated sequentially, conditioned on all previously generated tokens. As a result, the decoding time (i.e. inference latency) increases linearly with the decoding length."* (Oh et al., 2022). This assumption is further validated by our experimentation. Indeed, Figure 34 reports linear regressions between the number of tokens and the runtime of the three selected LRMs, on the MATH500 over the 5 seeds we used.

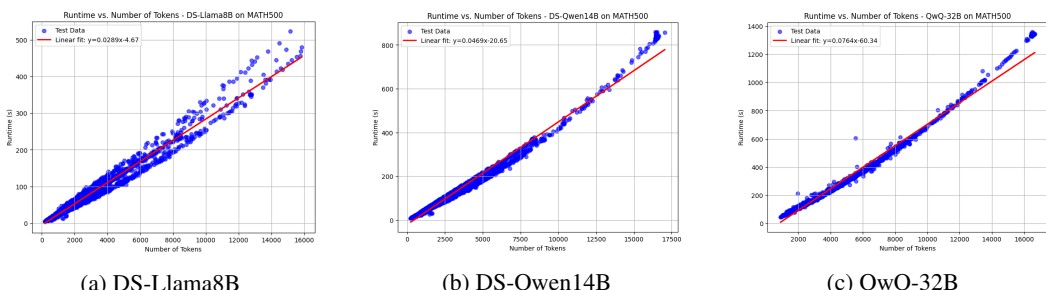

| (a) DS-Llama8B | (b) DS-Qwen14B | (c) QwQ-32B |

Figure 34: Linear relationship between number of tokens and runtime

**Estimation of $r_{\text{stopped}}$.** Assuming linear relationship between number of tokens generated and the runtime, we estimate $r_{\text{stopped}}$ by fitting a linear regression between the number of generated tokens and the observed runtime of the full runs (see Figure 34).

$$r_{\text{stopped}} \approx \alpha \cdot N_{\text{ST-ES}} + \beta \tag{7}$$

where $N_{\text{tokens}}$ is the number of tokens generated in a full run. For each early-stopped samples, we predict its runtime using the regression model.

| Dataset | Models | $r_{standard}$ (s) | Config. | $r_{stopped}$ (s) | $r_{step\ classifier}$ (s) | $r_{completion}$ (s) | $r_{ST\text{-}ES}$ (s) | Speed-up ($\uparrow$) |
|---------|--------|------------|---------|-----------|------------------|-------------|------------|----------------|
| MATH500 | DS-Llama8B | 102.32 | ST-ES (95%) | 89.86 | 0.32 | 3.42 | 93.59 | ×1.09 |
| | | | ST-ES (90%) | 79.99 | 0.29 | 3.14 | 83.42 | ×1.23 |
| | | | ST-ES (85%) | 62.59 | 0.21 | 2.86 | 65.66 | ×1.56 |
| | | | ST-ES (Router) | 70.85 | 0.25 | 2.99 | 74.10 | ×1.38 |
| | DS-Qwen14B | 138.25 | ST-ES (95%) | 124.99 | 0.34 | 5.97 | 131.30 | ×1.05 |
| | | | ST-ES (90%) | 118.92 | 0.34 | 5.65 | 124.91 | ×1.11 |
| | | | ST-ES (85%) | 86.52 | 0.24 | 5.09 | 91.85 | ×1.51 |
| | | | ST-ES (Router) | 97.66 | 0.25 | 5.16 | 103.08 | ×1.34 |
| | QwQ-32B | 281.51 | ST-ES (95%) | 217.05 | 0.18 | 8.47 | 225.7 | ×1.25 |
| | | | ST-ES (90%) | 199.22 | 0.17 | 7.63 | 207.02 | ×1.36 |
| | | | ST-ES (85%) | 179.56 | 0.16 | 7.87 | 187.59 | ×1.50 |
| | | | ST-ES (Router) | 212.6 | 0.22 | 7.64 | 220.47 | ×1.28 |
| GSM8K | DS-Llama8B | 25.83 | ST-ES (95%) | 16.37 | 0.06 | 2.36 | 18.79 | ×1.37 |
| | | | ST-ES (90%) | 12.60 | 0.05 | 2.31 | 14.96 | ×1.73 |
| | | | ST-ES (85%) | 10.11 | 0.04 | 2.30 | 12.45 | ×2.07 |
| | DS-Qwen14B | 27.74 | ST-ES (95%) | 17.85 | 0.04 | 3.67 | 21.56 | ×1.29 |
| | | | ST-ES (90%) | 18.99 | 0.04 | 3.66 | 22.69 | ×1.22 |
| | | | ST-ES (85%) | 15.86 | 0.04 | 3.74 | 19.64 | ×1.41 |
| | QwQ-32B | 117.28 | ST-ES (95%) | 82.93 | 0.09 | 8.04 | 91.06 | ×1.29 |
| | | | ST-ES (90%) | 75.23 | 0.08 | 8.04 | 83.35 | ×1.41 |
| | | | ST-ES (85%) | 63.83 | 0.06 | 8.36 | 72.25 | ×1.62 |

Table 11: Latency analysis of the Step-Tagging Early-Stopping criteria - For each datasets and models, we reported the average runtime per sample, across the 5 seeds. Each runtime $r$ is expressed in seconds. We reported the standard runtime $r_{standard}$ corresponding to the runtime during standard inference. The runtime $r_{ST\text{-}ES}$ is the sum of the runtime corresponding to stopped inference ($r_{stopped}$), the step classifier ($r_{step\ classifier}$), and the final completion ($r_{completion}$). The runtime Speed-up is computed between the $r_{ST\text{-}ES}$ and $r_{standard}$. For ST-ES (Router), we also added the average runtime of the BERT Router - 0.01s

**Latency analysis.** Table 11 reports the latency analysis of the experiments. The runtime analysis demonstrates a speed-up ranging from **1.09** to **2.07**, meaning that the framework leads to faster model inference for every configurations, compared to the standard inference. Importantly, the Table shows a *very low runtime of the step classifiers* (around 0.01s per annotation steps), being two magnitudes lower than the auto-regressive generation of the models. This is because we employed BERT classifiers. It is worth noting that the final completion have more impact on the overall runtime, but is still limited compared to the model's generation runtime since we only allowed a specific token budget (additional 100 tokens).

**Online latency.** Even though we showed that the Step-Tagging framework reduce the latency of the model's generation, we suspect that an online implementation of the framework could result in higher latency. Indeed, although the step classification itself is fast and does not hurt the efficiency gains, our framework requires to pause the model's generation frequently to perform step classification. To this means, this process can introduce additional latency as suggested by Yang et al. (2025). However, their findings suggest that this additional latency should be limited.

**Takeaway.** This section present a latency analysis of our Step-Tagging framework. The latency introduced by the classification process of the BERT models appears to be minimal compared to the generation's latency of the reasoning models. Future work should look at comparing runtime of offline and online implementations to validate our findings.

## J.2 TRAINING-INFERENCE COST TRADE-OFF

Our framework requires calibration methods to select the correct constraints, as well as training of BERT Step-Taggers to label the steps during LRMs generation. While this is a on-time exercise, these requirements implies additional computation. This section offers an analysis of the training-inference cost trade-off of our Step-Tagging Early-Stopping framework.

**Methodology.** The training cost of our Step-Tagging framework includes three components:

- **Training inference.** First, we first need to infer training samples of reasoning datasets. The resulting traces are used to train our BERT classifiers, as well as calibrating our framework (i.e. selecting the constraints).
- **Annotation cost.** Second, we need to annotate the reasoning traces to label our dataset used to train our BERT classifiers.
- **Training BERT Classifiers.** Finally, we need to train our BERT classifiers to then apply our framework on test samples.

Each steps implies additional costs. To allow a fair comparison with our gains obtained by our framework, we will quantify these using the saved-runtime compared to standard inference against the token-count at inference of our ST-ES framework. We focus on the DS-Qwen14B model, and assume that same analysis would scale on other models given that same token-count saving can be achieved.

**On the size of training datasets.** In our experiments on MATH500 and GSM8K, we deliberately used a larger training dataset than necessary for the Step-Tagging Early-Stopping framework to demonstrate its effectiveness (training datasets being twice the size of testing datasets). While this choice inflates the training cost, it ensures that the model performance is not limited by data availability. As shown in Appendix K, the framework can achieve comparable results using substantially smaller training datasets (10-50% of the test set size). Table 12 illustrates this.

| Dataset | Training Samples | Testing Samples | % Test | # Training Steps |
|---|---|---|---|---|
| Original MATH500 | 1,000 | 500 | 200% | 49,537 |
| Original GSM8K | 3,000 | 1,319 | 227% | 20,890 |
| MATH500 (*) | 150 | 500 | 25% | 5,865 |
| GSM8K (*) | 330 | 1,319 | 25% | 2,361 |
| AIME | 30 | 60 | 50% | 6,054 |
| GPQA | 40 | 158 | 25% | 5,373 |
| MMLU-Pro | 140 | 1,260 | 11% | 5,106 |

Table 12: Percentage and number of steps train data

To allow a fair comparison of training-inference costs across datasets, we introduced an additional training setting for MATH500 and GSM8K (annotated with *), where the size of the training dataset was reduced to same proportion as in the other datasets (25% of the test set). This setting enables us to compare the computational cost of our ST-ES framework under consistent conditions.

**MATH500 and GSM8K.** Figure 35 show the cumulative Saved Runtime of the ST-ES configurations obtained compared to the standard inference baseline, against the cumulative number of inference tokens on the test MATH500 and GSM8K datasets. First, we observe that the inference of the DS-Qwen14B model, as well as the anotation of the traces by `GPT-4o-mini` represent a significant amount of the cumulated runtime. In comparison, the training of Step-Taggers is of lower magintude (e.g. for MATH500 - 20 minutes of training runtime for BERT vs. 5.5 hours for the Training + annotation inference runtime, see lines in red). Furthermore, the training runtime of original datasets inflates the training runtime (reaching more than 30 and 60 hours on full training datasets for GSM8K and MATH500, respectively - see lines in gray).

When considering the smaller training datasets (* - see red lines), we observe that our ST-ES configurations almost recover the cost of training during inference compared to standard inference. Specifically, the ST-ES (85%) configurations lies above the red line after 800k and 400k inference tokens generated for the MATH500 and GSM8K datasets, respectively.

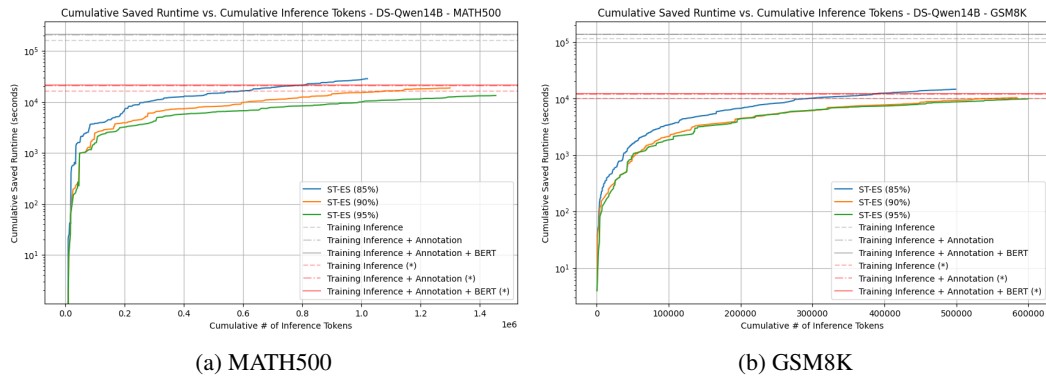

(a) MATH500        (b) GSM8K

Figure 35: Training-Inference cost trade-off

**AIME.** Similarly, our ST-ES framework lead to interesting training-inference trade-off on the AIME dataset. Indeed, the ST-ES (85%) configuration recovers all training cost by 300k inference token generated. We also observe that other configurations seems to scale well. Since training runtime are fixed (one time exercise), we expected consistent runtime saving when scaling the inference of our ST-ES configurations, leading to even more saved runtime.

**GPQA and MMLU-Pro.** Same observation can be made on the GPQA dataset, where the ST-ES (50%) configurations almost recover all training cost by 450k inference token generated. The GPQA dataset includes much more samples than AIME, explaining why the curves appears more stable. Interestingly, the ST-ES (50%) configurations on the MMLU-Pro dataset recovers the training cost at around 750k inference token generated, and leads to more than two fold training runtime saving at full test dataset inference. This is because we selected a training dataset of size being only 11% of the test dataset size, significantly decreasing the training runtime. It demonstrates that on low resources setting, selecting a small training dataset can reduce the training cost, while still leading to high efficiency gains and maintaining the accuracy.

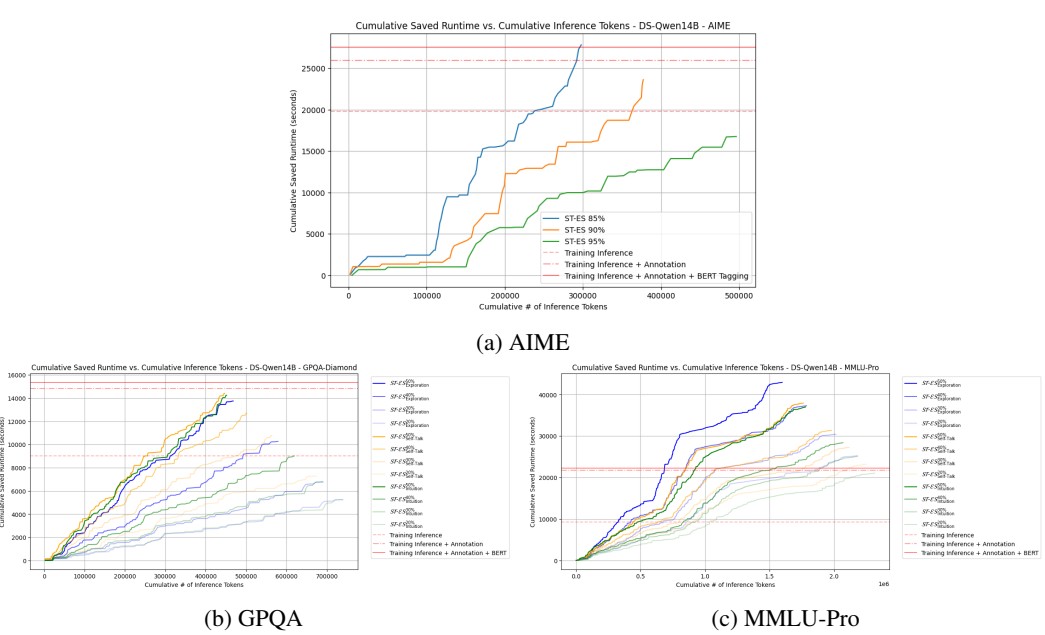

(a) AIME

(b) GPQA        (c) MMLU-Pro

Figure 36: Training-Inference cost trade-off

**Takeaways.** Overall, we demonstrated that our Step-Tagging Early-Stopping framework can recover its one-time training and calibration cost during inference, and continues to continues to deliver additional runtime saving compared to standard inference as more inference tokens are generated. The size of the training datasets mainly impact the trade-off. In particular, we showed that the training cost can be substantially reduced - by using smaller training datasets - while retaining the performance of our framework.

### J.3 IMPACT OF THE BERT-ROUTER PERFORMANCE ON THE ST-ES FRAMEWORK

The MATH500 dataset includes 5 levels of complexity and models needs more tokens to solve harder problems, leading to different optimal Step-Tagging Early-Stopping configurations. To demonstrate the flexibility of our framework in adapting the computation based on the needs of the user, we defined the ST-ES Router configuration. Specifically, we routed inference configuration (i.e. the constraint selection) of our ST-ES framework based on the complexity level of the problem. Our router is a binary BERT Classifier trained on the MATH500 questions, predicting easy (levels $\{1, 2\}$) or hard problems (levels $\{3, 4, 5\}$).

**Motivation.** Even though we obtained satisfying performances, it remains unclear how the performance of the BERT-Router affects the inference outcomes. To mitigate this limitation, this section looks at how routing errors affects the efficiency performance of our ST-ES framework.

**Claim.** We observed though our experimentation that more complex problems requires more computation (i.e. more token-count) to reach specific accuracy. Therefore, our objective by introducing the BERT-Router is to allocate more computation to more complex problems (levels $\{3, 4, 5\}$), and decrease the computation for easier questions (levels $\{1, 2\}$). Intuitively, an inaccurate router would cut the token-count too early for the complex problems, and therefore decrease their accuracy. Conversely, an inaccurate router would allocate too much computation for easy problems, and therefore vanish any efficiency gains from our Early-Stopping framework on these problems.

**Methodology.** To verify such claim, we selected the DS-Llama8B model, since we obtained the best results on the MATH500 dataset. As we conducted the same methodology on other models, we expect similar results for other models.

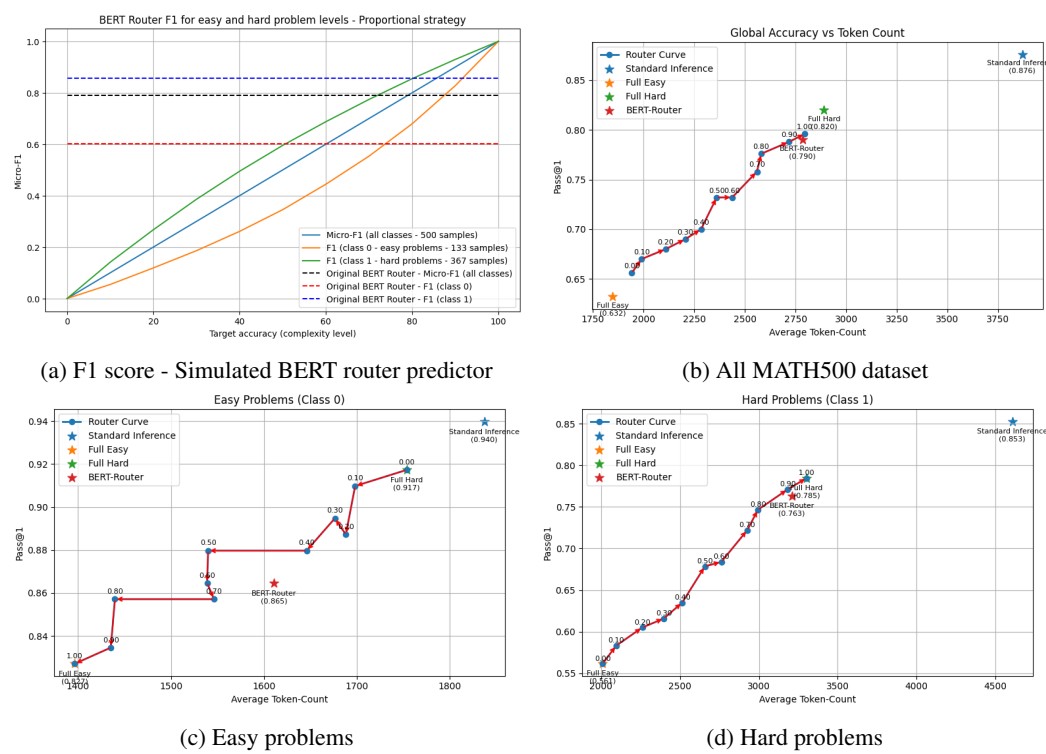

(a) F1 score - Simulated BERT router predictor

(b) All MATH500 dataset

(c) Easy problems

(d) Hard problems

Figure 37: Analysis of the BERT-Router error's propagation - DS-Llama8B (seed 42) - Router curves represents the different accuracy vs. token-count trade-off at different F1-scores of the simulated BERT-router performance. The F1-score of the BERT-router corresponds to the labels above the blue points. The red arrows represents the curves linking the blue points, as the BERT-router F1-score increases. The yellow and green stars represents the results of the ST-ES when easy and hard configuration are applied to all the points, respectively. The red star represent the performance of our original BERT-router.

**Proportional performance:** First, it is worth noting that the dataset split that we defined to train the BERT-Router on easy and hard problems is imbalanced (133 easy vs. 367 hard problems). To ground our analysis more realistically, we introduce a proportionally accurate router, that would get prediction correct following the dataset's distribution. Indeed, hard problems constitute roughly 73% of the dataset, so a balanced router would disproportionally impact the overall metric. We defined 11 target accuracy from the router, from 0.00 to 1.00 micro-F1, with a path of 0.10. Using the router's prediction, we applied our ST-ES framework to obtain tuples of (accuracy, token-count) for each configurations. Figure 37(a) describes the strategy we adopted, along with the original router accuracy (see Section P).

**Impact of the router on the ST-ES efficiency.** Figure 37 presents the impact of the BERT-Router performance on the model's inference. Specifically, we observe how gradually more accurate router leads to more efficient inference.

**Easy problems.** Figure 37(b) shows the results restraint to easy problems. First, we observe that the accuracy as well as the token-count gradually decreases as the router becomes more accurate. Indeed, when the router is inaccurate (i.e. Full Hard configuration and F1 on class 0 is 0.00), the router would allocate a too much computation for an easy problem. As the router becomes more accurate, the router selects more appropriate configuration for easy problems, therefore reducing the overall inference cost for easy problems. Our original BERT-router lies at the same Pass@1 level of the simulated F1-score of 0.6, but have a slightly higher average token-count. This makes sense with our results: on simple problems, the BERT-router obtained an F1-score of 0.6. However, as it is slighly over-predicting class 1, it tends to allocate Hard configuration on some easy problems, increasing its overall computations.

**Hard problems.** Conversely, Figure 37(c) presents the results restraint to hard problems. In this case, we observe the opposite phenomenon. Inaccurate routers allocates small budget configurations, drastically decreasing the accuracy of the model on hard problems (more than 0.3 accuracy drop for the Full Easy configuration). As the router becomes more accurate, it allows more computation by allocating the Hard configuration to hard problems. Similarly, our original BERT-Router obtained scores close to Full Hard, but tends to allocate easy configuration to some hard samples.

**Overall dataset.** Figure 37(d) shows the results over the full datasets. We observe similar trends as the Hard problem (Figure 37(c)). This is due for two reasons. First, the hard problems are more frequent in the dataset (around 73% of the samples), so it explains why the trends are more similar to this configuration. Second, the different of average token-count between extreme Router performance (0.0 vs. 1.0 micro-F1) is much higher on harder problems (around 500 tokens for easy problems, vs. 1,500 tokens for harder problems). Therefore, the saved tokens from the good selection of easy problems are overridden by the additional budget required to solve harder problems.

**Takeaways.** Overall, we confirmed our claims: The performance of the router has a significant impact on the efficiency of our ST-ES inference. Router errors have asymmetric impact: over-allocation of computation on easy problems mainly hurts efficiency, while under-allocation of computation on hard problems hurts both accuracy and efficiency. Good selection of the configuration given the difficulty of the problems could unveiled higher efficiency gains. To further confirm our findings, it would be interesting to compare our results with a dataset including much more easy problems.

## K    FURTHER VALIDATING THE STEP-TAGGING EARLY-STOPPING FRAMEWORK

**Objective.** In the main body of the paper, we tested our framework by applying it on 2 renown mathematical datasets, namely GSM8K and MATH500, and on three popular LRMs. To further validate our framework, this section seek to report additional experiments lead on three additional types of dataset: AIME, GPQA-Diamond, and MMLU-Pro (presented in Table 13).

**Generalization to harder tasks.** First, we observe that GSM8K and MATH500 are two datasets that seems to be saturated. Indeed, we obtained Avg@5 ranging from 0.83 to 0.95, meaning that models are capable of solving almost every samples, even though needing a significant amount of resources. To address this phenomenon, we introduce the AIME (American Invitational Mathematics Examination) dataset, often considered as more challenging compared to the two later, which could be explained by the nature of the datasets. Indeed, Sun et al. (2025) compares existing benchmarks, and reveals that GSM8K, MATH500, and AIME datasets are of increasing level of difficulty, namely Grade School, Competition, and Olympiad, respectively.

**Generalization to other reasoning tasks.** Secondly, our main analysis focuses on problem solving of Mathematical questions. To observe the applicability of our framework to other tasks, we introduce GPQA-Diamond and MMLU-Pro, two datasets containing different types of questions. First, GPQA-Diamond contains 198 questions about Biology, Physics and Chemistry wrote by PhD students. Second, MMLU-Pro includes 12,000 samples from 14 different fields, such as Business, Philosophy, Computer Science, or History.

| Dataset | Ref. | Type | # Full | # Train used | # Test |
|---|---|---|---|---|---|
| AIME (22-24) | (Project-Numina, 2025) | Maths Olympiad | 90 | 30 | 60 |
| GPQA-Diamond | Rein et al. (2023) | Science (PhD-level) | 198 | 40 | 158 |
| MMLU-Pro | Wang et al. (2024) | Multi-Task academic | 12,000 | 140 | 1,260 |

Table 13: Description of additional datasets - **AIME:** we selected AIME-22 as training and calibration dataset, and AIME-23 as well as AIME-24 as test datasets. **GPQA-Diamond:** we selected 20% of the dataset as training, stratified per category of problems. **MMLU-Pro:** We sampled uniformly 200 samples per categories of the original dataset to construct our dataset, and selected 50% of the dataset as training and calibration samples, stratified per category (i.e. 50 samples per category).

**Claims.** By applying our framework to these datasets, we seek to validate the following claims:

- **Generalization of the framework:** A satisfying performance of our framework on harder mathematics tasks, as well as on different domains would validate the good generalization of our framework and early-stopping criteria.
- **Validation of our taxonomy:** While our taxonomy has been derived from relatively easy mathematical examples, a good performance of the step-taggers would further validate our ReasonType taxonomy in the sense that classes of the taxonomy are generalizable to harder mathematical problems, as well as other tasks than mathematical questions.

**Methodology.** To proceed, we applied our framework, as well as the baseline detailed in Section 5, on the DS-Qwen-14B model (since the model obtained less pronounced results). We ran a single inference, with deterministic decoding and a seed of 42. Results report the Pass@1 metric, and estimated runtime for the Step-Tagging configurations (as per Appendix J.1).

**Evaluation strategies.** In Section 5, we mentioned that we relied on the Math-Verify evaluation tool to assess the model's performance on Mathematical questions. For the AIME (22-24) dataset, it applies well since the questions are still in the math domain. However, for GPQA-Diamond and MMLU-Pro, the questions are more general and often recall knowledge. Therefore, Math-Verify was not applicable. For this reason, we followed the author's guidelines of each dataset. Specifically, each sample is composed of a question including multiple possible answer (formulated as a choice of letter), and we prompted the model to deliver the best one. We then created an evaluation strategy leveraging regex parsing to obtain the solution letter from the models, as suggested by the authors of the dataset (see Appendix A.3.1. (Rein et al., 2023)).

### K.1 On harder mathematical tasks

To determine the constraints, we first conducted a calibration study on the AIME-22 dataset. Since AIME is an Olympiad conducted over the three different years (22-24), we assumed that the difficulty and type of questions are equivalent over the three years. Therefore, we are looking to get the constraints from the 2022 version to then apply them on the AIME-23 and AIME-24 datasets.

**Calibration.** As in Section 5.3, Figure 38 presents the Pass@1 vs. Token-Count for the step types $\tau$ of the taxonomy, for value of $\delta \in [1, 50]$. First, we note that the model outputs are significantly larger, being almost 3 times the one obtained on the MATH500 dataset (12k vs. 4k). In addition, even though the dataset is small (only 30 samples), we observe different efficiency trade-offs depending on the step types. Specifically, we can see that the *Intuition* and *Exploration* step-types lie closely to the Pareto front, especially from 6,000 token count.

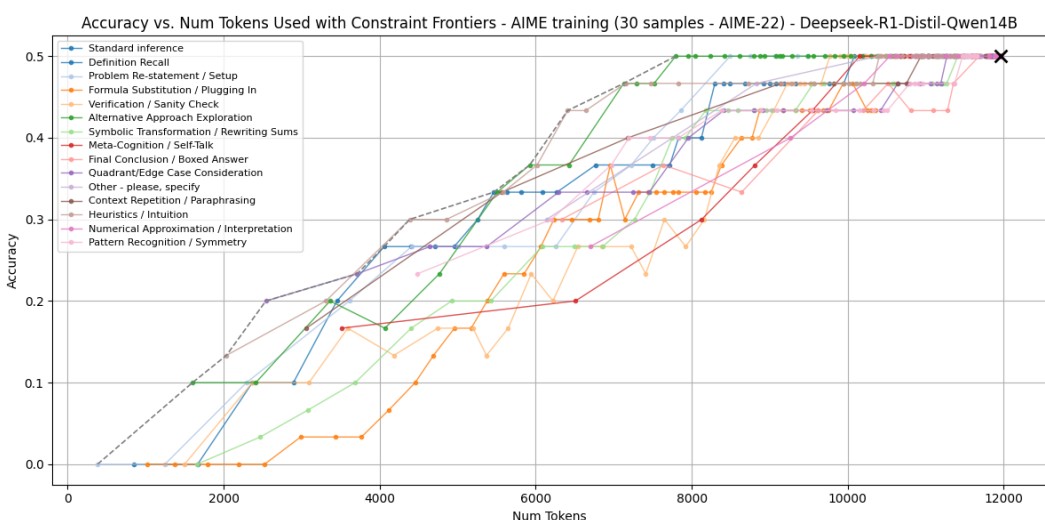

Figure 38: Early-Stopping selection using Pareto - DS-Qwen14B on AIME-22 - seed $42$

Furthermore, we also observe that the constraint curves for each step-types converges much quicker to the accuracy of the standard run (highlighted in black cross), compared to the other datasets (see Section O). This is an important observation because it would means that our framework scales well on harder dataset (i.e. more verbose generation), giving even more opportunity for efficient analysis.

We suspect that this is due for two reasons. First, the outputs are much larger, giving more opportunity for the model to back-track, and over-verify since the model generates more steps. Second, the dataset includes a limited number of samples, so this observation might not scale well with more samples and the accuracy is sensible to the number of samples (each sample contributes highly to the final accuracy).

**Constraint selection.** From Figure 38, we selected 3 constraints, reported in Table 14.

| AIME (22-24) | ST-ES 85% | ST-ES 90% | ST-ES 95% |
|---|---|---|---|
| $\{\tau, \delta\}$ | Intuition / 6 | Exploration / 16 | Problem Re-Statement / 25 |

Table 14: Selected constraints for DS-Qwen14B on the AIME dataset

**Performance of the Step-Tagging classifiers.** To implement our framework to the AIME-23 and AIME-24 datasets, we trained BERT classifiers on the selected step-types (see Table 14). To further compare the performance obtained with the other datasets, we included two additional step-types, namely *Self-Talk* and *Verification*. Figure 39 present the Micro and Macro F1 for the AIME-23 and AIME-24 datasets. Even though the training dataset was much smaller (around 5k steps from only 30 samples of AIME-22 vs. $\approx$40k for MATH500 and GSM8K), we still obtained satisfying results. It confirms that our taxonomy is transferable to other datasets, and that not much training data is needed to obtain performant classifiers.

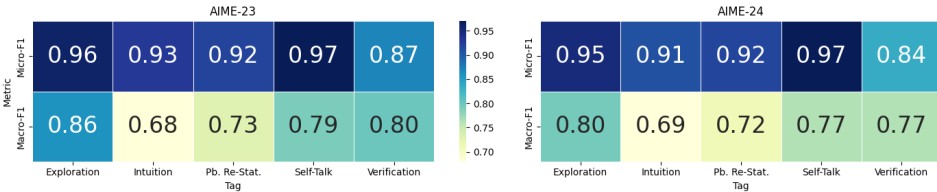

Figure 39: Performance of the Step-Taggers on AIME-23 and AIME-24 - DS-Qwen14B - seed 42

**Evaluation of the Step-Tagging Early-Stopping framework.** After training our step-taggers, we applied our framework to AIME-23 and AIME-24. Figure 40 present the average token count against the Avg@1 for the DS-Qwen14B model on the AIME-23 and AIME-24 datasets. Each plot compares the performance trade-offs between the baselines and the ST-ES criteria. Table 15 contains the detailed metrics to compare each approach, including the token-count, the runtime and the Pass@1.

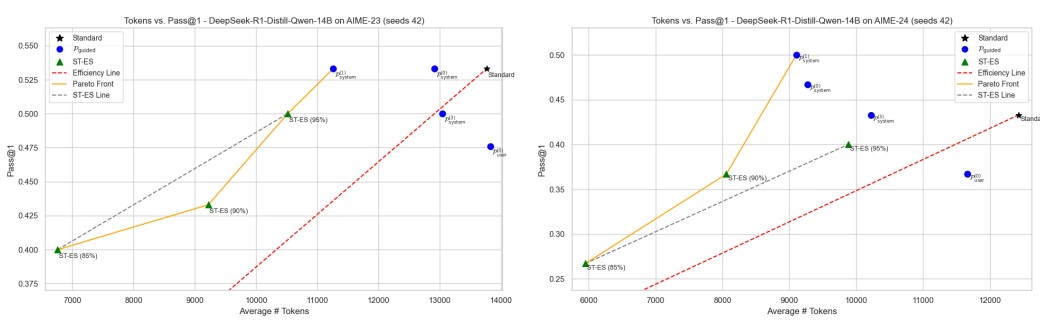

| (a) DS-Qwen14B on AIME-23 | (b) DS-Qwen14B on AIME-24 |

Figure 40: Number of Tokens vs. Pass@1 - $\mathcal{P}_{\text{guided}}$ Baselines and ST-ES criteria

| Config. | AIME-23 | | | | | AIME-24 | | | | |
|---------|---------|-----------|---------|---------------|--------|---------|-----------|---------|---------------|--------|
| | # Tokens | Saved (%) | Runtime | Speed-up (↑) | Pass@1 | # Tokens | Saved (%) | Runtime | Speed-up (↑) | Pass@1 |
| Standard | 13765.33 | - | 781.30 | - | 0.533 | 12427.33 | - | 683.29 | - | 0.433 |
| Basel. $\mathcal{IES}$ | 12399.70 | 9.92 | 714.28 | 1.09 | 0.533 | 11231.07 | 9.62 | 624.70 | 1.09 | 0.500 |
| Basel. $\mathcal{P}_{\text{user}}^{(0)}$ | 13832.03 | -0.48 | 810.92 | 0.96 | 0.467 | 11664.77 | 6.13 | 680.72 | 1.00 | 0.367 |
| Basel. $\mathcal{P}_{\text{system}}^{(0)}$ | 12919.23 | 6.15 | 723.38 | 1.08 | 0.533 | 9272.7 | 25.38 | 455.59 | 1.49 | 0.467 |
| Basel. $\mathcal{P}_{\text{system}}^{(1)}$ | 11257.23 | 18.22 | 602.21 | 1.29 | 0.533 | 9105.2 | 26.73 | 483.04 | 1.41 | 0.500 |
| Basel. $\mathcal{P}_{\text{system}}^{(3)}$ | 13046.93 | 5.22 | 756.71 | 1.03 | 0.500 | 10224.43 | 17.73 | 551.29 | 1.24 | 0.433 |
| ST-ES (95%) | 10515.7 | 23.61 | 587.53 | 1.33 | 0.500 | 9880.3 | 20.49 | 530.82 | 1.29 | 0.400 |
| ST-ES (90%) | 9218.20 | 33.03 | 515.78 | 1.51 | 0.433 | 8058.67 | 35.15 | 425.90 | 1.60 | 0.367 |
| ST-ES (85%) | 6757.03 | 50.91 | 376.89 | 2.07 | 0.400 | 5949.73 | 52.12 | 309.86 | 2.21 | 0.267 |

Table 15: Performance of Step-Tagging Early stopping - DS-Qwen14B - seed 42

**Generalization to harder mathematical tasks.** We observe more nuanced results on the AIME dataset than on MATH500 for the DS-Qwen14B model. First, the $\mathcal{P}_{\text{guided}}$ baselines appears much weaker on this dataset, with limited token-count savings (up to 26% over the two datasets). In comparison, the Step-Tagging approaches lead to more efficient generation, with token-count saving from 20% to 52%. Specifically, Figure 40 shows the ST-ES 95% on AIME-23 leading to same accuracy as $\mathcal{P}_{\text{system}}^{(3)}$ (0.500), for a much lower token-count (9.2k vs. 13k). Similarly, the ST-ES 90% on AIME-24 obtained same accuracy as $\mathcal{P}_{\text{user}}^{(0)}$ (0.367), with higher token-count savings (35.15% vs. 6.13%). For higher token-count saving, it is worth noting that the accuracy drop of the ST-ES methods is slighlty higher than other datasets (-13% and -16% for ST-ES 85% on AIME-23 and AIME-24, respectively). This can be explained by the nature of the dataset. AIME only contains 30 samples, so the correctness of each sample highly contributes to the overall accuracy.

**Takeaway.** Overall, our results confirms the applicability of our framework to harder mathematical datasets. Harder tasks such as AIME lead to the generation of much more tokens ($\approx 12k$), therefore this finding is important as it could results in higher token-count saving.

## K.2 BEYOND MATHEMATICAL TASKS

**Calibration.** Compared to the mathematical domain, GPQA and MMLU-Pro datasets requires knowledge and advanced reasoning in other fields such as Biology or Physics. Therefore, the evaluation strategy of the outputs of models is less straightforward since the usage of advanced evaluation libraries such as Math-Verify is not possible. For this reason, the authors of the benchmarks specified possible solutions in the prompt of the models, and asked the model to select the correct solution.

Even though this method allowed the users to reach the correct performance of the model by parsing the final solution at the end of the inference, it is hardly applicable for early-stopping algorithms since it is hard to parse the current solution of the problem due to formatting issues. Indeed, the model might generate the correct solution inside its reasoning trace, but sometimes does not mention the answer letter, or does not write the correct answer in the exact same format as instructed.

For this reason, our $\mathcal{IES}$ baseline was not applicable, and we re-framed our calibration study to take this limitation into account. Instead of computing the Pareto curve using the early-stopped accuracy and token-count, we plot the token-count against the different values of $\delta$ for each step-types. We then selected different step-types $\tau$ presenting different behaviors and impact on the model, and *targeted specific token-count saving* instead of expected accuracy drop. Once the configurations selected, we then prompted back the models to express their current best answer (see Section 5).

**Calibration for GPQA-Diamond.** Figure 41 shows our adapted calibration process for the GPQA-Diamond train dataset for the DS-Qwen14B model. First, we observe different profiles for each step-type curves. This is linked to the frequency of each step-types in the training inference. To compare the Step-Tagging early-stopping performance of different curve profiles, we selected three step-types, namely *Self-Talk*, *Exploration*, and *Intuition*. As well as being used as constraints for other datasets (see Section O), Figure 41 shows different convergence rate for these step-types. For these three different step-types, we selected constraints matching expected token-count saving, namely 50%, 40%, 30%, and 20% compared to standard inference. We refer to these configuration as ST-ES A, B, C and D, respectively.

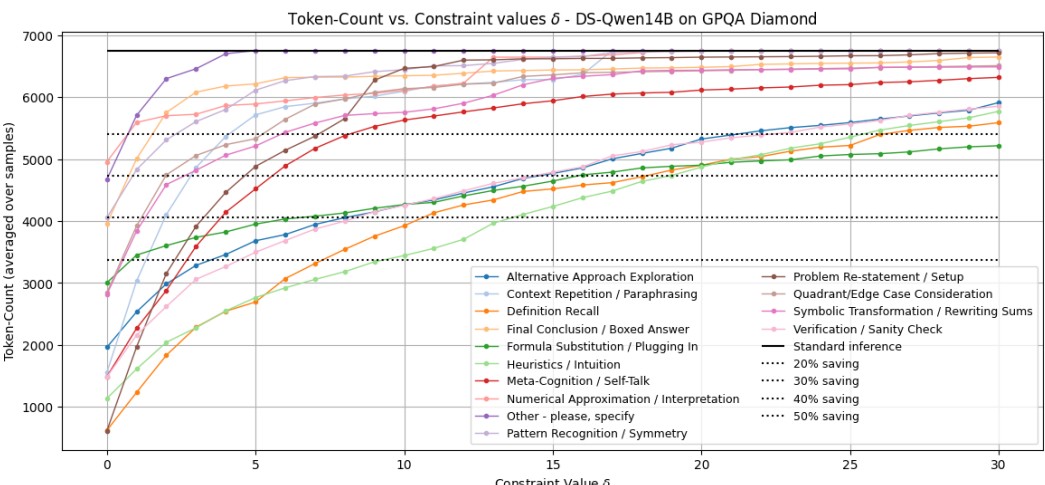

Figure 41: Early-Stopping selection using Token-Count - DS-Qwen14B on GPQA - seed 42

**Constraint selection.** From Figure 41, we show the constraints for the three step-types in Table 16.

| **Datasets** | $\mathcal{ST}\text{-}\mathcal{ES}_\tau^A$ **(50%)** | $\mathcal{ST}\text{-}\mathcal{ES}_\tau^B$ **(40%)** | $\mathcal{ST}\text{-}\mathcal{ES}_\tau^C$ **(30%)** | $\mathcal{ST}\text{-}\mathcal{ES}_\tau^D$ **(20%)** |
|---|---|---|---|---|
| **GPQA-Diamond** | Self-Talk / 3 Exploration / 4 Intuition / 6 | Self-Talk / 4 Exploration / 8 Intuition / 14 | Self-Talk / 6 Exploration / 15 Intuition / 20 | Self-Talk / 8 Exploration / 20 Intuition / 25 |

Table 16: Selected constraints for DS-Qwen14B on the GPQA-Diamond dataset

**Calibration for MMLU-Pro.** Similarly, Figure 42 shows our adapted calibration process on the MMLU-Pro train dataset for the DS-Qwen14B model. We are also observing that the token-count trade-off depends on the step-types. It is worth noting that the average standard token-count is lower for this dataset (around 7k). Moreover, the curves converge towards the standard inference more quickly than for GPQA-Diamond. In fact, all curves reach at least 50% of the token count for $\delta \geq 5$, compared to $\delta \geq 10$ for the GPQA dataset. For this reason, we selected lower expected token-count saving, namely 35%, 30%, 25%, and 20% compared to standard inference, and we selected same step-types as per GPQA-Diamond. We refer to these configuration as ST-ES A, B, C and D, respectively.

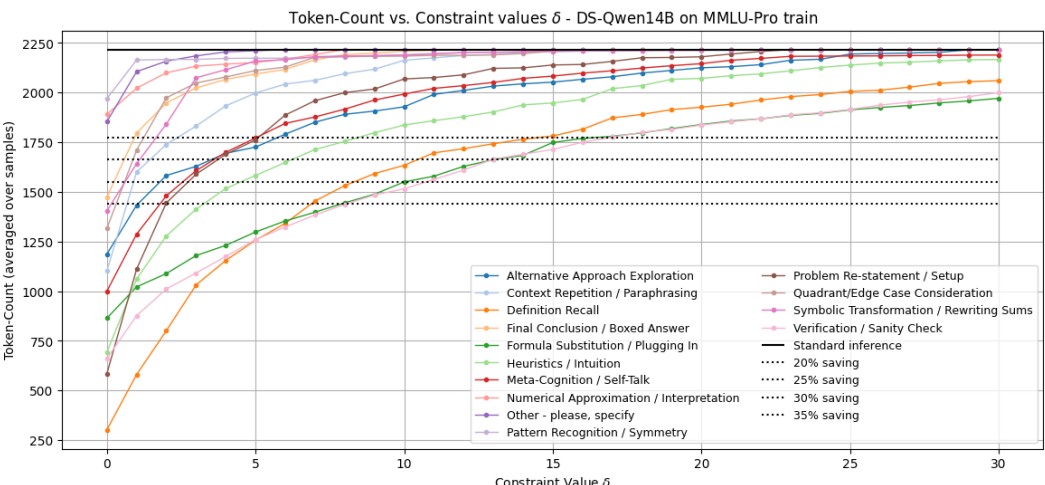

Figure 42: Early-Stopping selection using Token-Count - DS-Qwen14B on MMLU-Pro - seed 42

**Constraint selection.** From Figure 42, we show the constraints for the three step-types in Table 17.

| **Datasets** | $\mathcal{ST}\text{-}\mathcal{ES}_\tau^A$ **(35%)** | $\mathcal{ST}\text{-}\mathcal{ES}_\tau^B$ **(30%)** | $\mathcal{ST}\text{-}\mathcal{ES}_\tau^C$ **(25%)** | $\mathcal{ST}\text{-}\mathcal{ES}_\tau^D$ **(20%)** |
|---|---|---|---|---|
| **MMLU-Pro** | Self-Talk / 2 | Self-Talk / 3 | Self-Talk / 4 | Self-Talk / 5 |
| | Exploration / 1 | Exploration / 2 | Exploration / 4 | Exploration / 6 |
| | Intuition / 3 | Intuition / 5 | Intuition / 6 | Intuition / 9 |

Table 17: Selected constraints for DS-Qwen14B on the MMLU-Pro dataset

**Performance of the Step-Tagging classifiers.** Once we selected our constraints, we trained our step-taggers on the training datasets. Figure 43 presents the Micro and Macro-F1 for the selected step-types to validate the generalization of our taxonomy. We observe satisfying performance of the step-taggers, specifically for *Exploration* and *Self-Talk* classes. Since the size of the training datasets are limited ($\approx$ 4-5k), we suspect that larger dataset could result in more accurate classifiers. The good performance of the classifiers suggests that our ReasonType taxonomy, coupled with our annotation process, generalizes well to non-mathematical tasks.

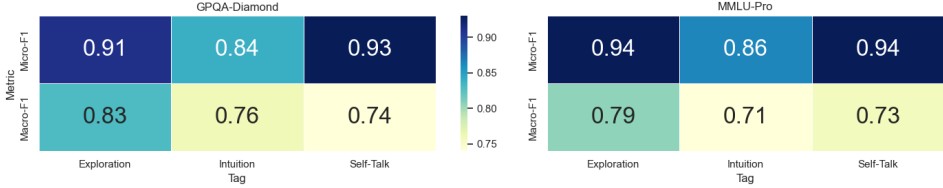

Figure 43: Performance of the Step-Taggers on GPQA and MMLU-Pro - DS-Qwen14B - seed 42

**Evaluation of the Step-Tagging Early-Stopping framework.** Figure 44 present the average token count against the Pass@1 for the DS-Qwen14B model on the GPQA-Diamond and MMLU-Pro datasets. Each plot compares the performance trade-offs between the baselines and the ST-ES criteria. Table 18 contains the detailed metrics to compare each approach.

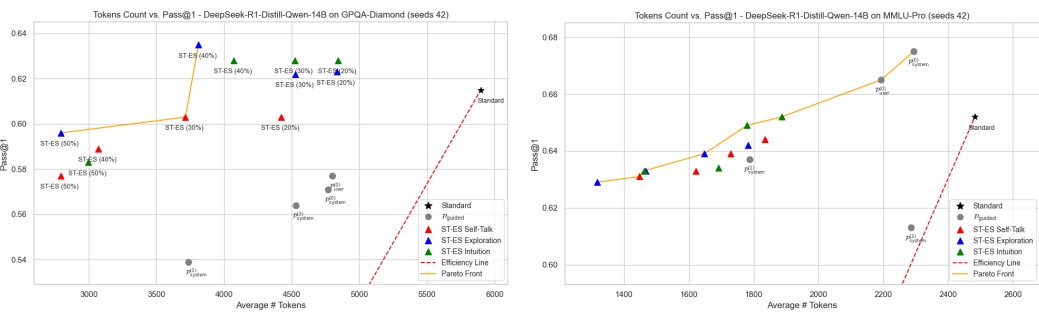

(a) DS-Qwen14B on GPQA-Diamond  (b) DS-Qwen14B on MMLU-Pro

Figure 44: Number of Tokens vs. Pass@1 - $\mathcal{P}_{\text{guided}}$ Baselines and ST-ES criteria

**Generalization to broader reasoning tasks.** First, we observe that all ST-ES criteria and baselines configurations lies above the efficiency line in red, meaning that all achieved more efficient generation than standard inference. Regarding the GPQA-Diamond dataset, the ST-ES obtained strong performance, with configurations forming a Pareto front. Specifically, the $\mathcal{ST}\text{-}\mathcal{ES}_{\tau}^{A}$ configurations reached around 50% token-count saving with minimal Pass@1 drop compared to the standard inference (from $-1.9\%$ to $-3.8\%$ depending on the step-type). Furthermore, the ST-ES obtained stronger results than $\mathcal{P}_{\text{guided}}$ baselines. Indeed, $\mathcal{ST}\text{-}\mathcal{ES}_{Self\text{-}Talk}^{C}$ and $\mathcal{ST}\text{-}\mathcal{ES}_{Exploration}^{B}$ obtained same token-count saving than $\mathcal{P}_{\text{system}}^{(1)}$ ($\approx 35\%$) while achieving higher accuracy (0.603 and 0.635 vs. 0.571, respectively). Similar observation can be done for the configurations C and D of step-types Exploration and Intuition compared to the three other baselines.

Our framework also obtained strong results on the MMLU-Pro dataset. Figure 44 shows that most ST-ES configurations lie on the Pareto front. The early-stopping criteria also generally obtained higher token-count saving than the $\mathcal{P}_{\text{guided}}$ baselines, with from 24% to 45% token-count reduction and minimal accuracy drop (less than $-2.3\%$ compared to standard inference).

| Config. | GPQA-Diamond | | | | | MMLU-Pro | | | | |
|---|---|---|---|---|---|---|---|---|---|---|
| | # Tokens | Saved (%) | Runtime | Speed-up (↑) | Pass@1 | # Tokens | Saved (%) | Runtime | Speed-up (↑) | Pass@1 |
| Standard | 5897.87 | - | 183.28 | - | 0.615 | 2501.93 | - | 72.44 | - | 0.652 |
| Basel. $\mathcal{P}_{user}^{(0)}$ | 4801.04 | 18.59 | 154.78 | 1.18 | 0.577 | 2194.10 | 12.30 | 63.75 | 1.14 | 0.665 |
| Basel. $\mathcal{P}_{system}^{(0)}$ | 4768.30 | 19.15 | 159.42 | 1.15 | 0.571 | 2292.93 | 8.35 | 64.57 | 1.12 | 0.675 |
| Basel. $\mathcal{P}_{system}^{(1)}$ | 3736.18 | 36.65 | 133.68 | 1.37 | 0.539 | 1705.59 | 31.83 | 59.86 | 1.21 | 0.637 |
| Basel. $\mathcal{P}_{system}^{(3)}$ | 4530.14 | 23.19 | 158.56 | 1.16 | 0.564 | 1935.21 | 22.65 | 61.69 | 1.17 | 0.613 |
| $\mathcal{ST}\text{-}\mathcal{ES}_{Self\text{-}Talk}^{D}$ | 4423.66 | 24.99 | 128.49 | 1.43 | 0.603 | 1835.21 | 26.64 | 54.28 | 1.33 | 0.644 |
| $\mathcal{ST}\text{-}\mathcal{ES}_{Self\text{-}Talk}^{C}$ | 3714.10 | 37.02 | 107.89 | 1.69 | 0.603 | 1729.70 | 30.32 | 51.24 | 1.41 | 0.639 |
| $\mathcal{ST}\text{-}\mathcal{ES}_{Self\text{-}Talk}^{B}$ | 3071.25 | 47.93 | 96.39 | 1.90 | 0.589 | 1621.66 | 35.18 | 48.14 | 1.50 | 0.633 |
| $\mathcal{ST}\text{-}\mathcal{ES}_{Self\text{-}Talk}^{A}$ | 2793.22 | 52.64 | 86.24 | 2.13 | 0.577 | 1447.69 | 42.14 | 43.18 | 1.68 | 0.631 |
| $\mathcal{ST}\text{-}\mathcal{ES}_{Exploration}^{D}$ | 4834.89 | 18.02 | 140.35 | 1.31 | 0.623 | 1782.79 | 28.74 | 52.29 | 1.39 | 0.642 |
| $\mathcal{ST}\text{-}\mathcal{ES}_{Exploration}^{C}$ | 4528.29 | 23.22 | 131.45 | 1.39 | 0.622 | 1647.88 | 34.14 | 48.41 | 1.49 | 0.639 |
| $\mathcal{ST}\text{-}\mathcal{ES}_{Exploration}^{B}$ | 3811.49 | 35.37 | 110.64 | 1.66 | 0.635 | 1467.91 | 41.33 | 43.22 | 1.68 | 0.633 |
| $\mathcal{ST}\text{-}\mathcal{ES}_{Exploration}^{A}$ | 3101.65 | 47.41 | 90.14 | 2.03 | 0.596 | 1318.09 | 47.32 | 38.94 | 1.86 | 0.629 |
| $\mathcal{ST}\text{-}\mathcal{ES}_{Intuition}^{D}$ | 4844.16 | 17.87 | 140.49 | 1.30 | 0.628 | 1886.90 | 23.99 | 55.91 | 1.29 | 0.652 |
| $\mathcal{ST}\text{-}\mathcal{ES}_{Intuition}^{C}$ | 4521.56 | 23.33 | 131.18 | 1.39 | 0.628 | 1779.91 | 28.29 | 52.91 | 1.37 | 0.649 |
| $\mathcal{ST}\text{-}\mathcal{ES}_{Intuition}^{B}$ | 4071.41 | 30.97 | 118.13 | 1.55 | 0.628 | 1692.05 | 31.84 | 50.44 | 1.44 | 0.634 |
| $\mathcal{ST}\text{-}\mathcal{ES}_{Intuition}^{A}$ | 2996.83 | 49.19 | 87.02 | 2.11 | 0.583 | 1461.90 | 41.11 | 43.93 | 1.65 | 0.633 |

Table 18: Performance of Step-Tagging Early stopping - DS-Qwen14B - seed 42

**Takeaways.** Our Step-Tagging framework generalizes well to non-mathematical task, with satisfying performance of the step-taggers and strong efficiency gains from our early-stopping criteria on both GPQA-Diamond and MMLU-Pro datasets.

## L WHY ST-ES YIELDS TO SMALLER GAINS ON QwQ-32B: EVIDENCE OF MORE CONSERVATIVE REASONING

In the Section 7, we highlighted that our ST-ES framework yields smaller efficiency gains on the QwQ-32B compared to the Deepseek models. Indeed, baselines appears stronger relatively to the ST-ES criteria, and accuracy drop is higher. We suggested two arguments to explain this phenomenon. This section will aim to provide additional materials supporting our claims.

**Objective.** One of our hypothesis was that QwQ-32B is less destructive in its way of generating answers. We observe, using our $\mathcal{IES}$ baseline as well as through the literature, that LRMs tends to formulate their answer early in their thinking process. However, we also observe that some models appears to be destructive: even after achieving to get the answer correct during intermediate steps, they continue generating reasoning that later overwrites or contradicts this earlier correct answer expressed (Muennighoff et al., 2025).

This destructive behavior is precisely what created opportunities for early-exit: if the correct answer temporarily appears and is later replaced by a wrong one, early stopping can prevents the overthinking behavior and preserve the accuracy of models. If QwQ-32B is more conservative, it tends to maintains the same answer throughout the reasoning process, and therefore would benefits less of early-stopping criteria.

**Methodology.** To conduct such analysis, we need to compare the accuracy of the reasoning traces at different checkpoints along each model's reasoning trace. For each samples, we:

1. Prune the reasoning trace to a fixed percentage (e.g. 10%, 20%, . . . , 90%) of its full length.
2. Evaluate whether this truncated trace already contains a correct answer.
3. Filter the reasoning traces to retain the samples that are correct at this checkpoint.
4. Iteratively continue the trace from the checkpoint to 100% and check whether the final output preserves or overwrites the earlier correct answer.

Specifically, we measure how frequently a model produces a correct intermediate answer, and subsequently destroys it later in the reasoning. To visualize this, we use heatmaps. The y-axis indicates the checkpoint used for filtering - at a given percentage, we select all samples containing a correct answer. The x-axis represents the evaluation checkpoint - for the same set of filtered samples, we evaluate the correctness of their reasoning when extended to a later percentage of the total trace.

**Deepseek models are destructive.** Figure 45 shows the resulting heatmaps for the three reasoning models (DS-Llama8B, DS-Qwen14B and QwQ-32B) on both MATH500 and GSM8K datasets. By construction, cells in the diagonals all have an accuracy of 1.0. In fact, this is because we filtered early-exit answers at a specific token-count percentage. Importantly, we observe darker areas above the diagonal for the Deepseek models. We interpret this observation as the Deepseek models more frequently overwriting their current answer. The darker areas represents the accuracy drops of the same filtered samples when adding more token-count. It means that when adding more thinking, the model either changes its answer, or does not express it anymore. This phenomenon is even more clearly visible on the GSM8K dataset.

**QwQ-32B is more conservative than Deepseek models.** Comparing QwQ-32B to the Deepseek models, we observe that the areas above the diagonal are less dark, meaning that the model more frequently retains its original correct answer, and QwQ-32B is therefore less destructive (i.e. more conservative). As well, the evaluation of the filtered early-exit reasoning traces for the full reasoning traces (evaluation at 100% - last column of the heatmap) leads to lower accuracy for the Deepseek models than for QwQ-32B. It means that for the filtered early-exit answers, the answers that were right earlier stays until the end of the reasoning more frequently for the QwQ than for the Deepseek models. This observation aligns with results obtained in Figure 7 in Section 7.

**Takeaways.** Our analysis confirms our hypothesis: QwQ-32B exhibits a more conservative reasoning process compared to the Deepseek models. Because our ST-ES framework aims to captures the correct intermediate answers before they are overwritten, a model that rarely overwrites its early answers leaves less opportunities for early-stopping intervention. This explains why ST-ES yiels smaller gains on QwQ-32B. In addition, we observe that QwQ-32B shows larger accuracy drops under our early-stopping criteria. This is because some samples may requires longer reasoning for this model, and early-stopping these traces prevents the model from reaching the correct final answer.

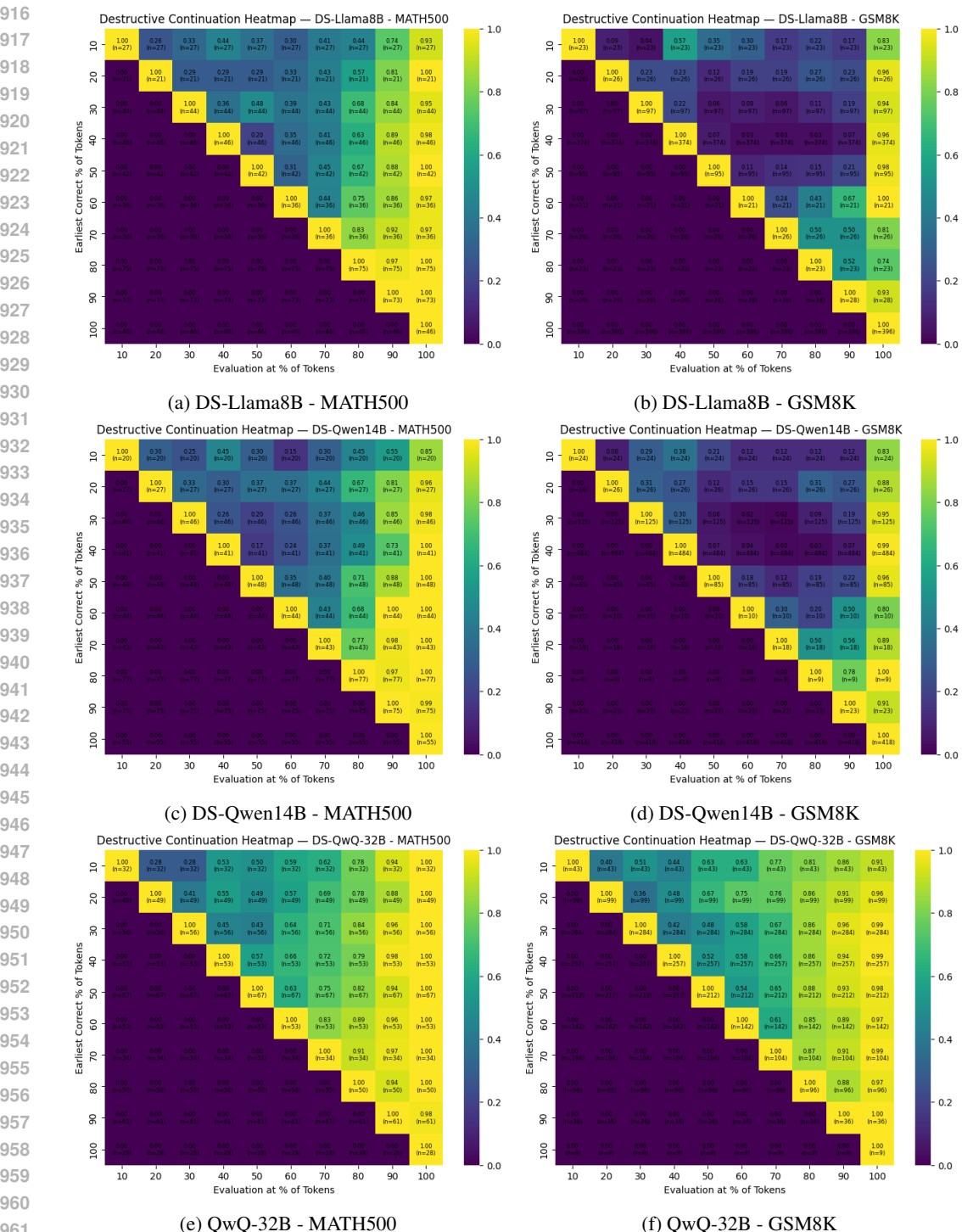

(a) DS-Llama8B - MATH500

(b) DS-Llama8B - GSM8K

(c) DS-Qwen14B - MATH500

(d) DS-Qwen14B - GSM8K

(e) QwQ-32B - MATH500

(f) QwQ-32B - GSM8K

Figure 45: Destructive reasoning continuation - DS-Llama8B, DS-Qwen14B and QwQ-32B on MATH500 and GSM8K - seed $42$ - The y-axis indicates the checkpoint used for filtering - at a given percentage of truncation, we select all samples containing a correct answer. The x-axis represents the evaluation checkpoint - for the same set of filtered samples, we evaluate the correctness of their reasoning when extending their generation to a higher percentage of truncation.

# M    ANALYSIS OF REASONING STEP TYPES

## M.1    PROMPTS

Figures 46 and 47 show the two prompts used on `GPT-4o-mini` to build the taxonomy and generate the step tags on the traces of the LRMs, respectively.

---

**Prompt Taxonomy**

Below is a reasoning trace of a reasoning language model, split by steps. In these examples, can you please identify the different type of steps? Suggest some reasoning-type labels for each of them.
- Step 1: {step_1}
- **[...]**
- Step t: {step_t}

---

Figure 46: Prompt used to generate the Taxonomy

---

**Prompt Taxonomy**

input=[ "role": "system", "content": "Classify the following reasoning step into one of the categories defined. Classes = {taxonomy}", "role": "user", "content": step ],

---

Figure 47: Prompt used to monitor the steps

Section M contains an additional analysis of the reasoning steps and tags issues to complete our work.

## M.2    STATISTICS ON THE REASONING STEPS COUNT AND TYPES

Table 19 presents statistics on the number of steps and `GPT-4o-mini` annotation for each models on both datasets we selected. Results are averaged for the seed 42 on test datasets.

| Dataset | Model | # Tok. / Steps | # Steps | Runtime |
|---------|-------|----------------|---------|---------|
| MATH500 | DS-Llama8B | 85.29 | 44.25 | 42.18 |
|         | DS-Qwen14B | 71.13 | 46.92 | 46.32 |
|         | QwQ-32B | 216.70 | 21.15 | 19.55 |
| GSM8K   | DS-Llama8B | 78.69 | 6.83 | 5.80 |
|         | DS-Qwen14B | 74.52 | 7.16 | 7.48 |
|         | QwQ-32B | 150.29 | 13.33 | 17.63 |

Table 19: Avg. # of steps and annotation runtime per sample

**Step occurrence.** Figure 48 presents the average number of consecutive steps of the same category. Notably, several high-frequent step types (such as *Formula Substitution*, *Verification* and *Formula Substitution*) tend to appear multiple times consecutively. Conversely, some steps tends to appear only once, where their average consecutive step is close to 1.

These observations support our design selection. Firstly, the sequence of repeated labels increases the robustness of the step classification and detection using our Step-Tagging module. Indeed, if a label appears multiple times in a row, local misclassifications are less likely to impact the overall framework. Secondly, the fact that certain steps tend to appear only one at a time (such as *Final Conclusion*, *Interpretation*, or *Context Repetition*) justifies our use of a token threshold $k$ for step delimitation. In fact, some categories often appear as single instances, suggesting that the step encapsulates a single type of thought.

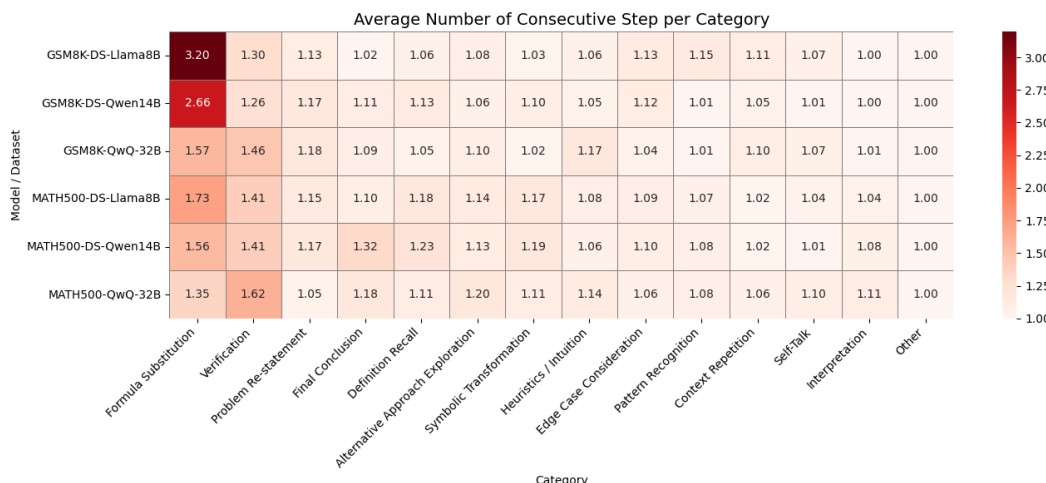

Figure 48: Avg. # of consecutive steps of same category

**Reasoning patterns.** We observe that our Step-Tagging framework allows the monitoring process to clearly follow the reasoning progression of the model. Figure 49 presents the step-types of the reasoning traces of the LRMs for a single sample of the MATH500 dataset. We selected samples that resulted in approximately the same number of steps across the models to allow a fair comparison between the models.

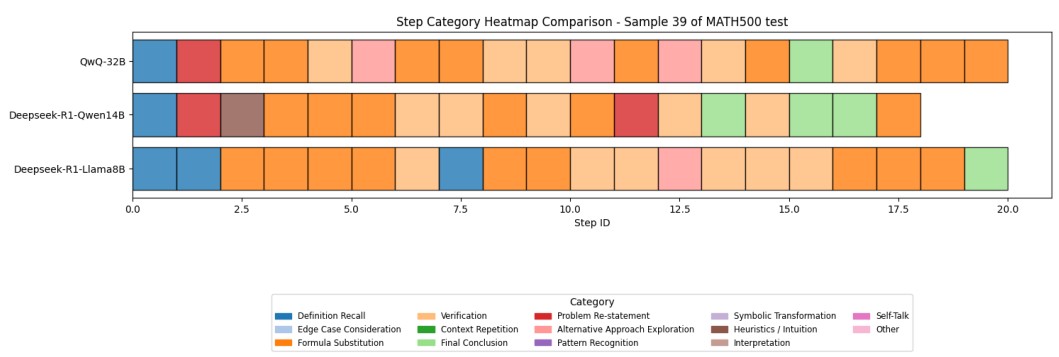

Figure 49: Reasoning patterns - Sample 109 of MATH500

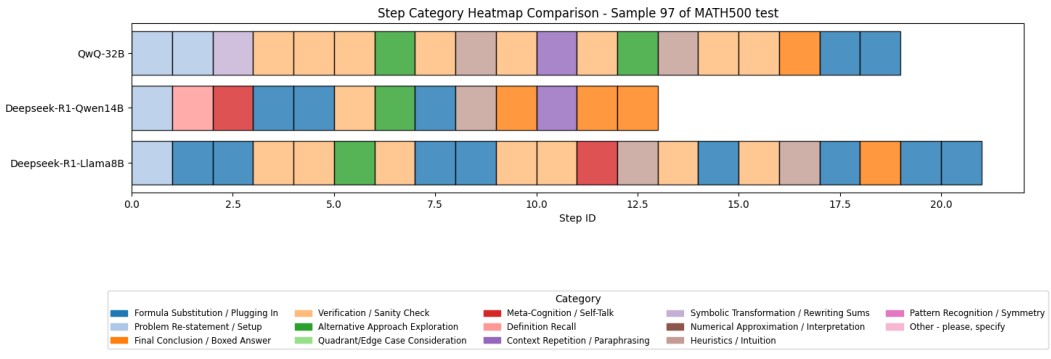

Figure 50: Reasoning patterns - Sample 97 of GSM8K

We observe a pattern in the reasoning traces. For both datasets, LRMs exhibit the same type of steps at the beginning of their generated output (e.g., *Definition Recall* or *Problem Re-statement*). Then we

notice a heavy use of *Verification* and *Formula Substitution*, helping the model to find and refine its current answer. Later, we observe the appearance of transition steps such as *Self-Talk* or *Alternative Approach Exploration*, which seems to lead to different answers and increases the diversity of the model's answers. We also observe that *Final Conclusion* steps appear in the middle of the reasoning traces, meaning that the model tends to draw intermediate conclusions but still pursues its reasoning, presumably because it is uncertain of the validity of the current solution.

**Visualization of reasoning step types.** To further assess the quality of the annotation, we computed the t-SNE projection of BERT-encoded reasoning steps from 500 DS-Llama8B traces (see Figure 51). For clarity, we excluded the most frequent step-types, *Verification* and *Formula Substitution*, as well as *Other* (since it does not contain any semantic meaning). Some step types (e.g., *Definition Recall*, *Problem Re-Statement*) are dispersed, which is likely due to question-specific semantics. However, we can observe distinct semantic clusters, particularly for *Interpretation*, *Self-Talk* and *Exploration*. Notably, the *Self-Talk* cluster lies semantically close to *Exploration*, reflecting their conceptual overlap: internal dialogue that often prompts the model to question its assumptions, encouraging alternative approaches.

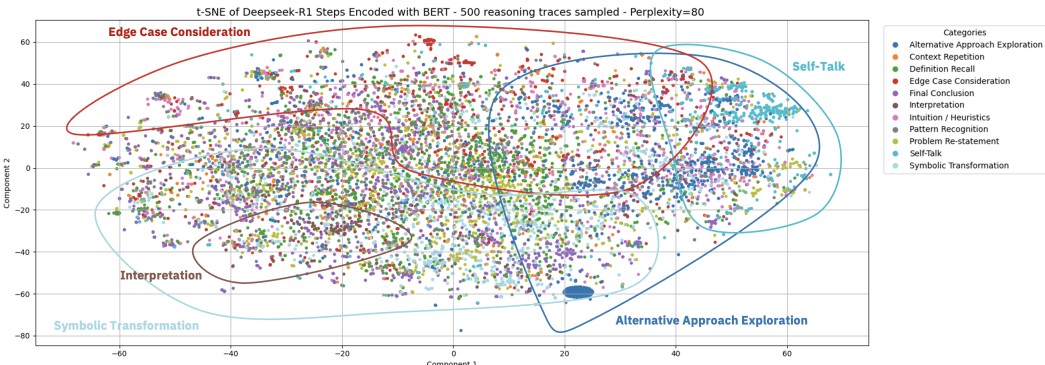

Figure 51: t-SNE visualization of 500 Deepseek-R1 reasoning steps encoded with bert-base-nli-mean-tokens (PPL = 80).

**Example of Step-Tagging reasoning step annotation by GPT-4o-mini.** Figure 52 present an example of reasoning trace annotated by GPT-4o-mini using our step-tags taxonomy. Figure 1 in the introduction is based on this example. This example is extracted from the DS-Qwen14B model on the sample 39 of the MATH500 dataset, using a seed of 42.

**Step 1 -** Okay, so I need to find the greatest common factor (GCF) of 6432 and 132, and then increase that GCF by 11. Hmm, let me remember how to find the GCF of two numbers. I think the best way is to use the Euclidean algorithm because it's systematic and works well for larger numbers.\n\n

**Step 2 -** First, let me write down the two numbers: 6432 and 132. Since 6432 is much larger than 132, I'll start by dividing 6432 by 132 and find the remainder. Then, I'll repeat the process with 132 and that remainder until the remainder is zero. The last non-zero remainder will be the GCF.\n\n

**Step 3 -** So, let's divide 6432 by 132. Hmm, I need to figure out how many times 132 goes into 6432. Maybe I can simplify this division. Let me see, 132 times 50 is 6600, which is more than 6432. So, it must be a little less than 50. Let me try 48 times 132. 48 times 100 is 4800, 48 times 32 is 1536, so adding those together, 4800 + 1536 is 6336. Hmm, 6336 is less than 6432. So, 132 times 48 is 6336. Subtracting that from 6432: 6432 - 6336 is 96. So, the remainder is 96.\n\n

**Step 4 -** Alright, now I need to find the GCF of 132 and 96. Let's apply the same method. Divide 132 by 96. 96 goes into 132 once, which is 96. Subtracting, 132 - 96 is 36. So, the remainder is 36.\n\n

**Step 5 -** Next step, find the GCF of 96 and 36. Dividing 96 by 36. 36 times 2 is 72, which is less than 96. Subtracting 72 from 96 gives 24. So, the remainder is 24.\n\n

**Step 6 -** Now, find the GCF of 36 and 24. Dividing 36 by 24, it goes once with a remainder. 36 - 24 is 12. So, remainder is 12.\n\n

**Step 7 -** Continuing, find the GCF of 24 and 12. Dividing 24 by 12, it goes exactly 2 times with no remainder. Since the remainder is zero, the last non-zero remainder is 12. Therefore, the GCF of 6432 and 132 is 12.\n\n

**Step 8 -** Wait, let me double-check that to make sure I didn't make a mistake. Starting from the top: 6432 divided by 132 is 48 with a remainder of 96. Then, 132 divided by 96 is 1 with a remainder of 36. Then, 96 divided by 36 is 2 with a remainder of 24. Then, 36 divided by 24 is 1 with a remainder of 12. Then, 24 divided by 12 is 2 with no remainder. Yep, that seems right. So, the GCF is indeed 12.\n\n

**Step 9 -** Now, the problem says to increase this GCF by 11. So, 12 plus 11 is... 23. Is that right? 12 + 11 is 23. Hmm, okay, that seems straightforward.\n\n

**Step 10 -** Just to be thorough, let me verify the GCF another way. Maybe by prime factorization. Let's factor both numbers into primes.\n\n

**Step 11 -** Starting with 132. 132 is even, so divide by 2: 132 / 2 = 66. 66 is also even, so divide by 2: 66 / 2 = 33. 33 is divisible by 3: 33 / 3 = 11. 11 is a prime. So, the prime factors of 132 are $2^2 * 3^1 * 11^1$.\n\n

**Step 12 -** Now, factorizing 6432. Let's see, 6432 is even, so divide by 2: 6432 / 2 = 3216. 3216 is even, divide by 2: 3216 / 2 = 1608. 1608 is even, divide by 2: 1608 / 2 = 804. 804 is even, divide by 2: 804 / 2 = 402. 402 is even, divide by 2: 402 / 2 = 201. Now, 201 is odd. Let's check divisibility by 3: 2 + 0 + 1 = 3, which is divisible by 3. So, 201 / 3 = 67. 67 is a prime number. So, putting it all together, the prime factors of 6432 are $2^5 * 3^1 * 67^1$.\n\n

**Step 13 -** Now, to find the GCF, we take the lowest power of the common prime factors. The common primes between 6432 and 132 are 2 and 3. In 132, 2 is squared, and in 6432, 2 is to the fifth power. So, we take the lower one, which is $2^2$. For 3, both have $3^1$. So, GCF is $2^2 * 3^1 = 4 * 3 = 12$. Yep, that matches my earlier result.\n\n

**Step 14 -** 'Therefore, the GCF is definitely 12. Adding 11 to that gives 23. So, the result is 23.\n\n"

**Step 15 -** I think that's solid. I don't see any mistakes in my calculations, and both methods gave me the same GCF. So, increasing it by 11 is straightforward. 12 + 11 is 23.\n\n

**Step 16 -** '**Final Answer** The result is boxed{23}. ¡/think¿ To find the greatest common factor (GCF) of 6432 and 132, we use the Euclidean algorithm: 1. Divide 6432 by 132, which gives a quotient of 48 and a remainder of 96. 2. Divide 132 by 96, which gives a quotient of 1 and a remainder of 36. 3. Divide 96 by 36, which gives a quotient of 2 and a remainder of 24. 4. Divide 36 by 24, which gives a quotient of 1 and a remainder of 12. 5. Divide 24 by 12, which gives a quotient of 2 and a remainder of 0.\n\n

**Step 17 -** The last non-zero remainder is 12, so the GCF of 6432 and 132 is 12.\n\n

**Step 18 -** Next, we increase this GCF by 11: 12 + 11 = 23. Thus, the result is boxed{23}.

**Legend:**
Definition Recall, Problem Re-statement, Heuristics / Intuition, Formula Substitution, Verification, Final Conclusion

Figure 52: Step annotation as per GPT-4o-mini using our step-tags taxonomy - Deepseek-R1-Qwen14b-Distil on MATH500 sample 39.

# N BASELINES

**Ideal Early-Stopping** $\mathcal{IES}$ **Algorithm.** Algorithm 3 presents the Ideal Early-Stopping $\mathcal{IES}$ baseline algorithm.

---

**Algorithm 3** Ideal Early-Stopping $\mathcal{IES}$

---

**Require:** Prompt $x$; reasoning delimiter $\alpha \in V$; minimal step size $k \in \mathbb{N}$; Ground-Truth Answer $y_{\text{gold}}$; Answer checker $\beta(\text{prediction}, \text{ground\_truth})$; Reasoning Language Model $\mathcal{M}$; tokenizer $\mathcal{T}$; EOS token $\gamma$;
1: $y \leftarrow \mathcal{T}(x)$                    ▷ Tokenize the input
2: $S_{running} \leftarrow []$;                    ▷ Initialize output
3: $t \leftarrow 0$
4: $b \leftarrow True$                    ▷ Initialize stopping criteria
5: **while** $b$ **do**                    ▷ Generate until constraint breaks
6:     Generate step $s_i$ using $\mathcal{M}, \alpha$, where $|s_i| > k$
7:     $y \leftarrow s_i$
8:     **if** $\beta(y, y_{\text{gold}})$ **then** $b \leftarrow False$                    ▷ Stop generation
9:     **else**
10:         Continue the generation
11:     **end if**
12:     $t \leftarrow t + 1$
13: **end while**
14: **return** y

---

**Prompt engineering** $\mathcal{P}_{\textbf{guided}}$. Figure 53 presents the different prompt variations as baseline that we defined.

**Prompt Baselines** $\mathcal{P}_{\textbf{guided}}$

User Prompt - $\mathcal{P}_{\text{user}}^{(0)}$

**User Prompt:** Please do not reason extensively, be succinct, and put your final answer within boxed{}. {question}

System Prompt $\mathcal{P}_{\text{system}}^{(0)}$

**System Prompt:** Respond concisely and confidently. Skip validations and over-verification steps.
**User Prompt:** {question}

System Prompt - $\mathcal{P}_{\text{system}}^{(1)}$

**System Prompt:** Respond concisely and confidently. Skip validations and over-verification steps. Here is an examples: Example 1: {FS_1}
**User Prompt:** {question}

System Prompt - $\mathcal{P}_{\text{system}}^{(3)}$

**System Prompt:** Respond concisely and confidently. Skip validations and over-verification steps. Here are some examples: Example 1: {FS_1} Example 2: {FS_2} Example 3: {FS_3}
**User Prompt:** {question}

Example 1 - Verification step

Wait, let me double-check. If I plug in $x = -3$ into the denominator, $(-3)^2 + (-3) - 6 = 9 - 3 - 6 = 0$. Yep, that works. For $x = 2$: $2^2 + 2 - 6 = 4 + 2 - 6 = 0$. Correct. So both roots are valid.

Example 2 - Verification step

Therefore, the graph of $y = 2(x^2 + x - 6)$ has vertical asymptotes at $x = -3$ and $x = 2$, so that's two vertical asymptotes. I don't think there's any chance that I made a mistake here, but maybe I should check by graphing the function or plugging in values close to $-3$ and $2$ to see if the function does go to infinity.

Example 3 - Verification step

Another test with $n = 3$. Let's compute manually. All non-empty subsets: Single elements: {1}, {2}, {3} with sums 1,2,3. Pairs: {1,2} $\rightarrow$ $2 - 1 = 1$ {1,3} $\rightarrow$ $3 - 1 = 2$ {2,3} $\rightarrow$ $3 - 2 = 1$. Triple: {1,2,3} $\rightarrow$ $3 - 2 + 1 = 2$. Total sum: $1 + 2 + 3 + 1 + 2 + 1 + 2 = 12$. Using the formula: contributions from each $k$ : $k = 3$: $3 * 2^2 * 1 = 3 * 4 = 12$. $k = 1$ and $k = 2$ contribute 0. So total sum 12, which matches.

Figure 53: Prompt baselines

## O  CALIBRATION OF THE ST-ES CRITERIA

Figures 54, 55 and 56 present the number of tokens vs. accuracy of every tag-types with values of threshold ranging from 0 to 20, for the DS-Qwen14B and QwQ-32B models on our train MATH500 and GSM8K datasets using the synthetic tags, respectively. Constraints selected using our methodology for each model and dataset are reported in Table 20.

| Model | Dataset | Level | Tag / Constraints | | | |
|---|---|---|---|---|---|---|
| | | | ST-ES (95%) | ST-ES (90%) | ST-ES (85%) | ST-ES (Router) |
| DS-Llama8B | MATH500 | 1 | Context Repetition / 0 | Context Repetition / 0 | **Self-Talk / 0** | Self-Talk / 0 |
| | | 2 | Final Answer / 0 | Symbolic Transformation / 2 | **Self-Talk / 0** | Self-Talk / 0 |
| | | 3 | Interpretation / 1 | **Intuition / 1** | Exploration / 3 | Intuition / 2 |
| | | 4 | Context Repetition / 1 | Context Repetition / 1 | **Intuition / 1** | Intuition / 2 |
| | | 5 | Exploration / 3 | Exploration / 3 | Edge Case / 1 | Intuition / 2 |
| | GSM8K | – | Verification / 3 | Verification / 1 | Verification / 0 | – |
| DS-Qwen14B | MATH500 | 1 | **Self-Talk / 0** | Verification / 5 | Exploration / 0 | Self-Talk / 0 |
| | | 2 | Exploration / 3 | **Self-Talk / 0** | Exploration / 1 | Self-Talk / 0 |
| | | 3 | Edge Case / 2 | Edge Case / 2 | **Exploration / 1** | Exploration / 3 |
| | | 4 | Exploration / 6 | **Exploration / 4** | Intuition / 2 | Exploration / 3 |
| | | 5 | Intuition / 6 | Self-Talk / 2 | **Exploration / 4** | Exploration / 3 |
| | GSM8K | – | Verification / 0 | Formula Substitution / 4 | Formula Substitution / 3 | – |
| QwQ-32B | MATH500 | 1 | **Exploration / 2** | Formula Substitution / 2 | Verification / 4 | Exploration / 2 |
| | | 2 | Final Answer / 2 | Intuition / 1 | **Exploration / 1** | Exploration / 2 |
| | | 3 | Verification / 6 | Verification / 5 | Verification / 4 | Intuition / 2 |
| | | 4 | **Intuition / 3** | Final Answer / 4 | Exploration / 2 | Intuition / 2 |
| | | 5 | Self-Talk / 3 | Self-Talk / 1 | Self-Talk / 1 | Intuition / 2 |
| | GSM8K | – | Intuition / 2 | Intuition / 1 | Exploration / 0 | – |

Table 20: Overview of tag/constraints determined using the training datasets, for each models.

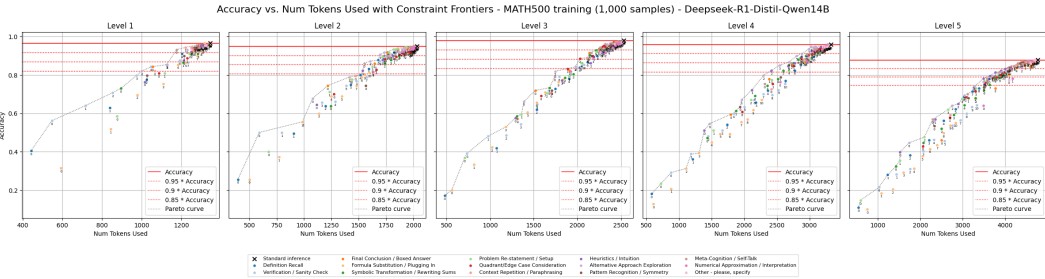

Figure 54: Early-Stopping selection using Pareto - Deepseek-R1-Qwen14B-Distil on MATH500 train (1,000 samples)

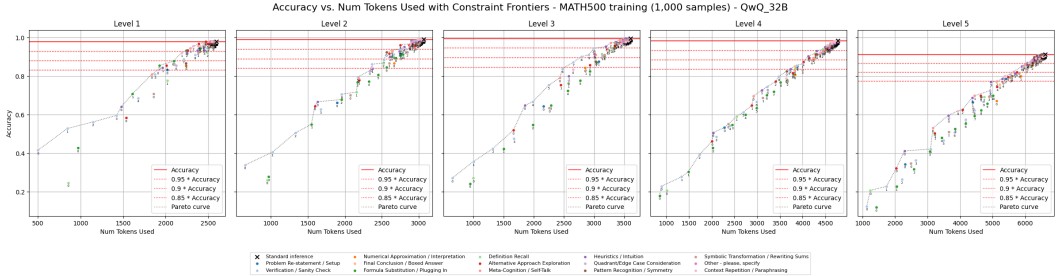

Figure 55: Early-Stopping selection using Pareto - QwQ-32B on MATH500 train (1,000 samples)

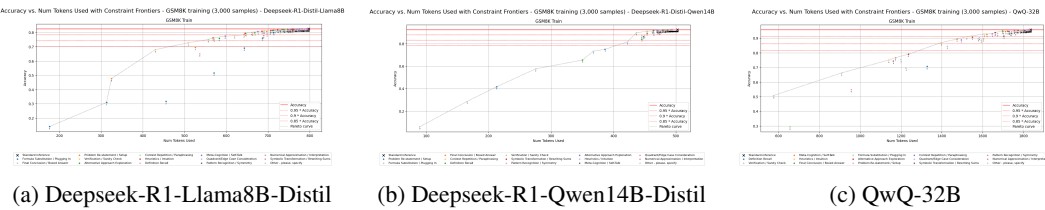

(a) Deepseek-R1-Llama8B-Distil   (b) Deepseek-R1-Qwen14B-Distil   (c) QwQ-32B

Figure 56: Early-Stopping selection using Pareto - GSM8K train (3,000 samples)

## P LLM-ROUTER: PROMPT COMPLEXITY CLASSIFICATION

Table 21 report the classifier performance of the LLM-Router module between classes {1,2} and {3,4,5} of the MATH500 dataset.

| Metric | Validation | Test |
|---------|------------|-------|
| Micro-F1 | 0.785 | 0.784 |
| Macro-F1 | 0.734 | 0.739 |

Table 21: Micro-F1 and Macro-F1 scores for validation and test sets

## Q STEP-TAGGING PERFORMANCE

Figures 57 and 58 presents the performance of the binary step-taggers trained on the training traces of the DS-Qwen14B and QwQ-32B, respectively.

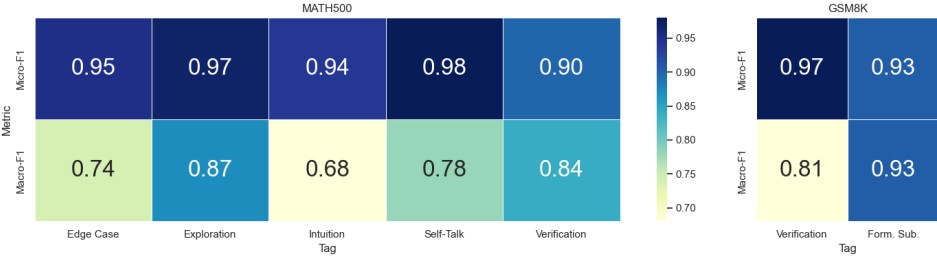

Figure 57: Step-Tagger performance - DS-Qwen14B

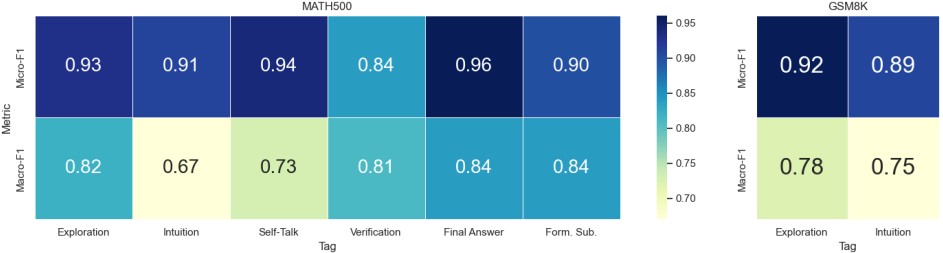

Figure 58: Step-Tagger performance - QwQ-32B

## R  IDEAL-EARLY-STOPPING

Figure 59 shows the number of steps and the accuracy of the standard vs. Ideal Early-Stopping $\mathcal{IES}$ criteria. Results are averaged over the 5 seeds, on the MATH500 test dataset.

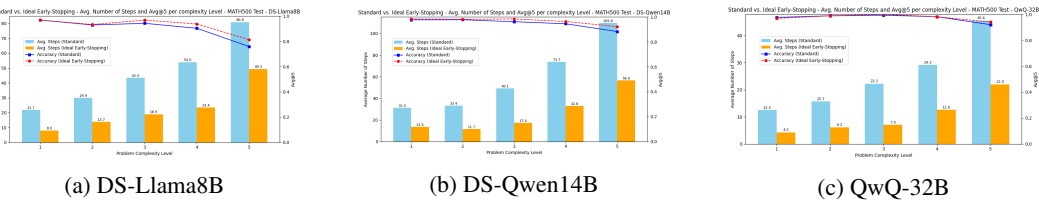

(a) DS-Llama8B  (b) DS-Qwen14B  (c) QwQ-32B

Figure 59: Standard vs. $\mathcal{IES}$ baseline - number of steps and Avg@5 across the 5 complexity levels of MATH500 test dataset - Results averaged over the 5 seeds

## S  PERFORMANCE OF THE ST-ES CRITERIA

Table 22 reports all the token-usage, the proportion of saved number of tokens, the Avg@5, the Pass@5 and the Cons@5 for all configurations. Results are averaged over the 5 seeds we used. We also show in Figure 60 and 61 the average token count against the Pass@5 and Con@5, respectively, for the three LRMs on the MATH500 and GSM8K datasets.

| Model | Config. | MATH500 | | | | | GSM8K | | | | |
|---|---|---|---|---|---|---|---|---|---|---|---|
| | | # Tokens | Saved (%) | Avg@5 | Pass@5 | Cons@5 | # Tokens | Saved (%) | Avg@5 | Pass@5 | Cons@5 |
| | Standard | 3655.0 | – | 0.878 | 0.970 | 0.726 | 958.3 | – | 0.829 | 0.943 | 0.651 |
| DS-8B | Basel. $\mathcal{IES}$ | 1916.6 | 47.56 | 0.911 | 0.980 | 0.780 | 385.3 | 59.79 | 0.847 | 0.952 | 0.726 |
| | Basel. $\mathcal{P}_{\text{user}}^{(0)}$ | 2989.6 | 18.21 | 0.866 | 0.952 | 0.722 | 525.8 | 45.13 | 0.771 | 0.917 | 0.579 |
| | Basel. $\mathcal{P}_{\text{system}}^{(0)}$ | 2634.4 | 27.92 | 0.817 | 0.960 | 0.592 | 456.9 | 52.32 | 0.763 | 0.895 | 0.574 |
| | Basel. $\mathcal{P}_{\text{system}}^{(1)}$ | 2139.5 | 41.46 | 0.782 | 0.942 | 0.526 | 560.8 | 41.48 | 0.754 | 0.914 | 0.537 |
| | Basel. $\mathcal{P}_{\text{system}}^{(3)}$ | 2565.3 | 29.81 | 0.789 | 0.952 | 0.540 | 830.5 | 13.34 | 0.748 | 0.904 | 0.541 |
| | ST-ES (95%) | 3260.5 | 10.79 | 0.883 | 0.972 | 0.730 | 673.8 | 29.69 | 0.818 | 0.933 | 0.663 |
| | ST-ES (90%) | 2949.3 | 19.31 | 0.859 | 0.964 | 0.666 | 568.5 | 40.67 | 0.799 | 0.931 | 0.604 |
| | ST-ES (85%) | 2413.9 | 33.95 | 0.801 | 0.940 | 0.556 | 492.1 | 48.65 | 0.745 | 0.923 | 0.474 |
| | ST-ES Router | 2656.2 | 27.33 | 0.848 | 0.956 | 0.686 | | | | | |
| | Standard | 3388.8 | – | 0.923 | 0.980 | 0.836 | 662.9 | – | 0.910 | 0.952 | 0.843 |
| DS-14B | Basel. $\mathcal{IES}$ | 1655.9 | 51.14 | 0.950 | 0.990 | 0.884 | 316.5 | 52.26 | 0.931 | 0.971 | 0.871 |
| | Basel. $\mathcal{P}_{\text{user}}^{(0)}$ | 2691.5 | 20.58 | 0.933 | 0.982 | 0.834 | 505.1 | 23.80 | 0.856 | 0.956 | 0.662 |
| | Basel. $\mathcal{P}_{\text{system}}^{(0)}$ | 2346.2 | 30.77 | 0.886 | 0.966 | 0.754 | 470.9 | 28.96 | 0.873 | 0.949 | 0.710 |
| | Basel. $\mathcal{P}_{\text{system}}^{(1)}$ | 2211.4 | 34.74 | 0.873 | 0.974 | 0.708 | 566.5 | 14.54 | 0.838 | 0.952 | 0.629 |
| | Basel. $\mathcal{P}_{\text{system}}^{(3)}$ | 2535.0 | 25.19 | 0.879 | 0.968 | 0.748 | 839.6 | -26.65 | 0.841 | 0.952 | 0.631 |
| | ST-ES (95%) | 3113.2 | 8.13 | 0.923 | 0.980 | 0.824 | 480.0 | 27.59 | 0.884 | 0.951 | 0.763 |
| | ST-ES (90%) | 2989.7 | 11.78 | 0.906 | 0.976 | 0.794 | 497.9 | 24.89 | 0.838 | 0.940 | 0.656 |
| | ST-ES (85%) | 2330.2 | 31.24 | 0.841 | 0.966 | 0.670 | 452.5 | 31.74 | 0.754 | 0.918 | 0.528 |
| | ST-ES Router | 2545.4 | 24.89 | 0.870 | 0.968 | 0.734 | | | | | |
| | Standard | 4475.3 | – | 0.954 | 0.984 | 0.898 | 2075.7 | – | 0.953 | 0.965 | 0.934 |
| QwQ-32B | Basel. $\mathcal{IES}$ | 2213.2 | 50.55 | 0.970 | 0.992 | 0.940 | 842.9 | 59.39 | 0.976 | 0.986 | 0.963 |
| | Basel. $\mathcal{P}_{\text{user}}^{(0)}$ | 2908.8 | 35.00 | 0.955 | 0.986 | 0.916 | 988.0 | 52.40 | 0.952 | 0.968 | 0.937 |
| | Basel. $\mathcal{P}_{\text{system}}^{(0)}$ | 3201.1 | 28.47 | 0.932 | 0.976 | 0.852 | 833.3 | 59.85 | 0.940 | 0.974 | 0.869 |
| | Basel. $\mathcal{P}_{\text{system}}^{(1)}$ | 3182.4 | 28.89 | 0.925 | 0.974 | 0.856 | 871.2 | 58.02 | 0.943 | 0.975 | 0.876 |
| | Basel. $\mathcal{P}_{\text{system}}^{(3)}$ | 3665.5 | 18.09 | 0.926 | 0.974 | 0.858 | 1387.3 | 33.16 | 0.935 | 0.974 | 0.855 |
| | ST-ES (95%) | 3679.4 | 17.78 | 0.921 | 0.980 | 0.786 | 1608.0 | 22.53 | 0.945 | 0.968 | 0.909 |
| | ST-ES (90%) | 3459.6 | 22.69 | 0.903 | 0.972 | 0.768 | 1506.3 | 27.43 | 0.935 | 0.967 | 0.888 |
| | ST-ES (85%) | 3218.3 | 28.09 | 0.878 | 0.976 | 0.690 | 1368.7 | 34.06 | 0.929 | 0.967 | 0.855 |
| | ST-ES Router | 3623.5 | 19.03 | 0.904 | 0.982 | 0.714 | | | | | |

Table 22: Performance of Step-Tagging Early stopping - 5 seeds (40, 41, 42, 43, 44)

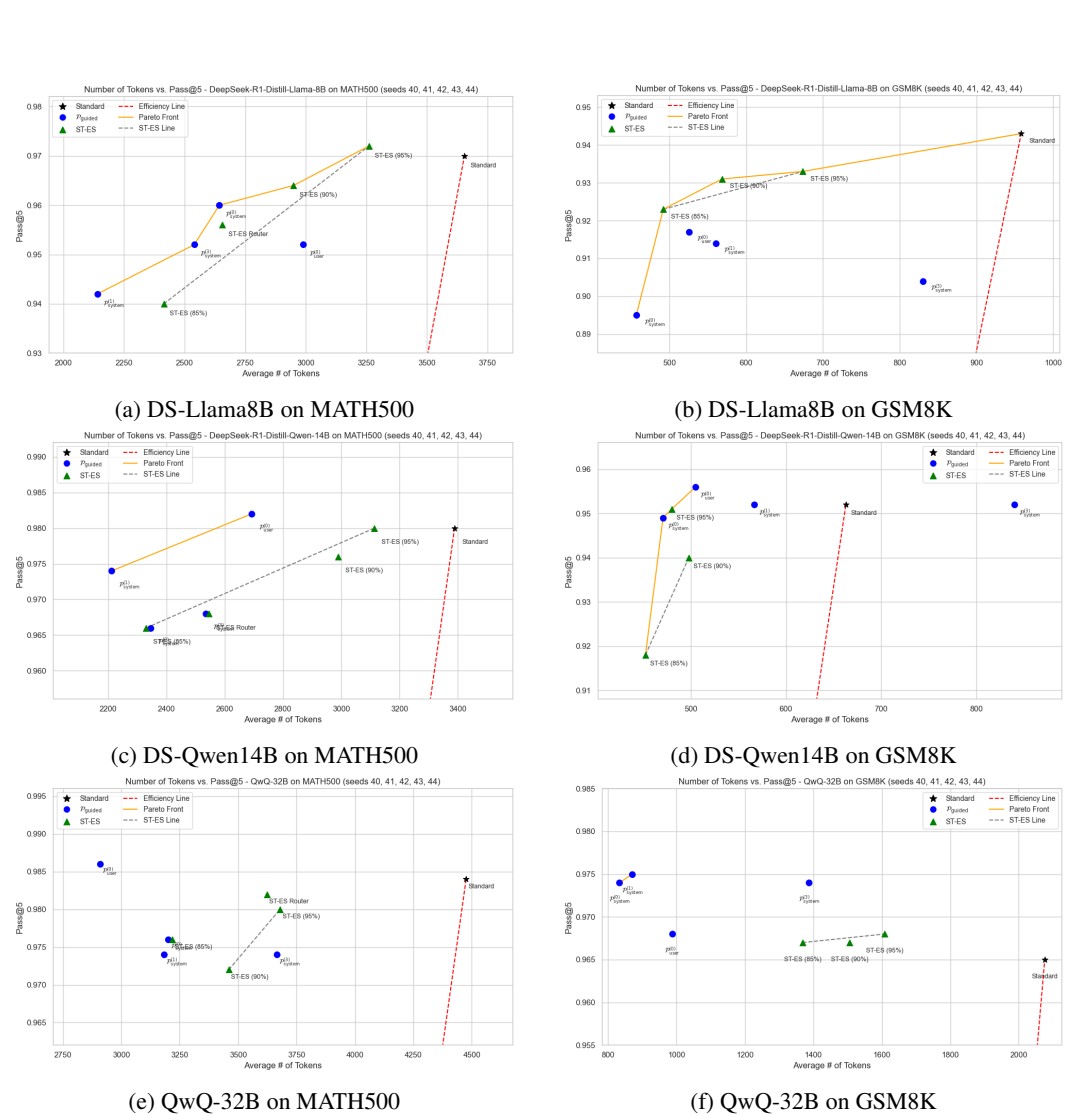

Figure 60: Number of Tokens vs. Pass@5 - $\mathcal{P}_{\text{guided}}$ Baselines and Step-Tagging Early-Stopping (ST-ES) criteria

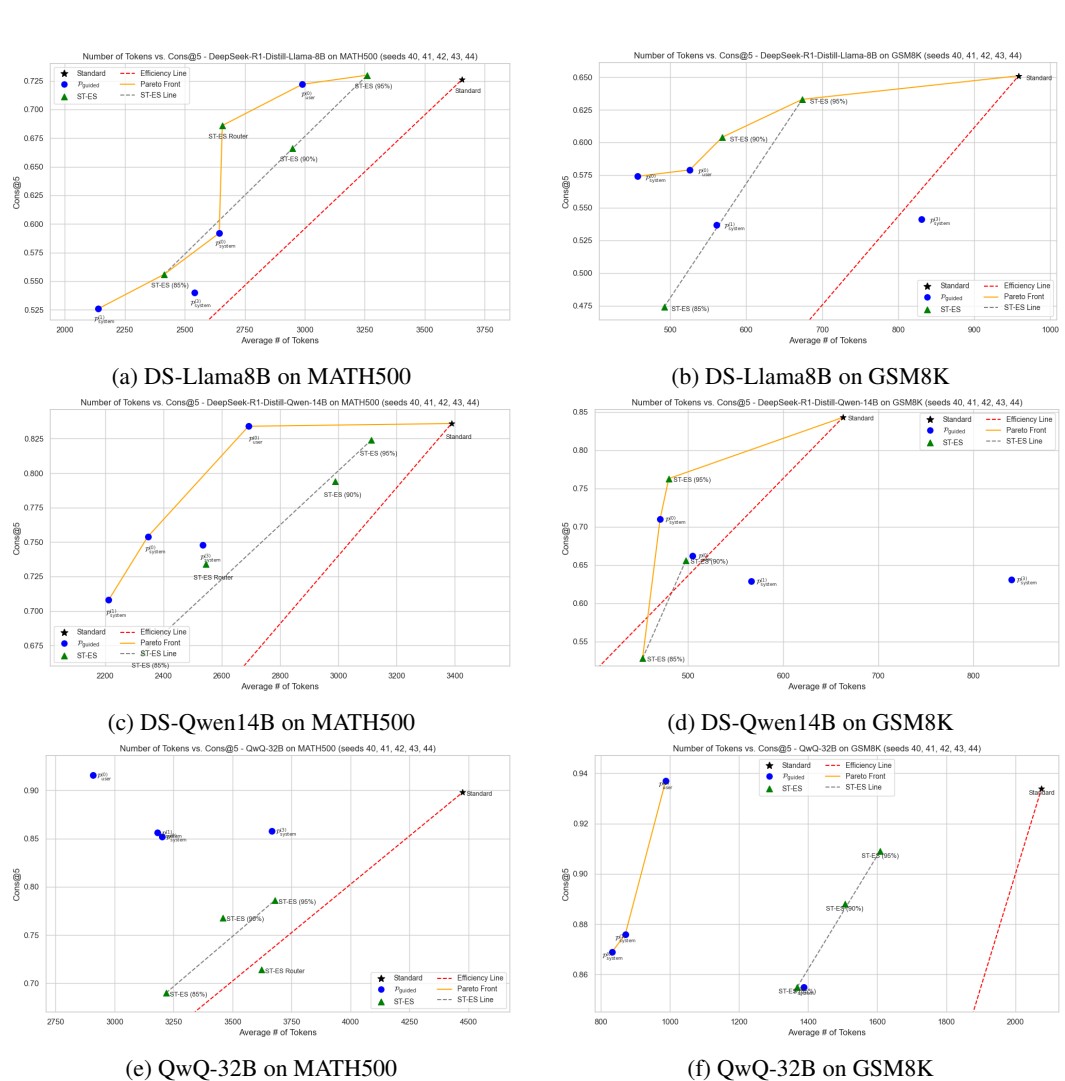

Figure 61: Number of Tokens vs. Cons@5 - $\mathcal{P}_{\text{guided}}$ Baselines and Step-Tagging Early-Stopping (ST-ES) criteria

