# OpenReview forum: "Step-Tagging: Toward controlling the generation of Language Reasoning Models through step monitoring"
_ICLR.cc/2026/Conference — Submitted to ICLR 2026_

### Official Review · Reviewer_2peB · 2025-10-16

**Soundness:** 2
**Presentation:** 3
**Contribution:** 2
**Rating:** 2
**Confidence:** 4

**Summary:**

This paper proposes **Step-Tagging**, a lightweight sentence-level classifier that labels each segment of a language reasoning model’s output with a reasoning-step type (the **ReasonType** taxonomy), enabling real-time monitoring and an interpretable early-stopping rule based on the frequency of specific step tags. Applied to open-source LRMs on MATH500 and GSM8K, Step-Tagging reduces generated tokens by ~30–40% while keeping accuracy comparable to standard inference, and often outperforms prompt-only efficiency baselines for smaller DeepSeek models. The method formalizes step segmentation (using a delimiter plus a minimum token length), trains binary tag detectors, and stops generation when a calibrated tag-count constraint is violated, then elicits the model’s current best answer—delivering controllable, efficient reasoning without heavy prompt engineering.

**Strengths:**

1. The paper is well written and easy to follow.
2. The empirical analysis is extensive and allows readers to clearly see how different factors affect the method’s performance.

**Weaknesses:**

1. The motivation is not fully articulated: although Lines 39–42 claim prior work “overlook[s] the possibility of monitoring the output,” the paper later only asserts a “new perspective” without explaining why it remedies prior limitations.
2. Comparisons to related work are incomplete. Prior studies on segmenting CoT steps [1, 2] substantially overlap with Section 3 and also discuss the unreliability of delimiter-based segmentation; even if the purpose here differs, this still weakens the contribution of Section 3.
3. The approach depends on training data to fit the Step-Tagging model and to calibrate the hyperparameter $\delta$, limiting applicability in low-resource settings.
4. While the method reports 30–40% token reduction, generating a ReasonType for every step during decoding may offset efficiency gains.

[1] Golovneva, O., Chen, M., Poff, S., Corredor, M., Zettlemoyer, L., Fazel-Zarandi, M., & Çelikyilmaz, A. (2022). *ROSCOE: A Suite of Metrics for Scoring Step-by-Step Reasoning.* ICLR2023.

[2] Luo, Y., Song, Y., Zhang, X., Liu, J., Wang, W., Chen, G., Su, W., & Zheng, B. (2025). *Deconstructing Long Chain-of-Thought: A Structured Reasoning Optimization Framework for Long CoT Distillation.* arXiv:2503.16385.

**Questions:**

1. Given that the ReasonType taxonomy in Figure 2 is derived from DeepSeek-R1-Distill-Llama-8B and QwQ-32B, how generalizable is it to other models?
2. $\delta$ is selected via a Pareto procedure using training data—how should $\delta$ be chosen when no train set is available?
3. Beyond math datasets, how does the method perform on broader reasoning benchmarks such as MMLU-Pro and GPQA?
4. Since GPT-4o-mini can be noisy, how do you ensure the quality of its generated training data?
5. Please revise formatting: e.g., the fonts in Figures 6–7 are too small to read, and Line 305’s “OpenAI et al. (2024)” should use \citep.

---

> ### Author Response · Authors · 2025-11-21
>
> Thank you for your review comments and feedback on our work. We appreciate your questions and your concerns about our work. It helps us to better understand our weaknesses and gives us ways of enhancing and strengthen our claims. Please find below answers to your questions:
>
> # Answer to questions:
>
>
> ## Question 1:
>
> We agree that it would be important to verify the generalisability of the taxonomy to other models. In a way, our experiments in Section P contributes to answer this question. Indeed, DS-Qwen14B was not used to derive our taxonomy. Therefore, the good performance of the BERT Classifiers on this model (see Appendix P) means that the taxonomy generalise to the DS-Qwen14B model.
>
> To further address your concern, we are running additional experiments on two additional reasoning models. Similarly as above, the goal of the experiment is to train BERT Classifiers following the same process as in our Experimental Design section. Good performance of the BERT Classifiers would validate the applicability of our taxonomy to the additional model tested.
>
> Would such experiment helps to address your concern?
>
>
> ## Question 2:
>
> We agree with your concern. Although, we observe through our experimentation that the value of $\delta$ seems to increase with the number of tokens of the standard inference. Across our experimentations, we can observe a common pattern. The reflection steps (such as Exploration, Self-Talk, or Intuition) seems to provide best accuracy/token_count trade-offs (see Appendix N in Table 16, these step-types appear frequently in the Table). In addition, our ablation study (Appendix H.3) and experimentation on the AIME dataset (see Appendix K) supports this statement.
>
> In addition, the value of $\delta$ increases with the difficulty of the problem (around 0 to 3 for GSM8K, 2 to 6 for MATH500, and 6 to 25 for AIME). While we have not run any ablation study to confirm this, it seems that the difficulty of the problem (and therefore the expected token-count from a model, for a given problem) could be used a proxy to determine the value of $\delta$. This is supported by the ST-ES Router that we implemented. We selected constraints per problem complexity, and routed the inference configuration.
>
>
> ## Question 3:
>
> We agree that it is important to validate our framework on other datasets than mathematical reasoning. To address this concern, we are conducting additional experimentation on the DS-Qwen14B model on the MMLU-Pro and GPQA-Diamond datasets (targeting other tasks than mathematics questions – as you suggested). We will include the results in a revised version of the paper soon. We hope that this additional experiment will help to address your concern.
>
> In addition, as suggested by Reviewer zExx, we conducted the same experiment on AIME, a harder mathematical dataset. We showed in Appendix K that our framework generalised well to this new dataset.
>
>
> ## Question 4:
>
> We agree that the annotation process of GPT-4o-mini can be noisy. For this reason, we included an analysis on the reliability of the GPT-4o-mini annotator in Appendix I.2. We compared the quality of the annotation across multiple runs of GPT-4o-mini, and compared it against 3 additional models.
>
>
> ## Question 5:
>
> Thank you for your feedback. We increased the size of the figures and captions, and corrected the citation in line 305.
>
>
> # Answer to Weaknesses:
>
> ## Weaknesses 1 and 2:
>
> Thank you for your feedback and for pointing out additional references that we missed. We will work on improving our motivation and Related Work section. We will upload a revised version soon to address your comments.
>
> ## Weaknesses 4:
>
> To better estimate costs implied by generating a ReasonType for every step during decoding, we conducted analysis of the inference runtime in Appendix J. We hope that such additional analysis will help to address your concern.
>
>
> If there are any further changes or clarifications needed, please let us know. Thank you once again for your comments and for your time!

---

> > ### Comment · Reviewer_2peB · 2025-11-23
> >
> > Thank you for the thoughtful rebuttal and the additional (and planned) experiments. However, my main concerns remain. Conceptually, the work is still not clearly distinguished from prior efforts on segmenting and scoring CoT steps (e.g., ROSCOE and related structured reasoning methods), and the rebuttal mostly promises to refine the writing rather than sharpen the core novelty or explain why the proposed taxonomy and stopping rule substantively go beyond existing controllers. Empirically, several key points are addressed by ongoing or planned experiments (additional models, non-math datasets), which are not part of the current submission, and the guidance on choosing the stopping hyperparameter without training data remains heuristic and problem-specific, limiting practical usability. Finally, while the new runtime analysis is appreciated, it does not convincingly dispel the concern that per-step tagging plus calibration may undercut the claimed efficiency gains. For these reasons, I prefer to maintain my original score.

---

> ### Author Response · Authors · 2025-11-26
>
> Thank you for your answer and for further detailing your concerns. Please find below answers to your questions:
>
> ## Weaknesses 1:
>
> We first want to clarify our work compared to the literature. We agree that our motivation was not fully articulated, and we thank you for providing feedback on this. To address our conceptual distinction compared to other works from the literature, we can distinguish two sub-fields of research that are closely related to our work:
>
> - Monitoring reasoning steps: First, as you highlighted, some works such as [1,2] introduces the concept of annotating the steps generated by LRMs. Our work is comparable to these two papers in the sense that these papers are treating the generation of LRMs as list of steps, and these approaches are tagging the steps using their own taxonomies. [1] is scoring the correctness of the steps, and defines a taxonomy of "Step-by-Step Reasoning Errors". [1] focuses on analysing the output of the LRMs and scores its correctness. [2] is parsing the generated output from LRM into 4 categories (Problem Restatement, Approach Exploration, Verification and Summary). [2] uses their parsed output to then fine-tune a model to improve its performance (Train-Time scaling). We acknowledge that these two papers are related to our work in the sense that they are also annotating reasoning steps. However, compared to our work, they are applied to different tasks (reasoning trace correctness, and Train-Time scaling, respectively). In addition, we annotate the reasoning steps differently: using our own taxonomy (fine-grained), and in a different setting (our annotation is performed online - as the LRM unfolds its reasoning - compared to [1,2] which is performed post-hoc).
>
> - Early-stopping controllers: Second, there is a growing literature in the field of Test-Time scaling, specifically looking at early-stop the generation of models using different signals. We can decompose the literature of stopping rules into two sub-fields. On the one hand, we observe static controllers: the stopping condition does not rely on the model's generation. These methods include token-count based early-stopping, and prompt compression. On the other hand, we observe dynamic early-stopping criteria which are either based on the model's confidence, or the entropy of the model's generation. In this case, both model's confidence or entropy are computed online (as the LRM unfolds its reasoning), so these early-stopping controller relies on the model's generation during inference. To the best of our knowledge, no prior inference-time efficiency framework uses online monitoring of the semantic content of the model’s generation to dynamically drive stopping decisions. To fill this gap, our claim is that differentiating the step-types that are iteratively generated by LRMs is important to early-stop.
>
> We added the following modifications in the paper:
> - Line 39-42: We re-worked the articulation of our motivation by pointing out the gap in the field of inference-time scaling, specifically mentioning the gap on early-stopping controllers.
> - Line 108-119: We included the missing references that you pointed out in your weakness 2.
> - Line 121-131: We highlighted the gap existing in early-stopping controllers, and formulated the existing gap more clearly.
> To complement our Related-Work section, we added two Tables to compare our work with the literature (see Appendix C.3 and C.4)
>
> If there are any further changes or clarifications needed, please let us know. Once again, thank you for your comments and for your time!

---

> ### Author Response · Authors · 2025-11-26
>
> Thank you for your answer and for further detailing your concerns. Please find below answers to your questions:
>
> ## Weaknesses 2:
>
> We agree that our section in Related-Work on Monitoring reasoning steps was incomplete, specifically mentioning and comparing existing methods that labels the reasoning steps. We suggested a revised version of this point in Line 108-119.
>
> As well, after carefully reviewing references [1,2] that you shared with us, we agree that our section 3 was incomplete. Indeed, [1] relies on the model to directly segment reasoning traces (the authors borrowed this approach from [3]). And [2] prompts an LLM to parse the raw reasoning output, and segment the reasoning. For this reason, we added another type of approach in Line 171-174 (see Section 3).
>
> Furthermore, in your weakness 2, you mention that [1,2] discuss the unreliability of delimiter-based segmentation. We have reviewed the two papers that you referenced in that perspective, however, we are confused as we do not see the discussion about the unreliability of delimiter-based segmentation in both of these papers. We agree with your point, and we acknowledge that delimiter-based segmentation methods are not an ideal definition of the reasoning steps (this point is also supported by the literature [4]). However, we found that this definition seems to be commonly used in the literature (see Section 3.1), and we conducted an extensive analysis to validate our step segmentation in Appendix G.
>
> [3]  Maxwell Nye, et al. Show your work: Scratchpads for intermediate computation with language models. arXiv preprint arXiv:2112.00114, 2021.
>
> [4] Jinu Lee, Julia Hockenmaier (2025). Evaluating Step-by-step Reasoning Traces: A Survey. EMNLP 2025 Findings (arXiv preprint arXiv:2502.12289v1)
>
> Once again, thank you for your comments and for your time!

---

> ### Author Response · Authors · 2025-11-26
>
> Thank you for your answer and for further detailing your concerns. Please find below answers to your questions:
>
> ## Additional experiments and materials
>
> We updated our version of the paper to include additional experiments on the GPQA and MMLU-Pro datasets in Appendix K.2. Using the DS-Qwen14B model, we obtained from around 20 to 50% token-count saving on the GPQA-Diamond, and from 25 to 45% token-count saving on the MMLU-Pro, with minimal accuracy drop compared to the standard inference. In addition, we reported satisfying performance of the step-classifiers on these two datasets, which further validate the applicability of some classes of our taxonomy to other tasks than mathematical questions.
>
> ## Planned experiments
>
> In addition, we are working on adding a few additional materials:
> - Further validating our taxonomy with additional models: We selected two additional reasoning models, and inferred them on MATH500 and GSM8K datasets. We then labelled each reasoning steps, and trained step-classifier on selected step-types from our taxonomy. The performance of the step-classifier would further validate the generalization of our taxonomy to other models.
> - Training-inference analysis: In addition to our Appendix J.1 are working on adding an additional section specifically analysing the cost implied by the calibration process.
> - We are also working on addressing other reviewer's concerns with additional materials, such as explaining the Weaker Performance on QwQ-32B, or analysing the impact of the performance of the BERT Router on the framework's efficiency.
>
> If there are any further changes or clarifications needed, please let us know. Once again, thank you for your comments and for your time!

---

> > ### Author Response · Authors · 2025-12-02
> >
> > Again, thank you for your review comments and feedback on our work. Please find bellow additional modifications that we included in our paper to further address your concerns:
> >
> > ## Question 1:
> >
> > To address you concern regarding the generalizability of our ReasonType taxonomy to other models, we included an additional experiment in Appendix I.3. Specifically, we selected two additional reasoning models (microsoft/Phi-4-reasoning and Qwen/Qwen3-30B-A3B-Thinking-2507), and inferred them on MATH500 and GSM8K datasets. We then labelled each reasoning steps using our annotation pipeline, and trained step-classifier on four different step-types from our taxonomy. We obtained satisfying performance of the step-classifier, comparable to the results obtained on same datasets for our 3 original LRMs. Because these models were not used to derive our ReasonType taxonomy, we interpret these results as validating the generalization of our taxonomy to other reasoning models.
> >
> > ## Weaknesses 3 and 4:
> >
> > To further address your concerns regarding the latency and additional computations implied by our calibration step, we added a training-inference trade-off analysis in Appendix J.2. We demonstrated that our Step-Tagging framework can recover its one-time training and calibration cost during inference, and continues to provide additional runtime saving compared to standard inference as more inference tokens are generated. We also discuss on this impact that the size of the training dataset has on the training-inference trade-off. We hope that such analysis can further address your concerns regarding the training cost of our framework.
> >
> > ## Additional experiments
> >
> > As stated in our previous comment, we also attached additional experiments to address other reviewer's concerns:
> >
> > - To further explain why our framework reduced effectiveness on the largest model (QwQ-32B), we included additional analysis in Appendix L. We demonstrated that QwQ-32B exhibits a more conservative and less destructive reasoning generation, meaning that it overwrites intermediate correct answer that it generates less frequently compared to the Deepseek models. Because our ST-ES framework aims to stop the generation before correct answers are overwritten, it leaves fewer opportunities to early-stop the QwQ-32B model compared to the Deepseek models.
> >
> > - An analysis on the robustness of our BERT router configuration for the MATH500 dataset was missing in our initial submission. To address this, we included an analysis quantifying the impact of the propagation of the BERT router errors to the overall accuracy/efficiency of our ST-ES framework, we included an ablation study in Appendix J.3. Specifically, we analysed how the performance of different BERT router impact our ST-ES framework - for easy and hard problems. We conclude that Router errors have asymmetric impact: over-allocation of computation on easy problems mainly hurts efficiency, while under-allocation of computation on hard problems hurts both accuracy and efficiency.
> >
> > If there are any further changes or clarifications needed, please let us know. Once again, thank you for your comments and for your time!

---

### Official Review · Reviewer_nRdQ · 2025-10-28

**Soundness:** 2
**Presentation:** 1
**Contribution:** 3
**Rating:** 4
**Confidence:** 4

**Summary:**

The paper proposes a taxonomy (ReasonType) for online sentence-level classification of reasoning steps. The authors show how using this taxonomy and counts of specific step tags can serve as interpretable early-stopping criteria calibrated via trade-offs of accuracy vs. tokens. Experiments on math datasets (MATH500 and GSM8K) on multiple LLM models show they can achieve 30–40% token reductions with comparable accuracy to standard generation.

**Strengths:**

1. The paper presents a taxonomy of reasoning steps in LLMs.
2. The authors correctly leverage their taxonomy to obtain token efficiency without damaging results.
3. The paper clearly presents their idea and method.

**Weaknesses:**

1. Tags are derived from GPT-4o-mini. The authors do not mention or run an ablation study on this training dataset.
2. Ablation on labels. The authors do not show the quality of their tags. They can extract a subset of their dataset and show a comparison with other models or human annotators.
3. The BERT router’s Micro-F1 ≈0.78 suggests routing errors may affect benefits. It is unclear how router errors propagate to overall accuracy/efficiency
4. Figures cannot be correctly visualized at the current font size.

**Questions:**

Apart from the points raised in the Weaknesses section, I also have the following questions:
1. How does this taxonomy and method generalize over non-math tasks?
2. When early-stopping hurts accuracy (e.g., QwQ-32B), which tags/thresholds are implicated? Could a multi-tag or stateful policy mitigate regressions?
3. Hw does Step-Tagging compare to prompt-only compression in both accuracy and serving cost \across loads? This can allow a fair comparison at equal budget.

---

> ### Author Response · Authors · 2025-11-21
>
> Thank you for your review comments and feedback on our work. We appreciate your questions and your concerns about our work. It helps us to better understand our weaknesses and gives us ways of enhancing and strengthen our claims. Please find below answers to your questions:
>
> # Answer to questions:
>
> ## Question 1:
>
> We agree that additional experiments on other tasks than math could help to generalize the framework further. We conducted additional experimentation on the DS-Qwen14B model, on the AIME dataset (stronger mathematical dataset), and on the MMLU-Pro and GPQA datasets (targeting other tasks than mathematics questions – as suggested by Reviewer 2peB).
>
> - More challenging mathematical dataset: Experimentations on the AIME (22-24) dataset are detailed in Appendix K. Experimentations showed that our framework transfer well when applied to a harder mathematical tasks domain, and show more pronounced results, especially when comparing our framework to the baselines.
>
> - Generalization to Other Tasks: We are wrapping-up our additional experimentations on the GPQA-Diamond and MMLU-Pro datasets and we will include the same analysis as for the AIME dataset in an upcoming submission. We hope that this additional experiment will help to address your concern.
>
>
> ## Question 2:
>
> We agree that our section 7 could be deepened to better understand the weaker performances on the QwQ-32B model for example. We are currently running additional ablation to address this point and add our results an upcoming submission.
>
>
> ## Question 3:
>
> We acknowledge that an analysis on the runtime and serving cost was missing in our paper, and we thanks you and other reviewers for sharing this concern. We added an analysis on the runtime in Appendix J. We are planning to add a runtime comparison with the prompt-guided baselines, as well as some additional ablation studies on the training-inference trade-off between the Step-Tagging approaches and the baselines.
>
> Although, could you please clarify your question? By serving cost, are you referring to runtime and token count? Or are you referring to other metrics?
>
>
> # Answer to Weaknesses:
>
> ## Weakness 1:
>
> To address your concern regarding the ReasonType taxonomy derived from GPT-4o-mini, we detailed our approach further in the Appendix I.1. We included some examples of the training data that helped us to derive the taxonomy. If it helps, we also suggest to add the reasoning steps that were passed the GPT-4o-mini to create the tags.
>
>
> ## Weakness 2:
>
> Even though the annotation process lead to the training of accurate BERT classfiers, we agree that such analysis was missing. We included an analysis on the reliability of the GPT-4o-mini annotator in Appendix I.2. We compared the quality of the annotation across multiple runs of GPT-4o-mini, and compared it against 3 additional models.
>
>
> ## Weakness 3:
>
> We agree that such analysis would be interesting, and important when relying on the BERT router approach. We are planning to include such analysis in an revised version soon. To do so, we will observe the Step-Tagging performance against different accuracy of the BERT router. We hope that this will helps to address your concerns.
>
>
> ## Weakness 4:
>
> We agree that the Figures were hardly readable. We increased the size of Figures in the version that we uploaded.
>
>
> If there are any further changes or clarifications needed, please let us know. Thank you once again for your comments and for your time!

---

> > ### Comment · Reviewer_nRdQ · 2025-11-27
> >
> > Thank you for your comments and commitment to the rebuttal phase.
> >
> > I think you should also update the paper to coherently include your new results. For instance, you abstract only mentions results for MATH500 and GSM8K. And your paper includes nearly 40 pages of appendices which is barely included in the main section of the paper. The content needs to be cohesively include in the main section of the paper.
> >
> > In Q3, I referred to the tokens.
> >
> > Currently, the paper still lacks from a good and clear explanation and report of the results and analysis.
> >
> > I will wait for the missing results you mentioned to write down my final comments.

---

> > > ### Author Response · Authors · 2025-12-02
> > >
> > > Again, thank you for your review comments and feedback on our work. Please find bellow additional modifications that we included in our paper to further address your concerns:
> > >
> > > ## Question 1:
> > >
> > > As well as experimentations on harder mathematical datasets as you suggested (AIME - see Appendix K.1), we conducted experimentations on non-mathematical datasets (GPQA-Diamond and MMLU-Pro) in Appendix K.2. We demonstrated that our framework transfers well to non-math tasks, with 20 to 50% token-count saving on the GPQA-Diamond, and from 25 to 45%, with minimal accuracy drop (-3.8% and -2.3% accuracy drop, respectively). We hope that this additional experiment will help to address your question 1.
> > >
> > > ## Question 2:
> > >
> > > To further explain why our framework reduced effectiveness on the largest model (QwQ-32B), we included additional analysis in Appendix L. We demonstrated that QwQ-32B exhibits a more conservative and less destructive reasoning generation, meaning that it overwrites intermediate correct answer that it generates less frequently compared to the Deepseek models. Because our ST-ES framework aims to stop the generation before correct answers are overwritten, it leaves fewer opportunities to early-stop the QwQ-32B model compared to the Deepseek models.
> > >
> > > Furthermore, while a multi-tag approach (as you suggested) could have been interesting to mitigate this problem, we observed that merging multiple tags of our taxonomy (therefore defining constraints on multiple tags) did not helped to enhance the performance obtained by our early-stopping criteria on this model. Indeed, in Appendix H.4, we conducted an analysis on alternative taxonomies (the objective was to further validate our ReasonType taxonomy by comparing it to alternative taxonomies). To this extend, we defined different taxonomies by merging similar labels into common high-level classes. We showed that alternative taxonomies applied to the QwQ-32B obtained similar to worst performances when compared to our ReasonType taxonomy. This finding supports that the lower accuracy drop obtained on the QwQ-32B model could not be mitigated by a multi-label approach.
> > >
> > > ## Question 3:
> > >
> > > Your question 3 referred to the comparison of the prompt-only compression in both accuracy and serving cost \across load. We acknowledge that plotting such Figure could enhance the understanding of our results. However, our selection of the ST-ES configurations is done by targeting specific expected accuracy drops. Moreover, our baseline does not includes specific token-count budgets. For these reasons, it is challenging to plot such Figure.
> > >
> > > That being said, we obtained comparable token-count saving for many of our experimentations. Specifically, for DS-Llama8B on MATH500 and GSM8K, we explicitly compared the Avg@5 obtained by prompt-only baselines and our ST-ES configurations and showed that our ST-ES framework obtained higher accuracy for similar token-count budget (see Lines 485 to 490).
> > >
> > > In addition, while applying our framework on GPQA-Diamond and MMLU-Pro, targeting expected accuracy drops was not possible (because computing the accuracy on early-stopped non-math problems is hard - see Lines 2707 to 2711 in Appendix K.2). For this reason, we targeted specific token-count saving during our calibration analysis. The results obtained allowed us to better compared our ST-ES configurations at same token-count budget compared to prompt-only baselines, specifically for the GPQA-Diamond dataset. To address your concern, we conducted a qualitative analysis of our ST-ES configurations compared to prompt-only baselines from Line 2830 to 2835 in Appendix K.2.
> > >
> > > ## Weakness 3:
> > >
> > > To address your concern regarding the propagation of the BERT router errors to the overall accuracy/efficiency of our ST-ES framework, we included an ablation study in Appendix J.3. Specifically, we analysed how the performance of different BERT router impact our ST-ES framework - for easy and hard problems. We conclude that Router errors have asymmetric impact: over-allocation of computation on easy problems mainly hurts efficiency, while under-allocation of computation on hard problems hurts both accuracy and efficiency.
> > >
> > > ## Better explaining and report our results
> > >
> > > We acknowledge that we included many additional results and ablation studies to our initial submission. In order to better communicate our results, we updated the abstract, introduction, and main results sections to reflect the full set of additional materials that we provided (e.g. additional experiments on AIME, GPQA and MMLU-Pro datasets). While we include extensive appendices to address the concerns of the reviewers and further support our claims, we modified our initial submission to ensure that the key findings currently placed in the appendices are summarized and discussed directly in the main text.
> > >
> > > If there are any further changes or clarifications needed, please let us know. Thank you once again for your comments and for your time!

---

### Official Review · Reviewer_a7RC · 2025-11-01

**Soundness:** 3
**Presentation:** 2
**Contribution:** 2
**Rating:** 4
**Confidence:** 3

**Summary:**

This paper proposes a framework called "Step-Tagging" aimed at addressing the inefficiency issue in Language Reasoning Models (LRMs). The framework introduces a novel taxonomy of reasoning steps (ReasonType), uses a lightweight classifier to tag the steps generated by LRMs in real-time, and implements an interpretable early stopping strategy based on the counts of specific steps. Experiments demonstrate that this method can reduce token consumption by 30-40% while maintaining comparable accuracy.

**Strengths:**

- A new taxonomy of reasoning steps with 13 categories is proposed, providing a tool for fine-grained understanding and monitoring of the LRM reasoning process.

- A lightweight sentence classifier module is designed, capable of identifying the type of steps being generated by LRMs in real-time, enabling online monitoring of the reasoning process.

- An interpretable early stopping mechanism is validated based on the frequency of specific step types, demonstrating significant token reduction while maintaining comparable performance to standard generation.

**Weaknesses:**

- An evaluation of the latency introduced by the Step-Tagger module in inference scenarios must be included in the paper. It needs to be demonstrated that the inference time of the classifier itself is significantly less than the time saved by token reduction.

- Although Appendix G argues for the choice's reasonableness, it remains a critical hyperparameter that needs manual calibration for each new model, increasing the method's application complexity.

- The P_guided baseline (especially the few-shot system prompt) performs very strongly, even outperforming the ST-ES method on the QwQ-32B model. Considering ST-ES requires thousands of labeled samples and additional model training, while the P_guided baseline is almost zero-cost or very low-cost, the paper should discuss this "training cost vs. inference benefit" trade-off more deeply in Section 7.

**Questions:**

See above.

---

> ### Author Response · Authors · 2025-11-21
>
> Thank you for your review comments and feedback on our work. We appreciate your questions and your concerns about our work. It helps us to better understand our weaknesses and gives us ways of enhancing and strengthen our claims.
>
> # Answer to questions:
>
>
> ## Question 1:
>
> First, we acknowledge that an analysis on the runtime was missing in our paper, and we thanks you and other reviewers for sharing this concern. We added such analysis in Appendix J. Our conclusions supports that the impact of the Step-Tagging classifiers is minimal on the runtime, and allow the user to computational benefits of the gains of our framework.
>
> ## Question 2:
>
> We acknowledge this limitation: the parameter k is a critical parameter to select as it depends on both models and datasets. As Reviewer zExx also suggested, a dynamic value of k could solve this problem and allow more flexibility and less complexity of our framework. For this reason, we would like to explore this aspect in future work.
>
> ## Question 3:
>
> We acknowledge that our framework imply a trade-off: To provide the user with more efficient inferences, our method requires calibration and compute beforehand (which can be viewed as a training process). To better estimate costs and overhead computations implied by our method, we will also add in an upcoming submission a training-inference trade-off analysis, as you suggested.
>
> If there are any further changes or clarifications needed, please let us know. Thank you once again for your comments and for your time!

---

> > ### Author Response · Authors · 2025-12-01
> >
> > Again, thank you for your review comments and feedback on our work. Please find bellow additional modifications that we included in our paper to further address your concerns:
> >
> > ## Question 3:
> >
> > To further address your concerns regarding the latency and additional computations implied by our calibration step, we added a training-inference trade-off analysis in Appendix J.2. We demonstrated that our Step-Tagging framework can recover its one-time training and calibration cost during inference, and continues to provide additional runtime saving compared to standard inference as more inference tokens are generated. We hope that such analysis can further address your concerns regarding the training cost of our framework.
> >
> > If there are any further changes or clarifications needed, please let us know. Thank you once again for your comments and for your time!

---

### Official Review · Reviewer_zExx · 2025-11-01

**Soundness:** 2
**Presentation:** 2
**Contribution:** 3
**Rating:** 6
**Confidence:** 3

**Summary:**

This paper addresses the issues of "overthinking" and inefficiency in current Language Reasoning Models (LRMs) by proposing a lightweight framework called "Step-Tagging." This framework utilizes a real-time sentence classifier to annotate the type of each step generated by a Large Language Model (LLM) during its reasoning process. To achieve this, the authors first introduce a taxonomy of reasoning steps called "ReasonType," which includes 13 distinct reasoning behaviors (e.g., "Problem Restatement," "Formula Instantiation," "Verification"). Building on this framework, the paper further develops an interpretable "Early-Stopping" mechanism. This mechanism dynamically halts the model's output by monitoring the frequency of specific types of reasoning steps, stopping when the model has either generated sufficient information or begins to produce redundant steps. This approach significantly reduces the number of generated tokens while maintaining answer accuracy. The method was validated on the MATH500 and GSM8K mathematical reasoning datasets across three open-source LLMs (DS-Llama8B, DS-Qwen14B, QwQ-32B). The results demonstrate that this method can reduce token generation by 30% to 40% with only a minor loss in accuracy. This work provides a novel approach and tool for enhancing the controllability and efficiency of language reasoning models.

**Strengths:**

1.  **High Innovativeness and Practicality:** The paper directly confronts the core pain point of low efficiency in current LLMs for complex reasoning tasks. The proposed Step-Tagging framework and ReasonType taxonomy offer a novel and practical perspective for understanding and controlling the model's "thought process." Compared to methods that rely on "black-box" approaches or complex prompt engineering, this framework is more interpretable and generalizable.

2.  **Clear Methodology and Complete Structure:** The paper is well-structured, with a clear logical chain from problem statement and literature review to the definition of reasoning steps, construction of the taxonomy, and the design and experimental validation of the Step-Tagging module and early-stopping strategy.

3.  **Sufficient and Solid Experimental Design:**
    *   **Multi-Model, Multi-Dataset Validation:** Experiments were conducted on three open-source models of varying sizes and architectures, as well as on two mainstream mathematical reasoning datasets, which strengthens the generalizability of the conclusions.
    *   **Comprehensive Ablation Studies:** The paper validates its core design choices through extensive ablation studies. For example, it thoroughly investigates and validates the selection of the reasoning step separator `k`, the effectiveness of the ReasonType taxonomy, and comparisons against a simple "step-counting" strategy.
    *   **Convincing Baseline Comparisons:** The inclusion of an "Ideal Early-Stopping" (IES) baseline and various "Prompt-guided efficiency" (Pguided) baselines makes the experimental comparisons fairer and more persuasive.

4.  **Inspirational for Future Research:** This work not only provides a practical tool for improving efficiency but also opens up new avenues for future research. The proposed ReasonType taxonomy and the analysis of model reasoning behavior (such as the frequency and sequential patterns of different reasoning steps) pave the way for studying the interpretability of LLMs' "chain of thought" and analyzing model behavior.

**Weaknesses:**

1.  **Taxonomy Subjectivity:** The "ReasonType" taxonomy was created with GPT-4o-mini, which introduces potential subjectivity and dependency on a specific model's capabilities.
2.  **Application Complexity:** The framework requires a calibration step ("Pareto-curve") to find the optimal stopping strategy for each model and task, which raises the barrier to adoption.
3.  **Offline vs. Online Gap:** Experiments were simulated offline. The potential latency from a real-time, on-the-fly implementation and its impact on performance were not analyzed.
4.  **Weaker Performance on QwQ-32B:** The method's reduced effectiveness on the largest model (QwQ-32B) was not deeply analyzed, representing a missed opportunity for deeper insight.

**Questions:**

1.  Dataset Selection and Model Capability: The datasets GSM8K and MATH500 represented a clear easy/hard distinction for earlier models. However, for the powerful models tested in this paper (like DS-Qwen14B), this gap may be less pronounced. Have you considered evaluating your framework on a more challenging dataset, such as AIME (American Invitational Mathematics Examination)? This could reveal more nuanced phenomena and further solidify the paper's conclusions regarding model behavior on complex, multi-step reasoning problems.
2.  Figure 3 Readability: The visualization in Figure 3, which displays the Pareto curves, is quite dense and difficult to read clearly. While the information is present, have you considered alternative ways to present this data? A revised visualization could significantly improve the clarity and impact of your results.
3.  Dynamic Parameter `k`: The step-length parameter `k` is sensitive to the model and task. Have you considered a dynamic adjustment method to make the framework more "plug-and-play"?
4.  Generalization to Other Tasks: What is the framework's potential on non-mathematical tasks like code generation or summarization, and how would the "ReasonType" taxonomy need to adapt?

---

> ### Author Response · Authors · 2025-11-21
>
> Thank you for your review comments and feedback on our work. We appreciate your questions and your concerns about our work. It helps us to better understand our weaknesses and gives us ways of enhancing and strengthen our claims.
>
>
> # Answer to questions:
>
>
> ## Questions 1 and 4:
>
> First, we would like to address your concerns on the generalisation of our framework to other datasets (i.e. questions 1 and 4). To do so, we conducted additional experimentation on the DS-Qwen14B model, on the AIME dataset (as you suggested), and on the MMLU-Pro and GPQA datasets (targeting other tasks than mathematics questions – as suggested by Reviewer 2peB).
>
> - More challenging mathematical dataset: Experimentations on the AIME (22-24) dataset are detailed in Appendix K. Experimentations showed that our framework transfer well when applied to a harder mathematical tasks domain, and show more pronounced results, especially when comparing our framework to the baselines.
>
> - Generalization to Other Tasks: We are wrapping-up our additional experimentations on the GPQA-Diamond and MMLU-Pro datasets and we will include the same analysis as for the AIME dataset in a revised version soon. While your question (4) was about the framework's potential on code generation or summarisation, we selected these two datasets for being suggested by Reviewer 2peB, as well as widely applied for reasoning tasks. We hope that this additional experiment will help to address your concern.
>
>
> ## Questions 2:
>
> Secondly, we uploaded a revised version of the paper, containing an updated version of the Figure 3. We agree that the Figure was hardly readable. To this extend, we suggested an alternative presentation of the results, allowing to better observe the impact of the parameters ($\delta$, $\tau$) on the model’s efficiency. To better distinguish the constraints lying on the Pareto curve, we added shaded regions that follow the steps of the Pareto frontier. Each segment is shaded using the color associated with the constraint (step type lying on the Pareto frontier), creating a more visual representation between the frontier and the constraint categories.
>
>
> ## Questions 3:
>
> Thirdly, we agree with your concern regarding the parameter $k$. We pointed out in Appendix G the challenge of selecting the parameter $k$ depending on the model and the task. We also mentioned, in the Limitation section (Appendix B) of our initial submission, the potential of defining a dynamic value of $k$ for the step definitions. We agree that it could be an interesting alternative approach, but it is not feasible for us to run additional ablation studies due to time constraint. However, we would like to implement it in future work.
>
>
> # Answer to Weaknesses:
>
>
> ## Weakness 1:
>
> The first concern raised was about the subjectivity of the taxonomy. Even though we demonstrated that the taxonomy coupled to the annotation process of the steps by GPT-4o-mini lead to satisfying performance of the BERT Classifiers (see Section 6 and Appendix P), we acknowledge that generating the taxonomy by GPT-4o-mini was a concern of our work as it only relied on the generation of specific models.
>
> To try to address this concern, we provided in Appendix I some details about the creation of the ReasonType taxonomy, as well as an analysis on the reliability of the annotation by GPT-4o-mini. We also would like to emphasis that, while our ReasonType taxonomy might not be optimal, it enabled us to obtain accurate BERT Classifiers (and so monitor the reasoning traces of models accurately), and perform efficient early-stopping.
>
> ## Weakness 2 and 3:
>
> We acknowledge that our framework implies a trade-off: To provide the user with more efficient inferences, our method requires calibration and compute beforehand (which can be viewed as a training process). To better estimate costs and overhead computations implied by our method, we conducted analysis of the inference runtime in Appendix J. Although our experiments are lead offline, we provided an estimation of the runtime of our framework.
>
> We are also planning on adding a training-inference trade-off analysis in a revised version, as suggested by Reviewer a7RC and Reviewer nRdQ.
>
>
> ## Weakness 4:
>
> We agree that our section 7 could be deepened to better understand the weaker performances on the QwQ-32B model. We are currently running additional ablation to address this point and add our results in a revised version.
>
>
> If there are any further changes or clarifications needed, please let us know. Thank you once again for your comments and for your time!

---

> > ### Author Response · Authors · 2025-12-01
> >
> > Again, thank you for your review comments and feedback on our work. Please find bellow additional modifications that we included in our paper to further address your concerns:
> >
> > ## Questions 1 and 4:
> >
> > As well as experimentations on harder mathematical datasets as you suggested (AIME - see Appendix K.1), we conducted experimentations on non-mathematical datasets (GPQA-Diamond and MMLU-Pro) in Appendix K.2. We demonstrated that our framework transfers well to non-math tasks, with 20 to 50% token-count saving on the GPQA-Diamond, and from 25 to 45%, with minimal accuracy drop (-3.8% and -2.3% accuracy drop, respectively). We hope that this additional experiment will help to address your question 4.
> >
> > ## Weaknesses 2 and 3:
> >
> > To further address your concerns regarding the latency and additional computations implied by our calibration step, we added a training-inference trade-off analysis in Appendix J.2. It demonstrated that our Step-Tagging framework can recover
> > its one-time training and calibration cost during inference, and continues to provide additional runtime saving compared to standard inference as more inference tokens are generated. We hope that such analysis can further address your concerns regarding the training cost of our framework.
> >
> > ## Weakness 4:
> >
> > To further explain why our framework reduced effectiveness on the largest model (QwQ-32B), we included additional analysis in Appendix L. We demonstrated that QwQ-32B exhibits a more conservative and less destructive reasoning generation, meaning that it overwrites intermediate correct answer that it generates less frequently compared to the Deepseek models. Because our ST-ES framework aims to stop the generation before correct answers are overwritten, it leaves fewer opportunities to early-stop the QwQ-32B model compared to Deepseek models. We hope that such analysis can further address your concern.
> >
> > If there are any further changes or clarifications needed, please let us know. Thank you once again for your comments and for your time!

---

### Author Response · Authors · 2025-12-02

Dear reviewers,

We thank you for your thoughtful comments, feedbacks and questions. We strongly believe that it helped us to improve our work, and better highlight and validate our claims. As we are moving towards the end of the rebuttal period, we would like to provide you with a summary of the additional experiments and modification that we made compared to our initial submission:

- Generalisation of our Step-Tagging framework: In our initial submission, we focused on the MATH500 and GSM8K for their wide adoption in the field of LRMs. One concern that was common to most of your reviews was that the generalisation to harder mathematical datasets and non-math tasks was not explored. To address this gap, we applied our framework to AIME (harder mathematical dataset - suggested by Reviewer zExx), as well as GPQA-Diamond and MMLU-Pro (non-mathematical dataset - suggested by Reviewer 2peB). Our experimentations are available in the Appendix K, and suggests that our approach as well as our Taxonomy generalize well to harder tasks, and other domains than math.

- Cost analysis: As many of you pointed out, a latency analysis was missing in our paper. As well, we did not analysed the training-inference cost trade-off of our approach. For this reason, we included an extensive analysis of the inference latency (Appendix J.1), and the training cost (including calibration and step-classifiers training) compared to the Step-Tagging Early-stopping inference (Appendix J.2). As well, as suggested by Reviewer nRdQ, we conducted an analysis on the impact of the performance of the BERT router configuration on the efficiency of our framework (Appendix J.3). Our results are available in the Appendix J.

- Generalization and Reliability of the ReasonType taxonomy: To address some of your concerns regarding the reliability of our ReasonType taxonomy and the GPT-4o-mini annotator, we (1) detailed our Taxonomy generation process (Appendix I.1), (2) compared the annotation agreement between different seeds of GPT-4o-mini and other models (Appendix I.2), and (3) applied our ReasonType taxonomy to other reasoning models (Appendix I.3). Our experiments are available in Appendix I.

- Explaining why ST-ES resulted poorly on QwQ-32B: To address this issue, we conducted additional ablation experiments to explain why results were worst on the QwQ-32B model. We demonstrated that QwQ-32B exhibits a more conservative and less destructive reasoning generation, meaning that it overwrites intermediate correct answer that it generates less frequently compared to the Deepseek models. Because our ST-ES framework aims to stop the generation before correct answers are overwritten, it leaves fewer opportunities to early-stop the QwQ-32B model compared to the Deepseek models. Our experiments are available in Appendix L.

- Presentation and conceptual gaps: To address concerns about the conceptual gaps, we provided explicit answers, reworked the Related Work section, and provided comparison Table. In addition, we revised some sections of the paper to ensure that the key findings placed in the appendices are summarized and discussed directly in the main text.

Once again, we deeply thanks the reviewers for their time, valuable reviews, and engagement throughout our rebuttal period. We hope that our modifications mitigated some of their concerns.

---

### Meta-Review · Area_Chair_joR6 · 2026-01-05

**Summary:**

The paper introduces Step-Tagging, a lightweight, online framework that classifies sentence-level reasoning steps using a fine-grained ReasonType taxonomy and leverages tag counts to drive interpretable early stopping during inference. Across multiple open-source reasoning models and benchmarks, the method reduces token usage by roughly 20–50% while preserving comparable accuracy, with larger gains on compute-intensive tasks. The author acknowledged several limitation of the work, e.g., difficulty of selecting hyper-parameter, which are important for the proposed method to work fine. Besides, several raised concerns, e.g., generalization, need more proof and evaluation. I think the paper needs some more refinement.

**Reviewer Concerns:**

Reviewer zExx: Questions the novelty and interpretability of step-aware early stopping and experimental design, while flagging taxonomy subjectivity, calibration complexity, offline evaluation, and weaker gains on QwQ-32B.

Reviewer a7RC: Finds the approach sound and useful but requests clearer latency evidence, easier hyperparameter selection, and a deeper comparison against strong prompt-only baselines given training costs.

Reviewer nRdQ: Values the taxonomy-driven efficiency gains yet criticizes presentation clarity, missing label-quality and router-error analyses, and asks for broader task generalization and fair budgeted comparisons.

Reviewer 2peB: Argues the conceptual novelty over prior CoT segmentation and controllers remains insufficiently distinguished, questions practicality without training data, and doubts net efficiency after per-step tagging overhead.

**Reviewer Scores:**

Some reviewers followed up in the discussion. I do not think the rebuttal has fully convinced the reviewers. The newly added comments are not introducing new evidence or claims. I tend to believe that reviewers will maintain the same scores.

---

### Decision · Program_Chairs · 2026-01-26

Reject